



# Quantifying Dust Deposition over the Atlantic Ocean

Emmanouil Proestakis[1,2], Vassilis Amiridis[1], Carlos Pérez García-Pando[3,4], Svetlana Tsyro[5], Jan Griesfeller[5], Antonis Gkikas[6], Thanasis Georgiou[1,7], María Gonçalves Ageitos[3,8], Jeronimo Escribano[3], Stelios Myriokefalitakis[1], Elisa Bergas Masso[3,8], Enza Di Tomaso[3,a], Sara Basart[3,b], Jan-Berend W. Stuut[9,10], Angela Benedetti[11]

[1]Institute for Astronomy, Astrophysics, Space Applications and Remote Sensing, National Observatory of Athens, Athens, Greece.
[2]School of Chemical and Environmental Engineering, Technical University of Crete, Chania, Greece.
[3]Barcelona Supercomputing Center (BSC), Barcelona, Spain.
[4]Catalan Institution for Research and Advanced Studies (ICREA), Barcelona, Spain.
[5]Research and Development Department, Norwegian Meteorological Institute, Oslo, Norway.
[6]Research Centre for Atmospheric Physics and Climatology, Academy of Athens, Athens, Greece.
[7]School of Physics, Faculty of Sciences, Aristotle University of Thessaloniki, Thessaloniki, Greece.
[8]Universitat Politècnica de Catalunya, Barcelona, Spain.
[9]Royal Netherlands Institute for Sea Research, The Netherlands.
[10]Vrije Universiteit Amsterdam, the Netherlands.
[11]Research Department, European Centre for Medium-Range Weather Forecasts, Reading, UK.
[a] now at: European Centre for Medium-Range Weather Forecasts (ECMWF), Bonn, Germany.
[b] now at: World Meteorological Organisation (WMO), Science and Innovation Department, 1201 Geneva, Switzerland.

*Correspondence to*: Emmanouil Proestakis (proestakis@noa.gr)

**Abstract.** Quantification of atmospheric dust deposition into the Atlantic Ocean is provided. The estimates rely on the four-dimensional structure of atmospheric dust provided by the European Space Agency (ESA) - "LIdar climatology of Vertical Aerosol Structure" (LIVAS) climate data record (CDR) established on the basis of Cloud–Aerosol Lidar and Infrared Pathfinder Satellite Observations (CALIPSO) – Cloud-Aerosol Lidar with Orthogonal Polarization (CALIOP) routine observations. The data record of atmospheric dust deposition rate is provided for the broader Atlantic Ocean region, the Caribbean Sea, and the Gulf of Mexico, confined between latitudes 60°S to 40°N, and is characterized by 5° (zonal) x 2° (meridional) spatial resolution, seasonal-mean temporal resolution, and for the period extending between 12/2006 and 11/2022. The estimates of dust deposition are evaluated on the basis of sediment-trap measurements of deposited lithogenic material implemented as reference dataset with good agreement between the two datasets, revealing the capacity of the satellite-based product to quantitatively provide the amount of dust deposited into the Atlantic Ocean region, as shown by the evaluation intercomparison, evaluation intercomparison characterized by correlation coefficient ~0.79 and mean bias of 5.42 mg/m$^2$d. Moreover, integration of the satellite-based dust deposition rate dataset into AeroVal allows assessment comparison of the variability amongst the dust deposition CDR and dust deposition field estimates provided by the Multiscale Online Nonhydrostatic AtmospheRe CHemistry (MONARCH), EMEP MSC-W, and EC-Earth3-Iron Earth System Models (ESM), with the comparison revealing the capacity of the satellite-based product to follow the seasonal activation of dust source regions





and the four-dimensional migration of dust transport pathways. Overall, the annual-mean amount of dust deposition into the Atlantic Ocean is estimated at $274.79 \pm 31.64$ Tg yr$^{-1}$, of which $243.98 \pm 23.89$ Tg yr$^{-1}$ of dust is deposited into the North

Atlantic Ocean and $30.81 \pm 10.49$ Tg yr$^{-1}$ of dust is deposited into the South Atlantic Ocean. Moreover, a negative statistically significant trend in Atlantic Ocean dust deposition is also revealed. The satellite-based dust deposition CDR is considered unique with respect to a wide range of potential applications, including compensating for geographical and temporal gaps of sediment-trap measurements, supporting evaluation assessments of model simulations, shedding light into physical processes related to the cycle of dust from emission to transport and eventually deposition, and providing a solid basis to better understand

dust biogeochemical impacts on oceanic ecosystems, as well as impacts on weather and climate.

## 1 Introduction

The ocean plays a key role in climate by modulating energy fluxes and exchanging climate-relevant gases with the atmosphere. According to the Intergovernmental Panel on Climate Change Fifth Assessment Report (IPCC AR5 2014), ~90% of the total energy in excess in the atmosphere was absorbed by the ocean between 1971 and 2010. At the same time, gaseous $CO_2$ is

absorbed in the surface layer of the ocean and becomes available for the process of "photosynthesis" performed by phytoplankton cells, contributing through the processes of the "biological carbon pump" and the "solubility carbon pump" (Volk and Hoffert, 1985; Ito and Follows, 2003) to the slowing down of the increase of atmospheric $CO_2$ that results from anthropogenic activities (Raupach et al., 2008). Moreover, phytoplankton abundance and variability regulate Ocean Colour, which in turn determines the extent of light penetration in the water column (Hostetler et al., 2018), affecting sea surface

temperature and resulting in potentially significant ocean-atmosphere feedbacks, such as possible determining the trajectory of tropical storms (Gnanadesikan et al., 2010).

Current estimates of primary production range between 30 and 70 Pg-C per year (Carr et al., 2006; Anav et al., 2013) with spatial distribution depending, among other factors, on the input of nutrients from the atmosphere (Krishnamurthy et al., 2010; Guerreiro et al., 2019; 2023; Myriokefalitakis et al., 2020). Among the key nutrients deposited into the open ocean, nitrogen

(N), phosphorus (P), silica ($SiO_2$), and iron (Fe) are critical for regulating phytoplankton growth, and consequently for modulating marine productivity, ocean colour, and the ocean's capacity to absorb $CO_2$. Among these atmospheric deposited species, Fe availability is the most predominant limiting nutrient for phytoplankton growth over large oceanic areas (Jickells et al., 2005; Okin et al., 2011). Indeed, due to the key role of phytoplankton in the conversion of $CO_2$ into organic carbon and to carbon sequestration, Fe deposition, and more specifically its bioavailable (dissolved) forms (e.g., aqueous, colloidal, or

nanoparticulate), likely plays a major role in oceanic primary productivity (Tagliabue et al., 2017) with impact on the global carbon cycle, hence modulating atmospheric $CO_2$ concentrations (Falkowski et al., 2000; Guerreiro et al., 2021) and in the long term the global climate. Another important biogeochemical parameter to characterize ocean productivity is marine nitrogen fixation, i.e., the reduction of gaseous $N_2$ to ammonium performed by marine organisms. $N_2$-fixing species (e.g.,



diazotrophs) have elevated Fe requirements and their growth may also be Fe-limited over large areas of the Atlantic Ocean
(Pabortsava et al., 2017; Schlosser et al, 2014).

However, iron concentrations in vast areas of the ocean are very low, further enhanced by the characteristic low solubility of iron in seawater (Boyd and Ellwood, 2010). Across the broader surface of the open ocean, aeolian dust is the principal source of Fe (~95%), followed by Fe-containing aerosols from biomass burning and fossil-fuel combustion emissions (Mahowald et al., 2009). It should be noted that observations and laboratory experiments suggest that the solubility of bioavailable dissolved
iron (DFe) in pyrogenic aerosols may be significantly higher than that in lithogenic aerosols, though considerably more sporadic than DFe from mineral dust (Ito et al., 2021). Mineral dust of natural sources, essentially composed of clay, silt, and soil particles (Adebiyi et al., 2023), is mechanically produced by surface winds breaking soil cohesion over surfaces with no vegetation and dry soil such as deserts. According to Yu et al. (2015) and Kok et al., (2021; 2023) North Africa, including the Saharan desert and the Sahel area, is the biggest producer of mineral dust contributing to approximately 50% of the global
atmospheric dust load (~2100 Tg yr$^{-1}$). Other natural sources of dust emission encompassing the South Atlantic Ocean include the desert areas of South Africa and South America, with estimated emissions of ~100 Tg yr$^{-1}$ and 190 Tg yr$^{-1}$, respectively (Kok et al., 2021; 2023). In addition to natural sources, the contribution of dust emitted into the atmosphere from anthropogenic activities remains elusive, with estimated values ranging between 10% and 50% (e.g., Ginoux et al., 2012). Upon emission into the atmosphere mineral dust particles are subject to aeolian transport over distances of thousands of kilometers downwind,
prior removal through wet deposition (i.e. scavenging through precipitation in the water or ice phase), dry deposition/gravitational settling, and turbulent mixing in the Planetary Boundary Layer (PBL) (Gao et al., 2003; Hand et al., 2004; Prospero et al., 2010; Mahowald et al., 2011; Van der Does et al., 2021).

To date, one of the biggest unknowns in the dust cycle remains the amount of atmospheric dust which is actually deposited into the open ocean. Ridley et al. (2012), on the basis of reanalysis models and satellite observations, estimated dust deposition
into the Atlantic Ocean of the order of approximately 218 ± 48 Tg yr$^{-1}$, though the model timeseries covered only two years. Huneeus et al. (2011) performed cross-evaluation of a dozen global aerosol models against observed dust deposition measurements and reported discrepancies of up to an order of magnitude. Kok et al. (2023) estimated that dust emissions from the Saharan, Namib, Kalahari, and Atakama hyper-arid, arid, semi-arid, and dry subhumid areas encompassing the broader Atlantic Ocean contribute to approximately 230 Tg yr$^{-1}$ and 86 Tg yr$^{-1}$ of dust deposition into the North and South Atlantic
Ocean, respectively. The estimated deposition fluxes were based on particles with geometric (volume-equivalent) diameter up to 20 μm, though larger dust particles have been measured transported in the atmosphere (Weinzierl et al., 2016; Ryder et al., 2018) and deposited in the ocean (van der Does et al., 2018), and not accounting for high-latitude dust emission sources (Cvetkovic et al., 2022). Reanalysis datasets of dust deposition, such as the Copernicus Atmosphere Monitoring Service (CAMS; Inness et al., 2019), are available but have not yet been validated with independent observations over the ocean.
Similarly, Yu et al. (2019) on the basis of a ten-year (2007-2016) analysis of Cloud Aerosol Lidar with Orthogonal Polarization (Winker et al., 2010), MODerate resolution Imaging Spectroradiometer (MODIS) (Remer et al., 2005), Multiangle Imaging Spectroradiometer (MISR) (Garay et al., 2020), and Infrared Atmospheric Sounding Interferometer (IASI) (Capelle et al.,





2014) observations estimated the amount of dust deposited into the Tropical Atlantic Ocean in the range 136-222 Tg yr$^{-1}$, though the estimations correspond only to the Saharan dust outflow region (5°S - 40°N).

A better understanding and quantification of the atmospheric dust spatiotemporal variability in terms of deposition across the broader ocean would facilitate addressing long-open questions such as the relationship between dust deposition and dust biogeochemical impact processes on oceanic marine ecosystems. For instance, most IPCC-class Earth System Models (ESMs) use simplified climatological representations of dust deposition and of its composition and solubility to account for the effect of atmospheric nutrient inputs on ocean biogeochemical cycles (Aumont et al., 2015; Seland et al., 2020), although it is widely

accepted that dust deposition is by nature highly episodic (Guieu et al., 2014). To date, models of the atmospheric iron cycle employed to simulate atmospheric Fe dissolution are characterized by different levels of complexity: from simple schemes including first-order rate processing constants applied to a globally uniform 3.5% of Fe in dust to more complex ones allowing different types of acidic species to interact with dust that account for mineral-specific dissolution rates and oxalate processing (Myriokefalitakis et al., 2018; Ito et al., 2019). Model-based estimations on the global atmospheric dissolved dust-related Fe

deposition fluxes into the ocean lie in the range 0.2–0.4 Tg-Fe yr$^{-1}$ for present-day conditions (Myriokefalitakis et al., 2018; Ito et al., 2019), a factor of ~2 higher than during the preindustrial times (Scanza et al., 2018; Bergas-Massó et al., 2023). However, atmospheric dust transport and deposition are also highly variable. Overall, is has been reported that dust mass increased up to 55±30% since the preindustrial era (Kok et al., 2023). However recent studies debate on the magnitude of dust transport over the Atlantic Ocean, whether there is in recent years a decrease (Ridley et al, 2014) or increase (Cuevas-Agulló

et al., 2023) on emission, transport, and deposition. Furthermore, future scenarios show a decrease in bioavailable Fe deposition in mid- and high-latitudes but an increase in equatorial regions such as the equatorial Pacific, Atlantic, and Indian Ocean. Those increases are sharper and have a broader extension for the SSP370 scenario, characterized by strong anthropogenic emission levels (Bergas-Massó et al., 2023). A potential increase (decrease) in dust transport and deposition over the ocean, could make iron and other nutrients, such as silica and phosphorus, more (less) available for phytoplankton (Gittings et al.,

2024), hence triggering changes in marine primary productivity (Rodríguez et al., 2023) and the oceanic carbon pump (Volk and Hoffert, 1985; Ito and Follows, 2003).

The complex interactions of such mechanics are still not well understood. An integrated approach of modeling, satellite, and in-situ observations is needed to quantify the strength and the spatiotemporal characteristics of mineral dust deposition in the surface of the open ocean. An approach allowing to facilitate a better representation of the mechanisms behind the spatial and

temporal variability of atmosphere-ocean interactions, key to interpret observed climatic change responses, and to better describe the future ones.

To date, satellite-based Earth Observations (EO) allow to address the link between atmospheric aerosol composition and ocean deposition over extensive areas and temporal periods. It is important though to note that aerosol optical depth (AOD) or dust optical depth (DOD), as indicators of columnar total aerosol and dust aerosol load respectively, cannot be directly employed

as proxies for total aerosol and dust deposition fluxes, as deposition processes are highly dependent on the vertical structure of aerosols in the atmosphere and the meteorologic conditions (Schepanski et al., 2009; Yu et al., 2013; 2015a; 2019). However,



a wealth of satellite-based observations on the four-dimensional (4-D) distribution of aerosols over oceans has become available during the past decades. The main satellite-based active systems include the Cloud-Aerosol Transport System (CATS) (McGill et al., 2015; Yorks et al., 2016; Proestakis et al., 2019) aboard the International Space Station (ISS; Rodier et al., 2015), the Atmospheric Laser Doppler Instrument (ALADIN) aboard Aeolus (Stoffelen et al., 2005; Ansmann et al., 2007), the joint European Space Agency (ESA) and JAXA's satellite Earth Cloud, Aerosol and Radiation Explorer (EarthCARE; Illingworth et al., 2015), and CALIOP aboard the Cloud-Aerosol Lidar and Infrared Pathfinder Satellite Observation (CALIPSO; Winker et al., 2007; 2009). In addition, novel techniques have been proposed, the one-step polarization Lidar Photometer Networking (one-step POLIPHON; Tesche et al., 2009; Ansmann et al., 2012) developed in the framework of the European Aerosol Research Lidar Network (EARLINET; www.earlinet.org/; last access: 21/08/2024; Pappalardo et al. 2014), allowing to decouple the atmospheric dust component from the total aerosol load and accordingly to estimate the dust mass concentration (Ansmann et al., 2019). To date, the one-step POLIPHON has been extensively to CALIOP and CATS optical products (Amiridis et al., 2013; Marinou et al., 2017; Proestakis et al., 2018; 2024). At the same time, state-of-the-art global atmospheric reanalysis datasets, such as the European Centre for Medium-Range Weather Forecasts (ECMWF) reanalysis (ERA5) (Hersbach et al., 2020) have been established, providing comprehensive climate and weather data, including information on the three-dimensional wind components. Such advances allow through synergetic implementation computation of dust mass fluxes in both zonal and meridional directions over specified oceanic areas and thus estimations of the deposited component. Finally, during the past decades the available observations of the ocean's interior in terms of atmospheric deposited lithogenic material have tremendously increased, thanks to the deployment of arrays of submerged sediment traps (Albani et al., 2016; Van der Does et al., 2016; Korte et al., 2017).

The challenge of the present work is to bring together this wealth of information to provide a full 4D reconstruction of the atmospheric dust component and accordingly estimate the atmospheric dust deposited component across the broader Atlantic Ocean. The paper is structured as follows: Sect. 2 provides an overview of the implemented datasets in terms of both satellite-based EOs and models (Sect. 2.1) and discusses the applied methodology (Sect. 2.2). Sect. 3 provides an overview of the three-dimensional (3-D) spatial distribution and temporal evolution of the dust aerosol in the atmosphere and the corresponding quantification of the dust deposited component across the broader Atlantic Ocean, based on more than 17 years of EOs (2007-2023). Sect. 4 provides a comprehensive evaluation of the EO-based dust deposition rate product, while comparison between the dust deposition rate product and ESMs is provided in Sect. 5. Sect. 6 provides and discusses quantification of the total atmospheric deposited dust mass into the broader Atlantic Ocean and Sect. 7 a summary of the study along with the main concluding remarks.



## 2 Datasets and Methodology

### 2.1 Datasets

#### 2.1.1 CALIPSO-CALIOP

The Cloud-Aerosol Lidar and Infrared Pathfinder Satellite Observation (CALIPSO) environmental satellite is a joint-mission
venture between the United States space agency National Aeronautics and Space Administration (NASA) and the French space
agency Centre National D'Études Spatiales (CNES), developed to provide insight and advance our fundamental understanding
on the role of aerosols and clouds on weather and climate (Winker et al., 2010). CALIPSO was launched on the 28th of April
2006 to join with CloudSat the international Afternoon-Train (A-Train) group of polar-orbiting sun-synchronous satellites
(Stephens et al., 2018), carrying a suite of three Earth-Observing instruments into space: an Imaging Infrared Radiometer (IIR;
Garnier et al., 2017), a wide field-of-view camera (WFC), and the Cloud-Aerosol Lidar with Orthogonal Polarization
(CALIOP) lidar (Hunt et al., 2009). CALIOP, the CALIPSO primal payload, consists of a two-wavelength elastic backscatter
Nd:YAG lidar system, emitting linearly polarized light-pulses at 532 and 1064 nm, and conducting range-resolved
measurements of the parallel and perpendicular backscattered components at 532 nm and of the total backscatter intensity at
1064 nm (Winker et al., 2009).
CALIOP Level 2 (L2) optical products are established on the basis of a sequence of successive sophisticated algorithms
ensuring daytime and nighttime calibration (Powell et al., 2009; Getzewich et al., 2018; Kar et al., 2018; Vaughan et al., 2019),
atmospheric layer detection (Vaughan et al., 2009), and cloud-aerosol discrimination (Liu et al., 2009; Liu et al., 2019; Zeng
et al., 2019). The detected atmospheric features classified as "tropospheric" or "stratospheric" aerosol are further sub-classified
between "marine", "dust", "polluted continental/smoke", "clean continental", "polluted dust", "elevated smoke", "dusty
marine", "PSC aerosol", "volcanic ash", and "sulfate/other" categories (Omar et al., 2009; Kim et al., 2018, Kar et al., 2019),
a classification crucial towards retrieval of extinction coefficient profiles on the basis of backscatter coefficient profiles (Young
and Vaughan, 2009). Here, we use CALIOP L2 (L2) Version 4.5 (V4.5) Aerosol Profile (APro) and Cloud Profile (CPro)
optical products (i.e., backscatter coefficient and particulate depolarization ratio at 532 nm), profile geolocation descriptors
(i.e., longitude, latitude, time), quality-assurance (i.e., Cloud-Aerosol-Discrimination) and atmospheric classification flags
(i.e., feature type, aerosol subtype) along the CALIPSO orbit path, provided at 5 km horizontal resolution and 60 m vertical
resolution, for the Atlantic Ocean geographical domain confined between 40°N and 60°S latitude, and for the temporal period
extending between 12/2006 and 11/2022.

#### 2.1.2 ERA5

The ERA5 dataset is a global reanalysis product that provides estimates of atmospheric, land and oceanic variables from 1950
onward, with continuous updates in near real-time up to the present day (Hersbach et al., 2020). ERA5 is produced by ECMWF,
for the Copernicus Climate Change Service (C3S), by combining historical observations with ECMWF's Integrated Forecast

System (IFS) model. Atmospheric variables are available at a horizontal resolution of 0.25°×0.25° considered as a continuous tiled surface with the point coordinates corresponding to the centroids of the tiles.

In the framework of the study, we use the ERA5 seasonally averaged zonal and meridional components of wind (m/s) from 12/2006 and 11/2022. The vertical resolution of ERA5, in pressure levels between 1000 hPa and 1 hPa, is converted to height above mean sea level based on geopotential (Φ) using an approximation for variation of gravity with altitude and assuming a spherical Earth and no centrifugal force effects (Hobbs, 2006). The ERA5 wind speed parameters in the original regular lat/lon grid in RoI are re-gridded into a uniform spatial grid of 2° latitude by 5° longitude and of seasonal temporal resolution. Figure 1 shows the ERA5 annual mean speed of the horizontal component of wind for the period extending between 12/2006 and 11/2022.

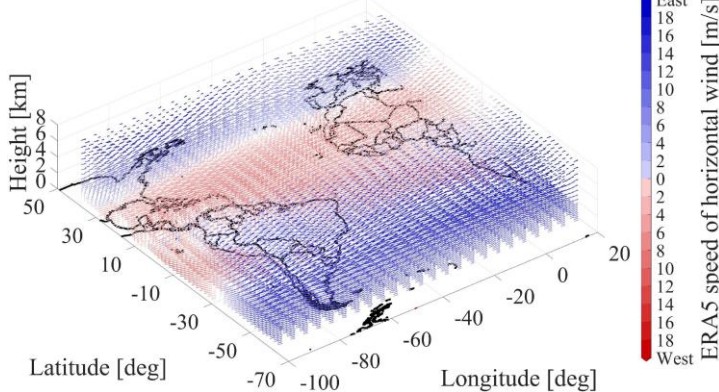

Figure 1: ERA5 annual mean speed of the horizontal component of wind for the period 12/2006 and 11/2022.

### 2.1.3 Dust deposition with the EC-Earth3-Iron Earth System Model

EC-Earth3 (Döscher et al., 2022) is a state-of-the-art Earth System Model (ESM), with a modular structure in which different Earth System components (atmosphere, ocean, sea ice, land surface, dynamic vegetation, atmospheric composition, and ocean biogeochemistry) can be coupled in various model configurations according to the specific scientific needs. In this work, we apply the EC-Earth3-Iron (Myriokefalitakis et al., 2021; Bergas Massó et al., 2022) version, with a configuration that accounts for atmospheric dynamics and land surface processes through the Integrated Forecast System (IFS) cycle 36r4 from the ECMWF, and the interactive simulation of atmospheric aerosols and reactive gas species via the Tracer Model 5 (TM5) (van Noije et al. 2021), including a complex aqueous phase chemistry and an interactive calculation of aerosol and in-cloud pH (Myriokefalitakis et al., 2022). The modal aerosol microphysical scheme M7 (Vignati et al., 2004) represents different aerosol components, and considers internally mixed particles in four water-soluble (nucleation, Aitken, accumulation, and coarse) and three insoluble modes (Aitken, accumulation, and coarse). Dust emission is calculated online, following Tegen et al. (2002, 2004). Freshly emitted dust aerosols are allocated in the accumulation and coarse insoluble modes, but they are allowed to become soluble via atmospheric processing (further details are provided in Table 1).



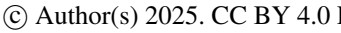

To reproduce close-to-reality present-day climate conditions, the model is executed in an atmosphere-chemistry mode using observed ocean conditions (sea surface temperature and sea ice concentration) as in the Atmospheric Model Intercomparison Project (AMIP) protocol (Gates et al., 1999). Furthermore, the atmospheric circulation is constrained towards ERA5 reanalysis data (Herbach et al., 2020), by adjusting the modeled wind vorticity and divergence through Newtonian relaxation towards the
reanalysis.

The simulation period spans from the year 1991 to 2020. For the 1991-2014 period, anthropogenic and biomass burning emissions are taken from the Coupled Model Intercomparison Phase 6 (CMIP6) historical datasets (Hoesly et al., 2018), while for the 2015-2020, we have selected an intermediate scenario from those defined in the CMIP6 protocol, the SSP2-4.5 (Gidden et al., 2019). The model is executed at the standard spatial resolution, with T255L91 for IFS (approximately 80 km at mid-
latitudes, and 91 vertical levels) and 3°x2° and 34 vertical levels for TM5.

**2.1.4 Dust simulation with the EMEP MSC-W Model**

EMEP MSC-W chemical transport model (herefrom referred to as EMEP model) has been used to perform simulations of the global load and deposition of mineral dust discussed in this paper. The EMEP model is extensively used for air quality assessments in Europe and globally, both for policy related issues and research studies. The parameterisation of windblown
dust emissions from deserts, arid areas, and arable lands used in the EMEP model is based on the works of Marticorena and Bergametti (1995), Marticorena et al. (1997), Alfaro and Gomes (2001), Gomes et al. (2003), and Zender et al. (2003). Dust particles up to 10 μm in diameter, represented by two size fractions (fine and coarse), are presently considered by the model. Aerosol extinction coefficient is diagnosed from the modeled 3d fields of mass concentrations of individual aerosols, including mineral dust, using Mass Extinction Coefficients based on Chin et al. (2002) and Hess et al. (1998).  The EMEP simulations
for the year 2020 have been made on a resolution of 0.5°x0.5°, driven by ECMWF IFS 3-hourly meteorological fields for the actual year. Dry and wet deposition of mineral dust, as well as DOD have been outputted on a daily basis. A comprehensive description of the EMEP MSC-W model can be found in Simpson et al. (2012), Simpson et al. (2024, in preparation), with further details provided in Table 1.

**2.1.5 Dust simulation with the MONARCH model**

The Multiscale Online Nonhydrostatic AtmospheRe CHemistry (MONARCH) model (Klose et al., 2021 and references therein) is a fully coupled atmosphere-chemistry model developed by the Earth Sciences Department of the Barcelona Supercomputing Center (BSC). MONARCH atmospheric dynamics rely on the Non-hydrostatic Multiscale Model on the B-grid (NMMB) (Janjic, 2003; Janjic and Gall, 2012). The model incorporates an advanced representation of the atmospheric dust cycle, including dust generation and uplift by surface winds and turbulence, transport through advection, diffusion and,
vertically, by turbulence and convection. Removal of dust from the atmosphere occurs in MONARCH through gravitational settling in the atmospheric column, dry deposition through turbulent diffusion, as well as via wet deposition, including in-cloud and below-cloud scavenging from stratiform and convective clouds (Pérez et al., 2011, Haustein et al., 2012; Klose et





al., 2021). Dust is represented by a sectional size distribution including 8 log-normal bins that cover up to 20 µm in diameter. For this work, dust emission from saltation is represented following Ginoux et al. (2001) with the modifications described in

Klose et al. (2021), and follows a size distribution at emission invariant with wind speed (Kok et al., 2011). Dust emissions are corrected by factors issued from a modified version of the Local Ensemble Transform Kalman Filter assimilation procedure where we target the correction of emissions in spatial scales of tens to few hundreds of kilometers. This system assimilates dust-filtered AOD from SNPP-VIIRS Deep Blue retrievals in the MONARCH with the dust filtering procedure described in Escribano et al. (2022). The lagged assimilation method uses 20 members, with a moving 15 days assimilation window around

and ahead each estimate. The control vector consists of dust emission scaling factors, which have a temporal resolution of three days and the native spatial resolution of the model. These factors improve the spatial distribution of dust emissions in the model, but they do not change significatively the accumulated emissions of dust in seasonal and yearly scales (Escribano et al., 2023).

Dust interacts with long- and short-wave radiation, which is resolved by the model radiation scheme RRTM-G (Iacono et al.,

2000, 2008). For the experiments presented here, dust particles are assumed non-spherical, long-wave optical properties are derived from the Optical Properties of Aerosols and Clouds (OPAC) dataset (Hess et al., 1998) and short-wave refractive indices consider internal mixtures of different minerals present in dust (Klose et al., 2021).

MONARCH experiments are run at the global scale (with 1°x1.4° horizontal resolution in latitude and longitude, and 48 vertical layers up to 10 hPa), for the year 2020 using more than 1 year of spin-up. The meteorological variables are initialized

every 24 hours from ERA-Interim reanalysis data (Berrisford et al., 2009; Dee et al., 2011), to constrain the modelled circulation towards close-to-reality fields, while the soil variables and dust are kept as calculated by the model in the initialization (further details are provided in Table 1).

Table 1: EC-Earth3-Iron ESM, EMEP MSC-W, and MONARCH configuration.

|  | EC-Earth3-Iron ESM | EMEP MSC-W | MONARCH |
|---|---|---|---|
| Resolution | IFS (T255L91), TM5 (3°x2°,34 layers) | 0.5° x 0.5 ° | 1° in latitude and x1.4° in longitude, 48 vertical layers |
| Meteorology | Online IFS 36r4 | ECMWF IFS 3-hourly | Online NNMB |
| Dust emission scheme | Based on Tegen et al. (2002, 2004) | Based on Marticorena and Bergametti (1995), Marticorena et al. (1997), Alfaro and Gomes (2001), Gomes et al. (2003), and Zender et al. (2003). Dust fluxes are distributed within the lowest, 45 m thick layer. | G01-UST of Klose et al. (2021) |
| Dust size distribution | 2 modes (accumulation, coarse) | 2 size bins (fine and coarse), up to 10 µm. | 8 size bins (with boundaries at 0.2, 0.36, 0.6, 1.2, 2.0, 3.6, 6.0, 12.0, 20.0 µm of diameter) |





| Dust dry deposition | Resistance based dry deposition scheme (land-use dependent), gravitational settling from the lowest layer | Landuse dependent dry deposition, gravitational settling from the lowest layer | Gravitational settling and turbulent diffusion schemes |
|---|---|---|---|
| Dust wet deposition | Scavenging by precipitation in convective clouds and in-cloud and below-cloud scavenging for stratiform clouds. | In-cloud scavenging (based on scavenging ratios) and sub-cloud washout. | In-cloud and below-cloud scavenging from stratiform and convective clouds |
| AOD calculation | Aerosol optical properties consider internal mixtures of the different aerosol components in each mode (van Noije et al., 2014). The refractive indices for dust particles are taken from the aerosol-climate model ECHAM-HAM (Zhang et al., 2012) | Calculated for fine and coarse dust using Mass Extinction coefficients (Chin et al., 2002; Hess et al., 1998). | Aerosol optics consider non-spherical aerosols with refractive indexes computed by representative mineral fractions (Table 6 in Klose et al. 2021) |
| References | van Noije et al. (2021), Myriokefalitakis et al. (2023) | Simpson et al. (2012), Simpson et al. (2024, *in preparation*). | Perez et al. (2011); Klose et al. (2021) and references therein |


## 2.2 Methodology

### 2.2.1 Decoupling the atmospheric pure-dust component from the total aerosol load

Decoupling of the atmospheric pure-dust component from the total aerosol load is performed on the basis of the one-step
POLIPHON (Tesche et al., 2009; Ansmann et al., 2012) technique, established in the framework of EARLINET
(www.earlinet.org/; last access: 21/08/2023; Pappalardo et al. 2014). As discussed and demonstrated in the framework of the
European Space Agency (ESA) - LIdar climatology of Vertical Aerosol Structure for space-based lidar simulation studies
activity (LIVAS; Amiridis et al., 2015) project, proper implementation of the one-step POLIPHON technique to CALIPSO L2
optical products at 532 nm towards derivation of the four-dimensional atmospheric dust climate data record requires a sequence
of intermediate steps, considerations, and assumptions. Figure 2 provides an illustration of the methodology towards
establishing the ESA-LIVAS atmospheric dust product of the CDR, on the basis of a CALIPSO nighttime granule over the
broader Atlantic Ocean on the 12th of August 2012 (Fig. 2a). As shown, two distinct dust plumes emitted from the broader
Saharan Desert and the desert areas of South America were present over the North Atlantic Ocean and South Atlantic Ocean,
respectively (areas "A" and "B" delineated by ellipses in yellow colour).

The first step relates to quality-assurance of the generated CALIPSO-based L2 atmospheric pure-dust product, following the
procedures established in the framework of the official CALIPSO Level 3 (L3) aerosol products (Winker et al. 2013; Tackett
et al., 2018) and the ESA-LIVAS database (Amiridis et al., 2013; Marinou et al., 2017; Proestakis et al., 2018; Proestakis et



al., 2024). The quality-assurance criteria (Table 2) are conservatively selected, aiming to balance between the removal of a significant number of low-quality features and the preservation of the dataset. The quality-screening procedures are iteratively applied to both CALIPSO L2 backscatter coefficient at 532 nm and particulate depolarization ratio at 532 nm profiles, prior

application of the one-step POLIPHON technique.

Table 2: Quality control procedures and filtering applied in CALIPSO data.

| | Quality Assurance procedures |
|---|---|
| 1 | Screen out all cloud features. |
| 2 | Aerosol extinction coefficient for "clear air" assigned equal 0.0 km$^{-1}$. |
| 3 | Screen out atmospheric features of CAD score outside the range [-100, -20]. |
| 4 | Screen out atmospheric features of Extinction QC flag $\neq$ 0, 1, 16 and 18. |
| 5 | Screen out atmospheric features of aerosol extinction uncertainty $\leq$ 99.9 km$^{-1}$. |
| 6 | Screen out misclassified cirrus fringes. |
| 7 | Screen out isolated aerosol features of horizontal resolution 80 km. |
| 8 | Features of large negative extinction coefficient values $\leq$ -0.2 km$^{-1}$, detected $\leq$ 60 m a.g.l., are removed. |
| 9 | Features of large positive extinction coefficient values $\geq$ 2.0 km$^{-1}$, detected $\leq$ 60 m a.g.l., are removed. |
| 10 | Features of large positive pure-dust extinction coefficient values $\geq$ 0.25 km$^{-1}$ (95$^{th}$ percentile) or large negative pure-dust extinction coefficient values $\leq$ -0.25 km$^{-1}$ are removed. |
| 11 | Pure-dust extinction coefficient values above 10 km above mean sea level (a.m.s.l) are removed. |
| 12 | "Clear-Sky" Mode |

Accordingly, the decoupling of the pure-dust and non-dust atmospheric components is performed under two conditions. The

first one relates to the consideration of atmospheric aerosol layers as external mixtures of two distinct aerosol-subtype classes with distinct depolarizing properties. Following the CALIPSO feature-type (Fig. 2b) and aerosol-subtype (Fig. 2c) classification algorithms and towards the overarching objective of decoupling the pure-dust component from the total aerosol load, the separation scheme assumes the "dust", "polluted dust", and "dusty marine" aerosol-subtypes as external mixtures of a "dust" and a "non-dust" component, while the rest of the "tropospheric" and "stratospheric" defined aerosol-subtype classes

is considered virtually dust-free (Kim et al., 2018, Kar et al., 2019). The first consideration, of external aerosol mixtures, allows for the second condition, that the total backscattered signal corresponds to the summation of the parallel and perpendicular backscattered signals by the two aerosol-subtype classes. Under these two conditions, and according to the one-step POLIPHON technique, the contribution of the pure-dust aerosol component to the total aerosol load, in terms of backscatter coefficient, is calculated by Eq. (1).


$$\beta_{\lambda,d}(z) = \beta_{\lambda,p}(z) \frac{\big(\delta_{\lambda,p}(z) - \delta_{\lambda,nd}(z)\big)\big(1 + \delta_{\lambda,nd}(z)\big)}{\big(\delta_{\lambda,d}(z) - \delta_{\lambda,nd}(z)\big)\big(1 + \delta_{\lambda,p}(z)\big)} \qquad (1)$$

In Eq. (1) the parameters "$\delta_{\lambda,p}(z)$", "$\beta_{\lambda,p}(z)$", and "$\beta_{\lambda,d}(z)$" correspond to the particulate depolarization ratio (Fig. 2d), total backscatter coefficient (Fig. 2e), and pure-dust backscatter coefficient (Fig. 2f), respectively, while the input constants of "$\delta_{\lambda,d}$"





and "$\delta_{\lambda,nd}$" correspond to the typical particulate depolarization ratio of the pure-dust and non-dust components of the external

aerosol mixture, expressed as functions of wavelength "λ" and height "z".

(a) CALIPSO nighttime granule 12/08/2012

(b) CALIPSO Feature Type

(c) CALIPSO Aerosol Subtype

(d) Particulate Depolarization Ratio at 532 nm

(e) Total Backscatter Coefficient at 532 nm

(f) Pure-dust Backscatter Coefficient at 532 nm

(g) Pure-dust Extinction Coefficient at 532 nm

(h) Pure-dust Mass Concentration



Figure 2: CALIPSO nighttime granule on the 12th of August 2012 (Fig. 2a), Feature-Type (Fig. 2b), Aerosol-Subtype (Fig. 2c), particulate depolarization ratio at 532nm (Fig. 2d) and total backscatter coefficient at 532nm (Fig. 2e), implemented towards the extracting the pure-dust atmospheric component in terms of backscatter coefficient at 532 nm (Fig. 2f), pure-dust

extinction-coefficient at 532 nm (Fig. 2g), and pure-dust mass concentration (Fig. 2h).

A crucial step towards proper implementation of the one-step POLIPHON technique to CALIPSO L2 optical products at 532 nm with the objective to decouple the pure-dust and non-dust atmospheric components of the total aerosol load is the proper consideration of the $\delta_{\lambda,d}$ and $\delta_{\lambda,nd}$ parameters. With respect to $\delta_{\lambda,d}$, an increasing number of studies report particulate

depolarization ratio measurements of dust-dominant aerosol layers around 0.31 ± 0.04 at 532 nm (Sugimoto et al., 2003; Esselborn et al., 2009; Freudenthaler et al., 2009; Ansmann et al. 2011; Gross et al., 2011; Wiegner et al., 2011; Mamouri et al., 2013; Baars et al., 2016; Hofer et al., 2017; Filioglou et al., 2020), corroborating on the assumption that the particulate depolarization ratio is a characteristic property of dust with little variability on a global scale. However, assumption on the $\delta_{\lambda,nd}$ requires consideration of the particulate depolarization ratio properties of the major non-dust tropospheric aerosol-

subtype categories (i.e., marine, biomass burning smoke, pollen, volcanic ash). The sea salt aerosol category is in general characterized by particulate depolarization ratio values of the order of 2-3 % at 532 nm, increasing however with decreasing Relative Humidity (RH) to values as high as 10-15% at 532 nm in the Marine Boundary Layer (MBL) – Free Troposphere entrainment zone (Haaring et al., 2017). The smoke aerosol category is in general characterized also by low particulate depolarization ratio values, of the order of 1-4 % at 532 nm (Müller et al., 2007b; Nicolae et al., 2013). Significantly higher

particulate depolarization ratio values are reported for pollen and volcanic ash, of the order of 4–15 % and 30-40 % at 532 nm, respectively (Ansmann et al., 2010; Gross et al., 2012; Noh et al.; 2013). However, pollen is usually confined within the Planetary Boundary Layer (PBL), is characterized by high seasonality, and observed in high concentrations at high latitudinal bands outside the domain of the study are, the dust belt, and the major dust transport pathways (Prospero et al., 2002). Volcanic ash emissions, despite the high intensity, are significantly less frequently observed than the dust, marine, smoke, and pollen

aerosol categories. Thus, an average $\delta_{nd}$ effect of 0.05 ± 0.02 at 532 nm is assumed for the broader non-dust aerosol-subtype class in the assumed external aerosol mixture (Marinou et al., 2017; Proestakis et al., 2024 ). Following the consideration of the $\delta_{\lambda,d}$ and $\delta_{\lambda,nd}$ central parameters, the one-step POLIPHON technique allows decoupling of the pure-dust and non-dust components of an assumed external aerosol mixture, of particulate depolarization ratio lower than $\delta_{\lambda,d}$ and greater than $\delta_{\lambda,nd}$, while cases of $\delta_{\lambda,p}(z) \leq \delta_{\lambda,nd}(z)$ are considered dust-free and cases of $\delta_{\lambda,d}(z) \leq \delta_{\lambda,p}(z)$ are considered composed entirely of

dust.

The one-step POLIPHON technique applied to CALIOP L2 optical products at 532 nm results to pure-dust backscatter coefficient profiles at 532nm ($\beta_{\lambda,d}(z)$) along the CALIPSO orbit-path, at uniform horizontal and vertical resolutions of 5 km and 60 m, respectively. Since CALIOP is an elastic backscatter lidar system, to convert the profiles of pure-dust backscatter



coefficient at 532 nm (Fig. 2f) into profiles of pure-dust extinction coefficient at 532 nm (Fig. 2g), suitable pure-dust extinction-

to-backscatter ratio (Lidar Ratio; LR) values are required (Eq. (2))

$$\alpha_{\lambda,d}(z) = LR_{\lambda,d} * \beta_{\lambda,d}(z)$$ (2)

The CALIPSO V4 algorithm assumes a universal LR of 44 sr at 532 nm for dust (Kim et al., 2018). However, recent studies report on the remarkable regional variability of dust LR at 532 nm (Floutsi et al., 2023). More specifically, the broader Atlantic

Ocean is affected in the north mainly by intense loads of dust originating from the Saharan Desert (Prospero, 1999; Kanitz et al., 2014; Marinou et al., 2017; Gkikas et al., 2022) and in the south by dust emissions from the desert areas located in South Africa (Bryant et al., 2007) and South America (Gassó and Torres, 2019). Thus, this study applies suitable dust LR at 532 nm values for the region of interest, towards the development of the atmospheric pure-dust product in terms of extinction coefficient at 532 nm. More specifically, in the domain of the north Atlantic Ocean Saharan dust outflow region a mean LR at

532 nm of 53.1 ± 8 sr is used (Tesche et al., 2009; Gross et al., 2011a; Gross et al., 2011b, Tesche et al., 2011; Kanitz et al., 2013; Kanitz et al., 2014; Gross et al., 2015; Weinzierl et al., 2016; Haaring et al., 2017; Rittmeister et al., 2017; Bolhmann et al., 2018; Floutsi et al., 2023), in the domain of South America a mean LR at 532 nm of 42 ± 17 sr is used (Kanitz et al., 2013), and the default CALIPSO V4 dust LR of 44 sr at 532 nm is used in the intermediate Atlantic Ocean region (Kim et al., 2018). Finally, regionally-dependent EARLINET-established extinction coefficient at 532 nm (Fig. 2g) to mass concentration

conversion factors (Ansmann et al., 2019) and typical particle density of $\rho_d$: 2.6 gcm$^{-3}$ for dust (Ansmann et al., 2012) are applied towards establishing the final pure-dust mass concentration product (Fig. 2h) along the CALIPSO orbit-path (Eq. (3)).

$$MC_d(z) = \rho \cdot c_{v,d} \cdot a_d(z)$$ (3)

Accordingly, a uniform spatial grid of 2° latitude by 5° longitude is established, for the Atlantic Ocean domain extending

between -60ºS and 40ºN. Iteration through all pure-dust mass concentration profiles within each 2°×5° grid is performed, to establish for each grid seasonal-mean atmospheric profiles of quality-assured pure-dust mass concentration, grouped by seasons (December-January-February, DJF; March-April-May, MAM; June-July-August, JJA; and September-October-November, SON) and for the period 12/2006-11/2022. The final dataset provides a four-dimensional (4D) reconstruction of the atmosphere, as shown in Figure 3, in terms of annual-mean total backscatter coefficient at 532 nm (Fig. 3a), pure-dust

backscatter coefficient at 532 nm (Fig. 3b), pure-dust extinction coefficient at 532 nm (Fig. 3c), and pure-dust mass concentration (Fig. 3d), with the later one subsequently implemented towards computation of the atmospheric pure-dust component deposited into the broader Atlantic Ocean.



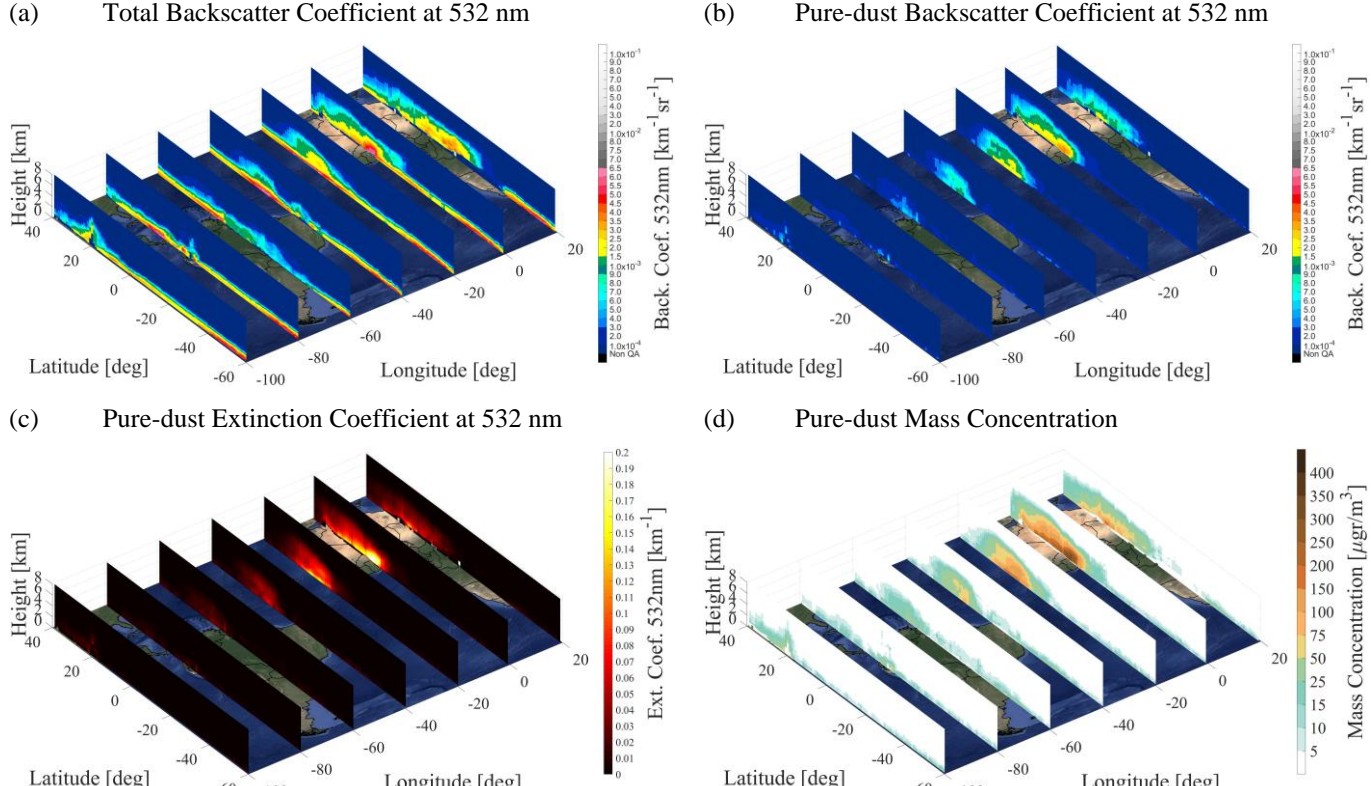

Figure 3: Annual-mean four-dimensional (4D) reconstruction of the atmosphere in terms of total backscatter coefficient at 532 nm (Fig. 3a), pure-dust backscatter coefficient at 532 nm (Fig. 3b), pure-dust extinction coefficient at 532 nm (Fig. 3c), and pure-dust mass concentration (Fig. 3d) in 1°×1° spatial resolution and for the period 12/2006-11/2022.

### 2.2.2 Extracting the atmospheric pure-dust component deposited into the Atlantic Ocean

The present section aims to capitalize on the established 4D reconstruction of the pure-dust atmospheric component in terms of mass concentration profiles, based on ERA5 zonal (U) and meridional (V) wind components, to extract a multiyear satellite-based estimation of the atmospheric pure-dust component deposited into the broader Atlantic Ocean, following the approach suggested by Yu et al. (2019). The estimation of pure-dust deposition fluxes is realized through several successive discrete steps, as visualized in Figure 4 for the indicative grid case extending between 20° and 22° latitude and -30° and -25° longitude of JJA 2020.

The first step accounts for the different vertical resolutions of the established LIVAS atmospheric pure-dust aerosol product and ERA5 horizontal wind components. More specifically, the pure-dust mass concentration atmospheric product follows the original high vertical resolution of CALIPSO of 60 m and 180 m resolution for the altitude range of −0.5-20.2 km and 20.2-30 km a.m.s.l., respectively (Sect. 2.1.1), while the ERA5 U and V wind components are provided in 37 pressure levels between 1000 hPa and 1 hPa, converted to height above mean sea level (Sect. 2.1.2). Thus, the first step performs a reconstruction of



the LIVAS pure-dust mass concentration product, from the higher vertical resolution of CALIPSO (Fig. 4a) to the lower

vertical resolution of ERA5 (Fig. 4c). However, the zonal and meridional atmospheric dust transport and the seasonal transition

of atmospheric transport pathways highly depend on meteorological conditions (Prospero et al., 1987), including among others,

the wind patterns (Fig. 4b). Thus, the second step accounts for decoupling the pure-dust mass concentration atmospheric

product into the zonal (eastward and westward) and meridional (northwards and southward) transported components, on the

400 basis of the (1) magnitude and (2) direction of ERA5 U and V horizontal wind components (Fig. 4c). The third step, on the

basis of (1) the zonal and meridional atmospheric transport components of pure-dust and (2) the U and V wind vector profiles

from ERA5 (Fig. 4c), provides the fluxes of pure-dust both in the zonal and meridional directions (Fig. 4d).

(a)

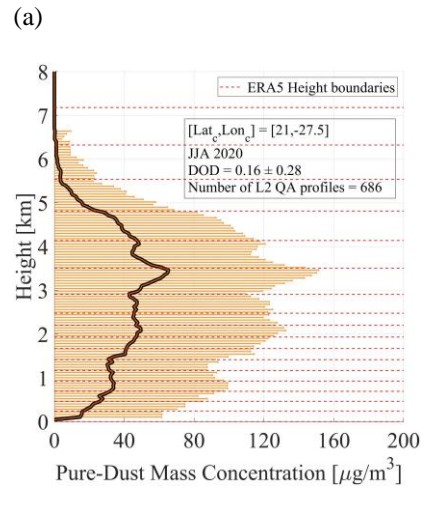

(b)

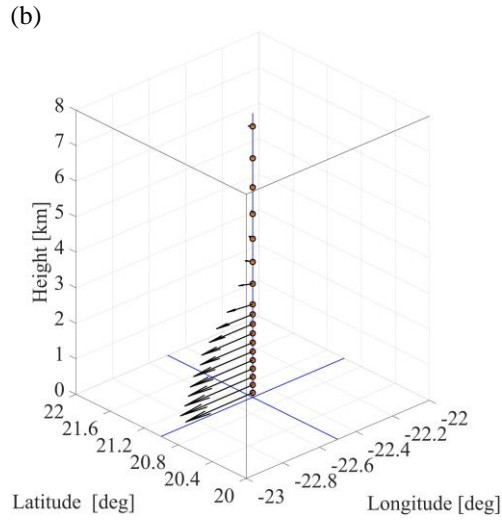

(c)

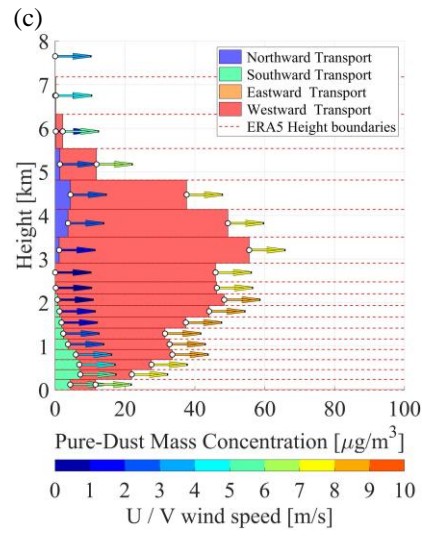

(d)

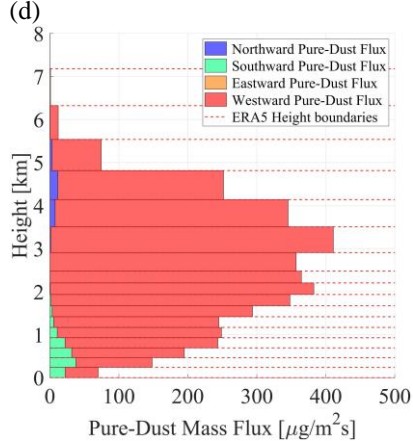



Figure 4: Illustration case of the followed methodology toward the estimation of pure-dust mass fluxes in the zonal and
meridional directions for the Atlantic Ocean area extending between 20° and 22° latitude and -30° and -25° longitude and for
JJA 2020. From top-left to bottom-right: the ESA-LIVAS mass-concentration pure-dust aerosol product (Fig. 4a), the ERA5
U and V profiles of wind (Fig. 4b), the profile of pure-dust mass concentration decoupled into zonal and meridional
atmospheric transport components (Fig. 4c), and the pure-dust mass fluxes in the zonal and meridional directions (Fig. 4d).

As a next step, a three-dimensional (3D) closed cuboid surface is assumed, of 5° length (zonal), 2° width (meridional), and 10
km height (vertical), with the base surface at 0 km a.m.s.l. The approach to extract the atmospheric dust component deposited
into the Atlantic Ocean is based on the condition that the net input-output mass flux through a Gaussian surface, without dust
sources or sinks present in the enclosed volume, should equal zero. Towards this condition, we assume no output or input dust
fluxes from the top surface area of the cuboid, and in general -climatologically- no presence of atmospheric dust at heights
beyond 10 km a.m.s.l. (Marinou et al., 2017), a hypothesis supported by the CALIPSO-based climatology over the region of
interest (not shown). Moreover, we assume no dust sources over the Atlantic Ocean domain, thus no dust input flux from the
surface area of the cuboid. To support this assumption the Clouds and Earth's Radiant Energy System (CERES) International
Geosphere–Biosphere Programme (IGBP) comprehensive map land classification is implemented (Schneider et al., 2013), and
more specifically we consider as Atlantic Ocean grids only surface areas within the domain of interest classified as at least
50% covered by "Water Bodies" and no more than 10% classified as "Closed Shrublands", "Open Shrublands", or "Bare Soid
and Rocks". Accordingly, the input and output pure-dust mass flow rates from-and-towards all neighbouring 3D cuboids
through all neighbouring zonal and meridional surfaces are calculated. Following the conservation of mass and the
aforementioned assumptions, the dust flow rate and accordingly the dust flux through the base of the conceptual cuboid
column, corresponding to the deposited dust, is derived by subtracting from all input components all output components. The
conceptual approach towards the estimating the atmospheric deposited dust component is illustrated in Figure 5, for the
Atlantic Ocean area extending between 20° and 22° latitude and -30° and -25° longitude and for JJA 2020, yielding dust
deposition rate in this case of 192.8 mg/m$^2$d.

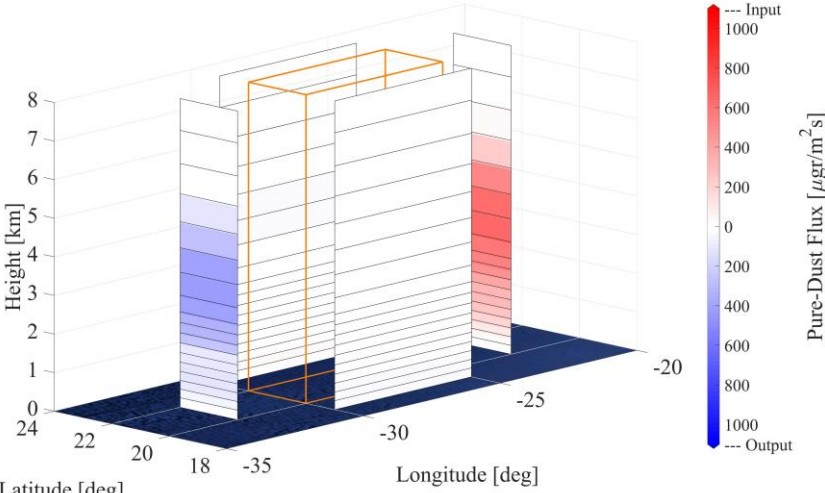

Figure 5: Illustration of the conceptual approach applied towards extracting the pure-dust atmospheric component deposited in the Atlantic Ocean, for the area extending between 20° and 22° latitude and -30° and -25° longitude and for JJA 2020.

The method provides the amount of dust deposited into the ocean based on the estimation of the pure-dust aerosol atmospheric component, however recurrently may result in not physical, either negative or extremely high, values of dust deposition rate.

More specifically, the atmospheric dust component deposited into the Atlantic Ocean (Fig.5) is determined by differentiation of zonal and meridional seasonal-mean profiles of pure-dust mass flow rates from ESA-LIVAS pure-dust climate data record (Fig.4), and thus the applied method is sensitive to CALIPSO orbital characteristics and CALIOP inherited limitations. Factors that may contaminate the 4D atmospheric dust product and subsequently result to not physical retrievals of the dust deposition rate along the dust transport pathways include (1) clouds misclassified as aerosols in terms of feature type classification and

dust layers misclassified as non-dust layers and vice versa in terms of aerosol subtype classification, (2) weighting effects resulting from complete attenuation of CALIOP lidar signal due to the presence of optically opaque atmospheric layers and underestimations due to the presence of tenuous and defuse atmospheric layers of signal-to-noise ratio (SNR) below the minimum detection threshold of CALIOP, and (3) high variability in sampling frequency of QA L2 profiles due to CALIPSO revisit frequency over specific grid areas and cloud-coverage variability. To account for dust deposition rate negative values,

indicating the ocean surface would act as emission source of dust, values lower than zero are masked as not physical. In addition, to account for unrealistic values of dust deposition rate over areas residing outside the documented atmospheric dust transport pathways (Prospero, 1999; Prospero, 2002; Kanitz et al., 2014; Marinou et al., 2017; Gassó and Torres, 2019; Gkikas et al., 2022), grids of virtually non-presence of dust on a seasonal-mean temporal resolution (DOD at 532 nm ≤ 0.01) are considered most probably contaminated by clouds misclassified as aerosols in terms of feature type classification and/or by

non-dust layers misclassified as dust layers in terms of aerosol subtype classification (Burton et al., 2013; Haaring et al., 2017a), and thus not contributing to dust deposited into the ocean. Finally, in order to reduce noise in the final grid, a 3×3



uniform filter is applied, which applies a moving window over the data replacing dust deposition rate values with the mean of the dust deposition rate values inside the moving window.

## 3 Atmospheric Dust and Dust Deposition

The present section aims to capitalize on the performed developments towards decoupling the atmospheric pure-dust component from the total aerosol load (Sect.2.2.1.) and extracting the atmospheric pure-dust component deposited into the Atlantic Ocean (Sect.2.2.2.) to provide (i) a comprehensive reconstruction of the atmospheric dust aerosol component and (ii) of the atmospheric dust aerosol component deposited into the Atlantic Ocean. The products are generated over the broader Atlantic Ocean, encompassing the dust emission sources of W. Africa and S. America, the Atlantic Ocean, Caribbean Sea, and

Gulf of Mexico regions, confined between latitudes 40°N to 60°S and of 5° (zonal) x 2° (meridional) spatial resolution, in a seasonal-mean temporal resolution, and of temporal coverage between 12/2006 and 11/2022. Figure 6 provides the annual- and seasonal- mean horizontal distributions of the pure-dust atmospheric aerosol component (Fig.6-left column) and the corresponding pure-dust component deposited into the ocean along the trans-Atlantic transport (Fig.6-right column).

The pure-dust atmospheric aerosol load is shown in terms of Dust Optical Depth (DOD) at 532 nm (Fig.6-left column),

computed through vertical integration of the L2 annual-mean and seasonal-mean quality-assured pure-dust extinction coefficient profiles at 532 nm within each grid of spatial resolution 2°×5°, providing the columnar pure-dust atmospheric load under cloud-free sky conditions. The standard deviation of the annual-mean and seasonal-mean of the EO-based products is in addition computed, both for atmospheric dust and dust deposition, providing a metric of the spread of dispersion and variability within each grid of spatial resolution 2°×5°. It should be noted that the variability within each grid is usually large,

driven by the large variability of the episodic nature of dust events, highly heterogeneous in both space and time, the large variability in the strength of the emission, atmospheric transport, and removal processes, and the variability of meteorological conditions (Prospero et al., 1987).

Overall, the horizontal distributions of DOD at 532 nm reveal similar patterns, although the magnitude of atmospheric dust load is characterized by high interannual, spatial, and temporal variability, both over land and ocean. Over land, arid regions

of little vegetation and of easily erodible soils contributing to dust life circle and encompassing the broader Atlantic Ocean include the western Saharan Desert in North Africa (Prospero et al., 2002; Huneuus et al., 2011; Marinou et al., 2017, Gkikas et al., 2022), the Etosha Pan (Namibia), Makgadikgadi Pan (Botswana), Kalahari Desert (Namibia, Botswana, South Africa) and Namib Desert (Angola and Namibia) in South Africa (Bryant et al., 2007; Vickery et al., 2013; Gkikas et al., 2022), and the Patagonia Desert (Rio Negro and Chubut provinces) in South America (Gassó and Torres, 2019; Gkikas et al., 2022).

Over the western Saharan Desert persistent intense loads of dust aerosol are observed throughout the year, dust emissions that are considered of both anthropogenic and natural origin (Ginoux et al., 2012). Dust emission mechanisms of anthropogenic origin include processing of soils through agricultural activities, such as cultivation and overgrazing (Ginoux et al., 2012). Natural dust emission micro-scale to synoptic-scale mobilization mechanisms, favoured by the development of the Saharan





heat low (SHL; Schepanski et al., 2017), include pressure gradients (Tyson et al., 1996; Klose et al., 2010), dust devils (Koch
and Renno, 2005), "haboobs" (Knippertz et al., 2007), and low-level jets (LLJ; Fiedler et al., 2013). Though inhomogeneous
in time, space, and of variable strength (Knippertz et al., 2009, 2011), the emission mechanisms over the western Saharan
Desert trigger uplift and accumulation in the atmosphere of several million tonnes of mineral dust (Kok et al., 2023). In terms
of DOD at 532 nm, on a seasonal-mean basis, and following the annual cycle of dust source activation (Washington et al.,
2009), less pronounced dust activity is apparent during DJF (0.159 ± 0.119; Fig. 6c) and higher dust activity during JJA (0.341
± 0.182; Fig. 6g), with intermediate observed DOD values during SON (0.174 ± 0.066; Fig. 6i) and MAM (0.325 ± 0.133; Fig.
6e).

In South Africa, the observed dust activity over the arid areas is characterized by high spatial and temporal variability, however
of lower dust emission strength compared to North Africa arid areas. Activation of the dust sources of Etosha and
Makgadikgadi pans is mostly related to lakes' inundation, resulting to high seasonal variability in dust emission strength
(Gkikas et al., 2022), while over the Namib Desert and along the broader Namibian coastline, dust emission activity is mainly
related to frequent berg winds (katabatic winds) blowing from inland Plateaus and towards the Atlantic Ocean, resulting to
low intra-annual variability in dust emission strength throughout the year (Eckardt and Kuring, 2005). Dust activity over the
dust source regions of this area is more pronounced primarily during SON (DOD at 532 nm of 0.011 ± 0.007; Fig. 6i),
secondarily during JJA (DOD at 532 nm of 0.008 ± 0.003; Fig. 6g) and DJF (DOD at 532 nm of 0.008 ± 0.004; Fig. 6c), and
of significantly lower dust activity during MAM (DOD at 532 nm of 0.005 ± 0.002; Fig. 6e). The observed annual-mean DOD
at 532 nm is computed at 0.008 ± 0.019 (Fig. 6a).

In South America and the Patagonian Desert, higher strength in dust emissions is observed over the broader arid area extending
between the river basins of Rio Negro and Chubut provinces and the southern end of the desert (McConnell, et al., 2007;
Mazzonia and Vazquez, 2009). Over this area, annual-mean DOD of 0.014 ± 0.024 (Fig. 6a) are observed. With respect to
seasonal dust activity, higher DOD values are recorded during SON (0.017 ± 0.008; Fig. 6i) and DJF (0.014 ± 0.004; Fig. 6c)
and lower during JJA (0.013 ± 0.008; Fig. 6g) and MAM (0.009 ± 0.003; Fig. 6e).

The export of dust layers entrained into the atmosphere and transport across the broader Atlantic Ocean is largely controlled
by the prevailing wind systems and the regional meteorology, shaping the major dust transport pathways (Adams et al., 2012;
Ben-Ami et al., 2012; Amiridis et al., 2013; Marinou et al., 2017; H. Yu et al., 2015b; Proestakis et al., 2024). In the northern
hemisphere, the westwards atmospheric transport of Saharan dust layers is largely modulated by the seasonal latitudinal
migration of the Intertropical Convergence Zone (ITCZ; Schneider et al., 2014). More specifically, during boreal summer
(JJA), the enhanced cyclonic circulation of the Saharan Heat Low (SHL; Schepanski et al., 2017), positioned between the
Hoggar Massif and Atlas Mountains (Lavaysse et al., 2009), increase the strength of the northeasterly Harmattan winds to the
west (Lavaysse et al., 2009; Parker et al., 2005a) and of the south-westerly African monsoon flow to the east (Parker et al.,
2005b), and vice versa (Schepanski et al., 2017). The increased pressure gradients of the SHL affect the position of the African
Easterly Jet (AEJ; Knippertz and Todd, 2012) and the development of the African Easterly Waves (AEW; Jones et al. 2003),
modulating the large-scale circulation systems over North Africa and determining the ITCZ position along the 10°-20°N route





(Fig. 6g), and in addition enhance dust emission and export towards and over the Atlantic Ocean (Schepanski et al., 2009; Doherty et al., 2012). Exported within the Saharan Air Layer (SAL; Carlson and Prospero 1972; Prospero and Carlson, 1981;

Braun 2010; Dunion 2011; Adams et al. 2012; Schepanski et al., 2009), the major transport highway for dust layers across the northern Atlantic Ocean, dust plumes during boreal summer are frequently transported as far as the Caribbean Basin and the coast of North America (Prospero, 1999; Prospero et al., 2014; Van der Does et al., 2018). In the winter season (DJF), the decreasing depth and extent of the SHL and the migration of the center of the cyclonic system southwest of the Hoggar Massif (Schepanski et al., 2017) result in generally lower levels of DOD (Fig. 6c). Moreover, during the winter season, the strong

Harmattan winds export plumes of dust south-westward over the Gulf of Guinea (Engelstaedter et al., 2006), mainly within the Marine Boundary Layer and the lower Troposphere (0-3 km a.m.s.l.) (Adams et al., 2012; Stuut et al., 2005; Tsamalis et al., 2013), where the dust plumes are transported westwards, mainly between the equator and 10°N, reaching as far as the Amazon Basin in South America (Huang et al., 2010; Prospero et al., 2014; Yu et al., 2015). During the intermediate seasons of spring (MAM) and autumn (SON), following the seasonal northward and southward migration of the ITCZ respectively,

dust plumes of intermediate depth and intensity between winter and summer are transported along the 5°-15°N latitudinal band across the North Atlantic Ocean (Fig. 6e/i).

In the southern hemisphere and in the case of dust plumes originating from the arid areas of South Africa (Eckardt and Kuring, 2005; Bryant et al., 2007; Ginoux et al., 2012; Vickery et al., 2013; Gkikas et al., 2022), barotropic easterly waves forming between continental high-pressure systems and the South Atlantic anticyclonic systems (Tyson et al., 1996) result in westwards

export and transport of short-range and short-lived dust layers (Vickery et al., 2013) across the southern Atlantic Ocean (Gkikas et al., 2022), mainly below 600 hPa and primarily in the latitudinal zone extending between 20°S and 10°S. In contrast, the relatively-weak dust plumes (Fig. 6a) (Foth et al., 2019) originating from the arid areas of South America (McConnell et al., 2007; Mazzonia and Vazquez, 2009; Ginoux et al., 2012), under favourable meteorological conditions related to strong easterly winds (Gassó et al., 2010; Gassó and Torres, 2019) are frequently advected over the South Atlantic Ocean, primarily in the

latitudinal zone extending between 50°S and 20°S, where subject to long-range aeolian transport the dust layers may reach as far as 20°W (Fig. 6b). Finally, it should be noted that the presence of low DOD values over the area extending below South Africa and to the east of 5°W (Fig. 6a) most probably results from the extended cloud coverage over the region (Gassó and Torres, 2019) and the presence of cubic-like sea salt emissions in the Marine Boundary Layer – Free Troposphere entrainment zone (Haaring et al., 2017) classified as dust-mixtures due to the increased depolarizing properties in CALIPSO aerosol-

subtype classification scheme (Kim et al., 2018), possible resulting to contaminating effects of the atmospheric dust dataset in this case.


Dust Optical Depth at 532 nm

Dust Deposition Rate (mg/m²d)





Figure 6: LIVAS Dust Optical Depth at 532 nm (DOD; left column) and Dust Deposition Rate (DDR; right column), provided in annual-mean (a and b), DJF (c and d), MAM (e and f), JJA (g and h), and SON (i and j), estimated for the period 12/2006-

11/2022.

The annual and seasonal variations of dust deposition rate (DDR; unit: mg·m$^{-2}$·day$^{-1}$) into the broader Atlantic Ocean, as derived by differentiation of the zonal and meridional pure-dust input/output mass flow rates of the atmospheric pure-dust aerosol component and on the basis of the mass conservation hypothesis, are provided in Figure 6-right column. In general, as

expected, there is a strong seasonality in dust deposition, with higher estimates of dust deposition rate revealed mainly during the hot seasons of the year and in the proximity of the major arid areas encompassing the broader Atlantic Ocean domain, decreasing with increasing distance from the dust emission sources, subject to both dry and wet deposition (Schepanski et al., 2009; Adler et al., 2018; Van der Does et al., 2020). More specifically, the observed spatial and temporal patterns of DDR (Fig. 6-right column) follow the seasonal shifts of the major trans-Atlantic dust transport pathways, both in terms of extent and

intensity, as shown and discussed in terms of DOD at 532 nm (Fig. 6-left column). It should be emphasized though that DDR and DOD spatiotemporal-patterns are not directly compatible, since DDR depends not only on the three-dimensional structure of atmospheric dust and the horizontal components of the wind profiles, but more importantly, on the zonal and meridional divergence of the dust mass fluxes. Hence, higher DDR values are apparent over areas not necessarily characterized by higher DOD values but over areas where the meridional and zonal gradients towards downwind adjusted areas are larger.

In the northern hemisphere (Table 3), dust deposition is largely modulated by the seasonal migration of the ITCZ (Schneider et al., 2014). In the winter and spring seasons, when the Saharan Desert Harmattan northeasterly trade-winds are stronger (Lavaysse et al., 2009; Parker et al., 2005a) and the trans-Atlantic dust transport route follows the latitudinal migration of ITCZ to the south of 10°N and 15°N respectively (Fig.6c/e), considerable amounts of dust are transported towards and over the Gulf of Guinea (Lons: 10°W-15°E / Lats: 6°S-6°N). In this case, removal of dust particles from the atmosphere is largely controlled

by intense rainfall (Schepanski et al., 2009), resulting, mainly through wet deposition, in estimated DDR values as high as 37.24 ± 8.36 mg·m$^{-2}$·day$^{-1}$ (Fig. 6d) and 46.49 ± 15.49 mg·m$^{-2}$·day$^{-1}$ (Fig. 6f), for DJF and MAM respectively. In the summer and autumn seasons, intersection of the weaken Saharan northeasterly and of the amplified Southern Africa southeasterly export pathways over the Gulf of Guinea, result to estimated DDR values as high as 8.09 ± 2.59 mg·m$^{-2}$·day$^{-1}$ (Fig. 6h) and 8.77 ± 5.61 mg·m$^{-2}$·day$^{-1}$ (Fig. 6j), respectively. To the north of the Gulf of Guinea, in the proximity of the western coast of

North Africa - Saharan Desert, high presence of dust is apparent throughout the year, which results in significant amounts of dust deposited into the domain of North-East Atlantic Ocean (Lons: 30°W-10°W / Lats: 10°S-40°N). The fluxes of dust deposition show a maximum of 37.94 ± 55.99 mg·m$^{-2}$·day$^{-1}$ in summer (Fig. 6h) and a minimum of 17.42 ± 17.91 mg·m$^{-2}$·day$^{-1}$ in autumn (Fig. 6j), while during spring (Fig. 6f) and winter (Fig. 6d) seasons intermediate values of 34.01 ± 34.05 mg·m$^{-2}$·day$^{-1}$ and 31.96 ± 27.03 mg·m$^{-2}$·day$^{-1}$, respectively, are observed. Moving further west, in the middle of the tropical Atlantic

Ocean (Lons: 60°W-30°W / Lats: 10°S-40°N), relatively high values of DDR also appear, subject to long-range atmospheric transport of dust (Weinzierl et al., 2016; van der Does et al., 2018; Drakaki et al., 2022) mainly within the SAL (Carlson and



Prospero 1972; Prospero and Carlson, 1981; Braun 2010; Dunion 2011; Adams et al. 2012; Schepanski et al., 2009). More specifically, estimated dust deposition fluxes over the NMAO area equal $19.46 \pm 18.41$ mg·m$^{-2}$·day$^{-1}$ in winter (Fig. 6d), $23.16 \pm 24.97$ mg·m$^{-2}$·day$^{-1}$ in spring (Fig. 6f), $24.04 \pm 29.31$ mg·m$^{-2}$·day$^{-1}$ in summer (Fig. 6h), and $14.29 \pm 16.31$ mg·m$^{-2}$·day$^{-1}$ in

autumn (Fig. 6j). With respect to the North-West Atlantic Ocean (Lons: 100°W-60°W / Lats: 10°N-40°N), during summer relatively high dust deposition fluxes are observed mainly in the Caribbean Sea - Southern United States - Gulf of Mexico area, with DDR values as high as $31.67 \pm 23.85$ mg·m$^{-2}$·day$^{-1}$ (Fig. 6h). However, following the seasonal northward and southward migration of the ITCZ and the weaker export of Saharan dust into the SAL (Schepanski et al., 2017) during spring and autumn seasons, significantly lower values of DDR are observed, equalling $8.77 \pm 7.88$ mg·m$^{-2}$·day$^{-1}$ (Fig. 6f) and $4.74 \pm$

$5.74$ mg·m$^{-2}$·day$^{-1}$ (Fig. 6j), respectively, and reaching a minimum of $1.31 \pm 2.19$ in winter (Fig. 6d). However, despite the relatively low amounts of dust reaching the east coast of South America or even beyond over the mainland (Huang et al., 2010; Prospero et al., 2014; Yu et al., 2015), several studies report on the vital role of the dust-related deposited nutrients (i.e., nitrogen, phosphorus, silica, and iron) to the sustainability of the Amazon rainforest (Koren et al., 2006; Tegen et al., 2006; Ansmann et al., 2009; Ben-Ami et al., 2010; Abouchami et al., 2013; Gläser et al., 2015).

In the southern hemisphere (Table 3), the amount of dust deposited into the broader south Atlantic Ocean is significantly lower compared to the northern hemisphere. More specifically, the relatively-weak dust plumes (Fig. 6a) (Foth et al., 2019) originating from the arid areas of South America (McConnell et al., 2007; Mazzonia and Vazquez, 2009; Ginoux et al., 2012) advected over the South West Atlantic Ocean (Lons: 65°W-35°W / Lats: 50°S-22°S), result in dust deposition fluxes over the southwest Atlantic Ocean area of $3.79 \pm 4.02$ mg·m$^{-2}$·day$^{-1}$ in winter (Fig. 6d), $2.97 \pm 2.26$ mg·m$^{-2}$·day$^{-1}$ in spring (Fig. 6f),

$2.17 \pm 1.38$ mg·m$^{-2}$·day$^{-1}$ in summer (Fig. 6h), and $1.41 \pm 2.32$ mg·m$^{-2}$·day$^{-1}$ in fall (Fig. 6j). With respect to dust plumes originating from the arid areas of South Africa (Eckardt and Kuring, 2005; Bryant et al., 2007; Ginoux et al., 2012; Vickery et al., 2013; Gkikas et al., 2022) (Lons: 10°W-15°E / Lats: 14°S-6°S), the westwards export and transport of short-range and short-lived dust layers (Vickery et al., 2013) across the southern Atlantic Ocean (Gkikas et al., 2022) result in relatively low values of dust deposition fluxes, equalling $3.14 \pm 3.59$ mg·m$^{-2}$·day$^{-1}$ in winter (Fig. 6d), $2.61 \pm 2.21$ mg·m$^{-2}$·day$^{-1}$ in spring

(Fig. 6f), $4.42 \pm 2.79$ mg·m$^{-2}$·day$^{-1}$ in summer (Fig. 6h), and $2.28 \pm 1.28$ mg·m$^{-2}$·day$^{-1}$ in fall (Fig. 6j).

Table 3: Seasonal DDR averages (in mg·m$^{-2}$·day$^{-1}$), representative for the period 12/2006-11/2022, along with the associated variability, for the North-East Atlantic Ocean (NEAO), North-Middle Atlantic Ocean (NMAO), North-West Atlantic Ocean (NWAO), Gulf of Guinea (GG), South-East Atlantic Ocean (SEAO), and South-West Atlantic Ocean (SWAO) sub-domains

of the Atlantic Ocean.

| | DDR (mg·m$^{-2}$·day$^{-1}$) | | | |
|---|---|---|---|---|
| | DJF | MAM | JJA | SON |
| North-East Atlantic Ocean (NEAO)<br>(Lons: 30°W-10°W / Lats: 10°S-40°N) | $31.96 \pm 27.03$ | $34.01 \pm 34.05$ | $37.94 \pm 55.99$ | $17.42 \pm 17.91$ |
| North-Middle Atlantic Ocean (NMAO)<br>(Lons: 60°W-30°W / Lats: 10°S-40°N) | $19.46 \pm 18.41$ | $23.16 \pm 24.97$ | $24.04 \pm 29.31$ | $14.29 \pm 16.31$ |





| | | | | |
|---|---|---|---|---|
| North-West Atlantic Ocean (NWAO) (Lons: 100°W-60°W / Lats: 10°N-40°N) | 1.31 ± 2.19 | 8.77 ± 7.88 | 31.67 ± 23.85 | 4.74 ± 5.74 |
| Gulf of Guinea (GG) (Lons: 10°W-15°E / Lats: 6°S-6°N) | 37.24 ± 8.36 | 46.49 ±15.49 | 8.09 ± 2.59 | 8.77 ± 5.61 |
| South-East Atlantic Ocean (SEAO) (Lons: 10°W-15°E / Lats: 14°S-6°S) | 3.14 ± 3.59 | 2.61 ± 2.21 | 4.42 ± 2.79 | 2.28 ± 1.28 |
| South-West Atlantic Ocean (SWAO) (Lons: 65°W-35°W / Lats: 50°S-22°S) | 3.79 ± 4.02 | 2.97 ± 2.26 | 2.17 ± 1.38 | 1.41 ± 2.32 |

## 4 Evaluation of EO-based Dust Deposition estimates

Towards verifying the accuracy, ensuring the reliability, and quantifying the uncertainties of the satellite-based estimations of dust deposition rate, implementation of in-situ observations of dust deposition fluxes as reference datasets is essential. However, numerous significant challenges inherent to the complex nature of oceanographic research hamper the feasibility of
establishing long-term and continuous in-situ measurements of high spatial coverage over extensive geographical areas and temporal periods, leading to limited availability of observational-reference datasets. In this study, we focus on the Albani et al. (2016) compiled data record of in-situ dust deposition flux measurements, as enriched by Yu et al. (2019) with more recent observations in the region of interest (Table 4). To be more specific, the utilized reference dataset (Albani et al., 2016; Yu et al., 2019) was established through revision and integration of pre-existing in-situ dust deposition flux measurements (Honjo
and Manganini, 1993; Kremling and Streu, 1993; Wefer and Fischer, 1993; Fischer et al. 1996; Jickells et al., 1996; 1998; Kuss and Kremling, 1999; Ratmeyer et al., 1999; Bory and Newton, 2000; Friese et al., 2017; Korte et al., 2017) across the broader Atlantic Ocean (Fig. 7a), through an effort for homogenization (Albani et al., 2016; Yu et al., 2019), resulting in a robust record suitable to be used for scientific studies related to modern climate deposition of dust aerosols (Ginoux et al., 2001; Tegen et al., 2002; Lawrence and Neff, 2009; Mahowald et al., 2009).


Table 4: Compilation of sediment-trap climatologies of dust deposition fluxes.

| Site No (##) | Latitude (deg. North) | Longitude (deg. East) | Observational Temporal Period | DDR (mg/m$^2$d) | Reference |
|---|---|---|---|---|---|
| 1. | 21.93 | -25.23 | 1986-1987 | 18.36 | Kremling and Streu, 1993; Jickells et al., 1996 |
| 2. | 21.15 | -20.69 | 1989-1990 | 56.11 | Ratmeyer et al., 1999; Wefer and Fischer, 1993 |
| 3. | 21.15 | -20.68 | 1989-1990 | 53.97 | Fischer et al. 1996; Wefer and Fischer, 1993 |
| 4. | 19 | -20.17 | 1990-1991 | 59.04 | Bory and Newton, 2000 |
| 5. | 18.5 | -21.08 | 1991 | 51.34 | Bory and Newton, 2000 |
| 6. | 11.48 | -21.02 | 1992-1993 | 61.97 | Ratmeyer et al., 1999 |
| 7. | 20.92 | -19.75 | 1988-1989 | 60.05 | Fischer et al., 1996; Wefer and Fischer, 1993 |
| 8. | 20.92 | -19.74 | 1989-1990 | 74.52 | Jickells et al., 1996; Wefer and Fischer, 1993 |





| | | | | | |
|---|---|---|---|---|---|
| 9. | 20.92 | -19.74 | 1990-1991 | 31.21 | Ratmeyer et al., 1999; Wefer and Fischer, 1993 |
| 10. | 29.11 | -15.45 | 1991-1992 | 11.37 | Ratmeyer et al., 1999; Wefer and Fischer, 1993 |
| 11. | 1.79 | -11.13 | 1989-1990 | 11.78 | Wefer and Fischer, 1993 |
| 12. | -2.18 | -9.9 | 1989-1990 | 3.29 | Wefer and Fischer, 1993 |
| 13. | 33.15 | -21.98 | 1993-1994 | 3.21 | Kuss and Kremling, 1999 |
| 14. | 33.15 | -21.98 | 1993-1994 | 6.41 | Kuss and Kremling, 1999 |
| 15. | -20.05 | 9.16 | 1989-1990 | 6.85 | Wefer and Fischer, 1993 |
| 16. | -20.5 | 9.16 | 1989-1990 | 10.41 | Wefer and Fischer, 1993 |
| 17. | -20.07 | 9.17 | 1989-1990 | 15.89 | Wefer and Fischer, 1993 |
| 18. | 32.08 | -64.25 | 1981-1991 | 5.21 | Jickells et al., 1998 |
| 19. | 21.05 | -31.17 | 1991-1992 | 10.22 | Bory and Newton, 2000 |
| 20. | 24.55 | -22.83 | 1990-1991 | 14.27 | Jickells et al., 1996 |
| 21. | 28 | -21.98 | 1990-1991 | 6.58 | Jickells et al., 1996 |
| 22. | 33.82 | -21.02 | 1989-1990 | 13.01 | Honjo and Manganini, 1993 |
| 23. | 13.81 | -37.82 | 2013 | 12 | Korte et al., 2017 |
| 24. | 12.39 | -38.63 | 2013 | 23 | Korte et al., 2017 |
| 25. | 12.06 | -49.19 | 2013 | 20 | Korte et al., 2017 |
| 26. | 12.02 | -57.04 | 2013 | 52 | Korte et al., 2017 |
| 27. | 12 | -23 | 2013 | 47 | Korte et al., 2017 |

This section aims to quantitatively evaluate the ability of the satellite-based derived dust deposition rate product to replicate the characteristics of dust deposition measured by sediment traps, while also quantifying associated uncertainties. However, it

should be emphasized that implementation of the in-situ observational dataset as reference record towards a direct and rigorous validation of the satellite-based dust deposition rate product established is not feasible, since most of the sediment-trap measurements date back to the 1980s and 1990s. The assessment follows the methodology of Yu et al. (2019) applied to account for the different temporal spans of sediment-trap measurements and satellite-based observations. More specifically, under the apparent limitations, the intercomparison is conducted on the basis of long-term climatological means, evaluating

the total dataset of spatially collocated satellite-based dust deposition rates against the correlative in-situ measurements (Fig. 7b), for the Atlantic Ocean sediment-trap sites (Fig. 7a). Overall, the performed evaluation reveals a general tendency of the satellite-based dust deposition rate product to overestimate the dust deposition rate measurements conducted at the sediment-trap observational sites. This feature can be attributed to several sources and factors driving the spatially correlative evaluation, as discussed hereinafter, resulting into the observed discrepancies.


Table 5: EO-based Dust Deposition Rate product overall evaluation metrics established on the basis of reference dust deposition rate measurements conducted at sediment-trap observational sites, including absolute biases, relative biases, Root Mean Square Error (RMSE), Correlation Coefficient, Slope ($S_{fit}$) and Interception ($I_{fit}$) of liner regression fit.

| Cor. Coef. | RMSE (mg/m²d) | Relative Bias (%) | Mean Bias (mg/m²d) | $S_{fit}$ | $I_{fit}$ (mg/m²d) |
|---|---|---|---|---|---|
| 0.79 | 15.97 | 19.82 | 5.42 | 0.85 | 9.49 |





First, with respect to CALIPSO-CALIOP observations, issues contaminating the dust deposition rate product resulting in the apparent overestimations can be attributed, among others, to misclassification of cloud layers (i.e., cirrus fringes) as aerosol layers (Liu et al., 2009; 2019) even under strict quality assurance criteria (Tackett et al., 2018) and to further erroneous subclassification of the classified layers as aerosol (Omar et al., 2009; Kim et al., 2018; Kar et al., 2019). For instance, marine aerosol layers (i.e., sea salt emissions) in the Marine Boundary Layer (MBL) - Free Troposphere (FT) entrainment zone

characterized of low relative humidity (RH) conditions tend to obtain a crystal cubic-like shape (Haarig et al., 2017a), leading due to increased depolarizing capacity to aerosol subtype classification distinction ambiguities (Burton et al., 2013).

Second, the application of the atmospheric dust decoupling technique on optical products of the CALIPSO mission (Winker et al., 2010), namely the one-step POLIPHON (Tesche et al., 2009), is performed on the basis of several assumptions with respect to the depolarizing properties of the dust (Esselborn et al., 2009; Freudenthaler et al., 2009; Ansmann et al., 2011; Groß

et al., 2011, 2015; Tesche et al., 2011; Veselovskii et al., 2016; Haarig et al., 2017b) and non-dust aerosol layers (Müller et al., 2007b; Ansmann et al., 2010; Groß et al., 2012; Nicolae et al., 2013; Noh et al., 2013; Haarig et al., 2017a; Bohlmann et al., 2021; Veselovskii et al., 2022), introducing further uncertainties in the final near-global atmospheric dust aerosol product (Amiridis et al., 2013; Marinou et al., 2017; Proestakis et al., 2018; 2024; Aslanoğlu et al., 2022). More significant is considered the impact of incorrect implementation of dust lidar ratio values for the layers classified as dust aerosol, allowing to convert

backscatter coefficient at 532 nm profiles to extinction coefficient at 532 nm profiles and accordingly to mass concentration profiles (Ansmann et al., 2019). For instance, implementation of the CALIPSO-default Version 4 (V4) dust aerosol subtype lidar ratio of 44 sr (Kim et al., 2018) and not of the updated for Saharan dust lidar ratio of 53.1 sr for North Atlantic Ocean dust outflow region (Floutsi et al., 2023) would reduce the rate of dust deposition and the estimate of the total deposited dust in the North Atlantic Ocean region up to ~20.7%. However, several studies and extensive experimental campaigns report

significantly higher Saharan dust lidar ratio values for the West Saharan Desert and the North Atlantic Ocean dust outflow region (Tesche et al., 2009; 2011; Groß et al., 2011a; 2011b; 2015; Kanitz et al., 2013; 2014; Weinzierl et al., 2016; Haarig et al., 2017b; Rittmeister et al., 2017; Bohlmann et al., 2018; Floutsi et al., 2023) than the CALIPSO V4 default (Kim et al., 2018), attributed mainly to the different minerology of dust particles emitted from different dust sources into the atmosphere (Castellanos et al., 2024).

Third, marine sediment trap observations of atmospheric dust deposition into the ocean are challenging to interpret (e.g., Kohfeld and Harrison 2001) as the provided measurements do not necessarily represent the true atmospheric dust deposition to the surface of the ocean at the monitoring locations of the mooring sites (Siegel and Deuser 1997, Bory et al. 2002). One of the main sources of uncertainties in the intercomparison of satellite-based dust deposition products and sediment trap records is the time lag between dust emission and export towards and over the broader Atlantic Ocean, related to the EO-based

estimation of dust deposition, and the arrival time at the depth of the operated sediment traps. More specifically, it is estimated that the Saharan dust layers leaving the African continent are transported westward at a speed of approximately 1000 km day$^{-1}$, needing about 5 to 7 days to cross as far as the eastern coast of the United States or the Gulf of Mexico (Huang et al. 2010;



Prospero et al., 2014; Weinzierl et al., 2016). Upon deposition, the atmospheric dust layers release into the ocean dust particles characterized by a size distribution spanning over several orders of magnitude, extending between 0.1 μm and more than 100

μm in diameter (Weinzierl et al., 2016; Ryder et al., 2018; an der Does et al., 2018). However, the different size of the deposited dust particles is related to different dust transport pathways through the water column, since finer particles are characterized by slower settling speed (Van der Jagt et al., 2018; Guerreiro et al., 2021). More specifically, it is estimated that finer dust particles, with sinking speeds of about 1 to 35 m d$^{-1}$ would have a lateral transport range during this period of more than 500 km from the oceanic surface deposition area, depending also on the sampling depth (Siegel et al. 1990). Overall, it is estimated

that the sinking speed of deposited particles towards the installed collectors operated at ~1200m leads to a transit period of ~5 days, while for deeper depth collectors the transit period ranges between 10 to 20 days (Fischer et al., 1996; Ratmeyer et al, 1999; Van der Does et al., 2016), introducing uncertainties on the temporal intercomparison process of the satellite-based dust deposition products and sediment trap records. With respect to the spatial intercomparison, during the transit period between atmospheric deposition and collection by the sediment traps, deposited dust layers are subject to currents of variable pathways.

For instance, upon deposition of Saharan dust to the proximity of the western coast of the Saharan desert, off Cape Blanc, the oceanic circulation carries the deposited layers thousands of kilometers southwards where the Canary Current meets either the westward North Equatorial Current (NEC) or the eastward North Equatorial Counter Current (NECC), possible resulting not only to extended diffusion of the deposited layers but also to different oceanic transport pathways, thus to collection of the deposited atmospheric dust particles over a relatively well-specified area by sediment traps established and operated over

distances of hundreds of kilometers apart (Ratmeyer et al, 1999).

During this period, deposited dust particles may be subject to several mechanic and chemical alteration processes, of hydrodynamic nature (i.e., fractionation and sorting; McCave et al., 1995), remineralization, coagulation or aggregation, disaggregation or decomposition (Duce et al., 1991; Jickells et al., 2005; Ratmeyer et al, 1999; Korte et al., 2017) or dissolution and microbiological partial disintegration (Alldredge et al., 1990). The temporal intercomparison may be further affected by

the oceanic seasonality, as reported by time-series of lithogenic fluxes in sediment traps, which is substantially different to the seasonality of atmospheric dust emission, transport, and deposition (Amiridis et al., 2013; Gkikas et al., 2022; Proestakis et al., 2024). More specifically, on the basis of an array of five moorings deployed below the SAL, Van der Does et al. (2016) and Korte et al. (2017) reported that several of the installed moorings demonstrated a clear seasonality in lithogenic fluxes, mainly during summer and autumn (Van der Does et al., 2016), while in other cases of sediment traps no clear seasonality was

evident (Korte et al., 2017). The outcomes of oceanic medium to weak seasonality support documented findings as reported by Ratmeyer et al. (1999) on the basis of lithogenic samples collected by several sediment traps operated in the Cape-Verde - Cape Blanc - Cape Verde Islands broader area.

Another source of uncertainties arises by the variable methods used to extract the dust-related component of the total mass collected by the sediment traps, containing quartz, clay minerals, and feldspars parts, the so called lithogenic fraction (Wefer

and Fischer, 1993; Fischer and Wefer, 1996; Neuer et al., 2002; Fischer and Karakas, 2009; Fischer et al., 2016). As discussed by McCave et al. (1995) and Ratmeyer et al. (1999), the comparison between different methods, also on the basis of the applied





instrumentation, is especially challenging, mainly due to the complexity of the analysis techniques and the detectable size ranges. Moreover, the detectable sensitivity may result to substantial differences (underestimation) against the EO-based dust deposition rate product established in the framework of the present study. Since a sufficient overlap of the sediment trap

lithogenic fraction sizes and the atmospheric dust PSD reported on the basis of extensive experimental campaigns over the Atlantic Ocean on the basis of airborne in situ measurements is not accounted for, a more direct and accurate comparison is not feasible. More specifically, while measured atmospheric dust PSDs range between 0.1 and 100 μm in terms of diameter (Weinzierl et al., 2016; Ryder et al., 2018), frequently measurements of so broad PSDs with sediment trap techniques, especially during the past decades, was particularly challenging. For instance, Ratmeyer et al. (1999) volume distribution

spectra analysis and grain size statistics were performed only for the fraction 6-63 μm, in order of the provided sediment trap record to be comparable with previously established grain size distributions with surface sediments in the proximity of the northwest Africa (Koopmann, 1981; Sarnthein et al., 1982; McCave et al., 1995).

Moreover, oceanic conditions are not always favourable, with strong currents ($> 12$ cm s$^{-1}$) and deep eddy penetration resulting to vertical displacement of mooring line possible contaminating the unbiased collection of settling particles (Knauer and Asper,

1989; Korte et al., 2017). It should be mentioned that though dust sedimentation measurements are frequently used to support quantitatively evaluation efforts of model outputs or satellite-based products, time-series of records are frequently partially interrupted or incomplete (Ratmeyer et al., 1999). Due to the highly episodic nature of dust (Prospero et al. 1987; Mahowald et al. 2003), evidence suggest that in some cases the available records are based on a relatively small number of dust events (Avila et al., 1997). It should be noted that temporal and spatial sampling limitations of dust deposition rate measurements

conducted at the sediment-trap observational sites, related to inherent challenges of oceanic research, may result to lack of observations over extensive geographical areas or to not full-year coverage, thus to records not fully representative of dust deposition patterns (Yu et al., 2019).

Finally, the sediment-trap observations record and the EO-based dust deposition product refer to different time spans. More specifically, the majority of the performed sediment-trap measurements were carried out in the 1980s and 1990s, approximately

a decade before CALIPSO optical products became available. Thus, the performed intercomparison serves more as an indirect evaluation of the satellite-based product of dust deposition rate against the sediment-trap observations rather than a rigorous validation.







(a)

(b)

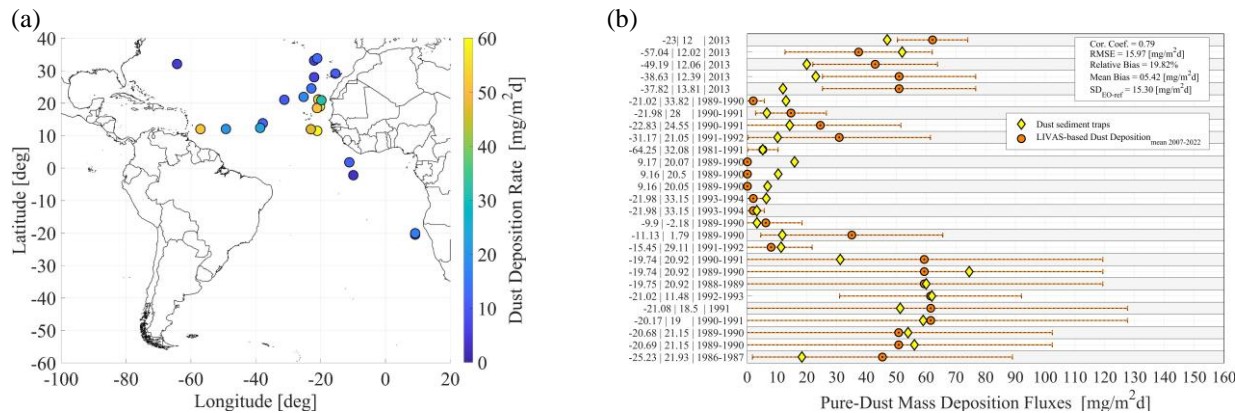

Figure 7: In-situ dust deposition flux measurements and locations of sediment-sites (Table 4) (a). Long-term evaluation of the satellite lidar-based Dust Deposition Rate product against the corresponding in-situ dust deposition fluxes climatology $(mg/m^2d)$ (b).


Overall, considering the several sources of uncertainties and the methodological factors driving the observed discrepancies, the EO-based dust deposition rate product and the sediment-trap observations are in rather good agreement (slope of 0.85, intercept of 9.49 mg/m²/day, and Pearson correlation coefficient ~0.79). The general trend of satellite-based dust deposition is to overestimate those from the in situ provided observations (mean bias of 5.42 mg/m²/day, relative bias of 19.82%, and RMSE

of 30.3 mg/m²/day). However, it is notable that the satellite-based dust deposition rate product consistently reproduces the dust deposition patterns recorded by the sediment traps installed and operated across the broader Atlantic Ocean, with a positional accuracy and magnitude generally within a factor of 2 compared to sediment trap measurements.

## 5 EO-based Dust Deposition Rate and ESMs

The present section aims to compare the EO-based dust deposition rate products and estimates of dust deposition provided by
Earth System Models (ESM). The objective of the comparison lies in identifying common dust spatial and seasonal patterns, addressing whether the EO-based dust deposition rate product shares similar characteristics in terms of spatio-temporal variability with model-based deposition estimates, implemented in AeroVal (https://aeroval.met.no/; last access: 21 January 2025). As an intermediate step, the EO-based atmospheric dust product of the ESA-LIVAS CDR in terms of DOD at 532 nm is compared against AErosol RObotic NETwork (AERONET; https://aeronet.gsfc.nasa.gov/; last access: 3 January 2025;
Holben et al., 1998) coarse-mode AOD optical product derived via the Spectral Deconvolution Algorithm (SDA; Eck et al., 1999; O'Neill et al., 2001a, b, 2003). This step is applied since the capacity of the ESA-LIVAS CDR to provide accurately the spatiotemporal variability of the atmospheric dust conditions is a crucial cornerstone for quantitatively and qualitatively quantifying the dust deposited component across the dust transport over the ocean. The objective of the comparison lies in



identifying common dust spatial and seasonal patterns, addressing whether the EO-based dust deposition rate product shares
similar characteristics in terms of spatio-temporal variability with model-based deposition estimates, implemented in AeroVal.
AeroVal is a web-based platform, developed at the Norwegian Meteorological Institute, designed for the evaluation of climate
and air quality models. The platform employs the pyaerocom library (a successor of AeroCom evaluation and visualization
tool) to collocate model data with observations from a variety of sources, including ground-based observation networks
(EBAS, EEA, AERONET) and satellites (MODIS, AATSR etc). AeroVal allows for the computation of statistics, such as
biases and correlations, and provides an interactive web interface to facilitate easy exploration of data, models intercomparison,
and evaluation statistics. It is utilized in several projects, including among others, the CMIP (Mortier et al., 2020), AeroCom
(e.g. Gliß et al., 2021), CAMS, and EMEP
(https://aeroval.met.no/pages/evaluation/?project=domos&experiment=Dust_AOD; last visit: 20/10/2024).

In this work, the Aeroval tool is used for consistency checks by comparing LIVAS DOD at 532 nm and Dust Deposition Rate
with simulation results from three atmospheric transport models, i.e. EMEP MSC-W (hereafter EMEP; Sect.2.1.4), EC-Earth3-
Iron (EC-Earth3; Sect.2.1.3), and MONARCH (Sect.2.1.5) for the year 2020. As the accuracy of dust deposition calculations
relies on the accuracy of the estimates of the dust load in the atmosphere, DODs from LIVAS and EMEP, EC-Earth3, and
MONARCH models are as a first-step evaluated against AERONET coarse-mode AOD (Dubovik et al., 2006), update
September 2024. The comparison is made on a basis of monthly mean DOD and deposition fluxes. Note that while model
results are produced at a daily resolution, LIVAS data are provided in the framework of the study at seasonal temporal
resolution, so that all three months within a season are assigned the same value.

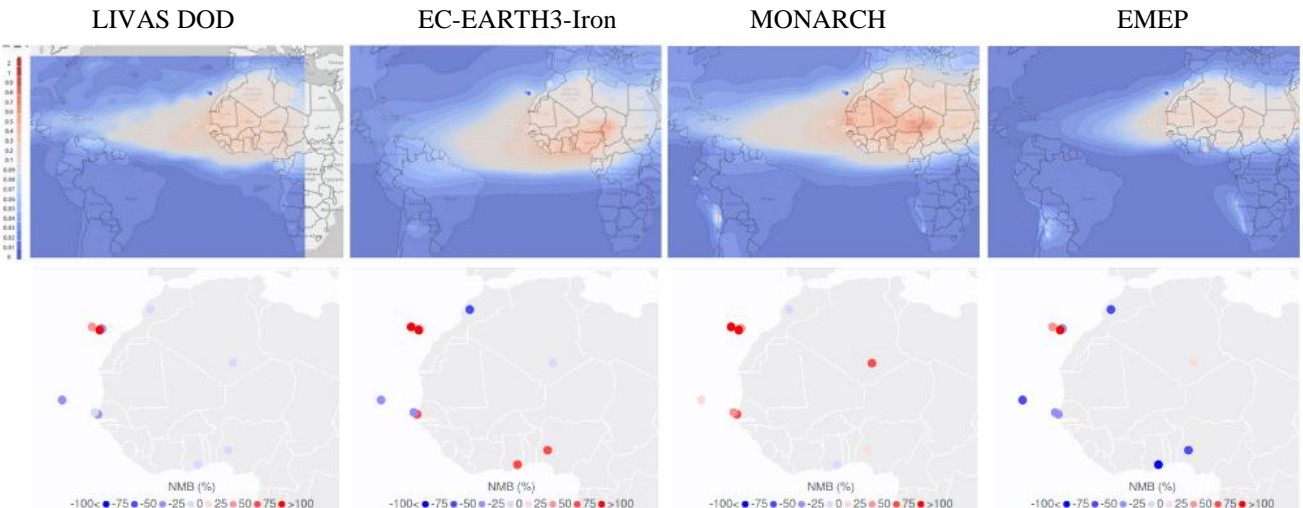

Figure 8: Yearly mean DOD (upper panels) and Normalized Mean Bias with respect to AERONET coarse-mode AOD (lower
panels) for LIVAS, EC-Earth3, MONARCH, and EMEP for the year 2020. On the upper maps, AERONET data is also
presented by the circles.

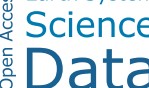

The maps of yearly mean DOD from LIVAS product and simulated by EC-Earth3, EMEP, and MONARCH model presented in Figure 8 (upper panels), show similar patterns of spatial distribution, representing the emissions of dust in the African deserts and its westward transport over the Atlantic Ocean. EC-Earth3 simulates the largest values of DOD, followed by MONARCH,

LIVAS, and EMEP. The lower panels in Figure 8 show yearly Normalized Mean Biases of LIVAS and modelled DOD with respect to AERONET coarse-mode AOD. The observations with acceptable time coverage (over 50%) within LIVAS estimated area were available at twelve AERONET sites: six continental and six sites west of the African coast (i.e. on Tenerife, Cape Verde, and La Palma). LIVAS DOD is around 20-30% lower compared to AERONET coarse-mode AOD at the continental sites and in Cape Verde while it is 47% higher on La Palma. For the sites on Tenerife, we see both under- and

over- estimation by LIVAS. LIVAS DOD is higher than AERONET coarse-mode AOD at the mountain sites Izaña (2410 m above sea level) by 44% and at Teide (3555m asl) by a factor of 5.5, where the greatest overestimation is for in October through December months (the only available observations at Teide). On the other hand, LIVAS underestimates by around 45% observations at La Laguna and Santa Cruz de Tenerife (observations only available for January-June and December), with the largest bias in June-July.

The largest difference among the models is that MONARCH, and especially EC-Earth3, simulate larger dust emissions from the Sahel desert compared to EMEP. DOD from MONARCH is the closest to AERONET coarse-mode AOD south of Sahel; EC-Earth3 overestimates it by 60-70%, while EMEP underestimates by 70-80%. The agreement is better between EMEP DOD and AERONET at one site in Sahara, whereas EC-Earth3 underestimates and MONARCH overestimates AERONET coarse-mode AOD at this site, respectively. MONARCH overestimates AERONET coarse-mode AOD at the African coastal sites

and the islands. At the low-altitude sites on Tenerife and Cape Verde, DOD from LIVAS and EC-Earth3 are closer to AERONET, while EMEP underestimates and MONARCH overestimates those. LIVAS and all three ESMs are biased rather high for all high-elevations AERONET sites (i.e., Tenerife and La Palma). The monthly variation of DOD and the scatterplot of LIVAS and models vs AERONET coarse-mode AOD at all considered sites are shown in Figure 8 (right and middle panels). However, it should be noted that LIVAS DOD at 523 nm and ESM dust outputs include both the fine-mode and coarse-mode

dust components while the AERONET AOD at 550 nm component constitutes of the coarse-mode fraction of dust, possible resulting to increasing comparison uncertainties.

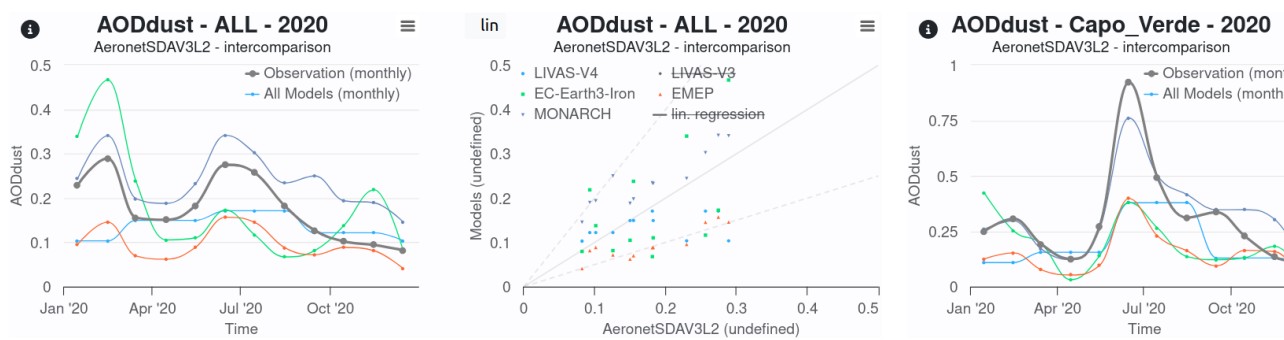



Figure 9: Monthly series (left) and scatterplot (middle) comparing DOD from LIVAS (seasonal mean) and simulated by the models (EC-EARTH3-Iron, MONARCH, and EMEP) with AERONET coarse-mode AOD at the sites shown in Fig. 8; and

the monthly series for Cape Verde (right) for the year 2020.

In general, LIVAS and ESMs describe quite well the observed monthly cycle of AERONET coarse-mode AOD observed. LIVAS underestimates coarse-mode AOD for January-February, and also June-July. EC-Earth3 overestimates coarse-mode AOD for January to March and for November, while underestimates for April through September. MONARCH shows an

overestimation of AERONET (larger in the autumn), whereas EMEP, showing a general underestimation, corresponds AERONET observations relatively well for the autumn months. In Figure 9 (right panel), we also present the monthly time series of calculated DOD and coarse-mode AOD at Cape Verde. This site can be considered representative of the dust plume transported westward off the African deserts over the subtropical eastern North Atlantic. The best agreement with AERONET coarse-mode AOD is seen for DOD from MONARCH. LIVAS DOD is quite like that from EMEP and EC-Earth3, and is

lower compared to AERONET coarse-mode AOD, especially in the summer months. The overall evaluation statistics (bias and spatial correlation) for LIVAS and modelled DOD with respect to AERONET coarse-mode AOD are summarized in Fig 10.

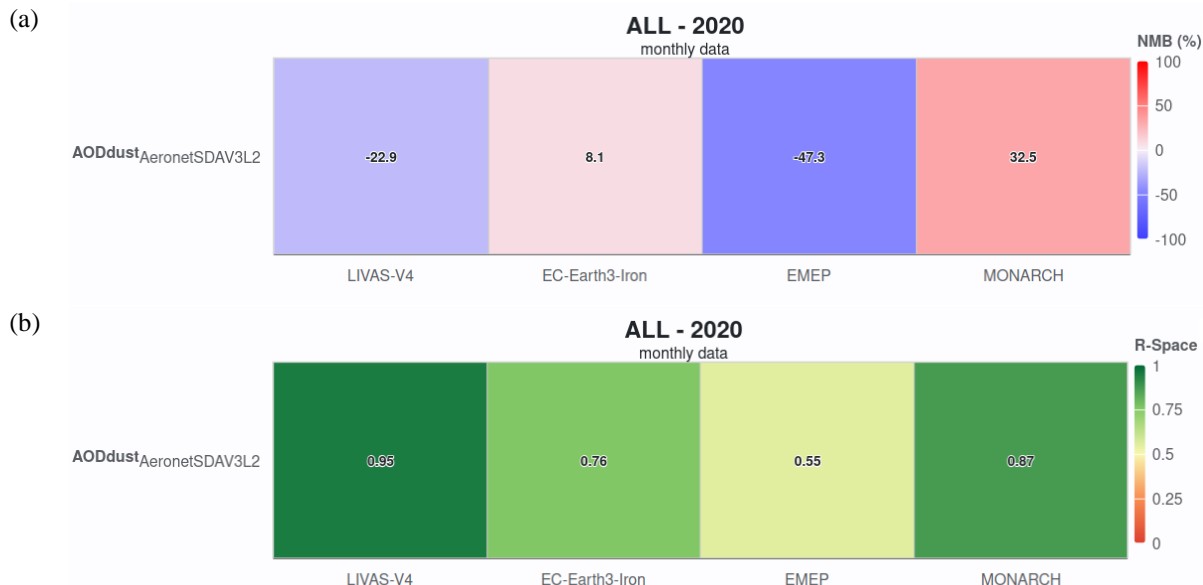

Figure 10: Overall evaluation statistics in terms of relative bias (NMB%; Fig.10a) and spatial correlation (R-Space; Fig.10b)

for the LIVAS DOD at 532 nm and the modelled DOD with respect to AERONET coarse-mode AOD (SDAVL2).

Table 6. Overall evaluation statistics in terms of normalized bias and spatial correlation for the LIVAS DOD at 532 nm and the modelled DOD with respect to AERONET coarse-mode AOD (SDAVL2).

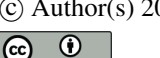



|  | LIVAS DOD | EC-Earth3-Iron | MONARCH | EMEP |
|---|---|---|---|---|
| NMB (%) | -22.9 | 8.1 | 32.5 | -47.3 |
| R-Space | 0.95 | 0.76 | 0.87 | 0.55 |

In further comparison (referred to as LIVAS_as_obs), we assess models' results against the ESA-LIVAS DOD at 532nm data. The comparison is done on the monthly basis at 2°x2° grid cells with available ESA-LIVAS DOD at 532 nm data. Figure 11 presents the maps of normalized mean biases with respect to LIVAS for EC-Earth3, MONARCH, and EMEP. Over the African deserts, MONARCH (which includes dust particles up to 20 μm in diameter) overestimates LIVAS DOD, while EC-Earth3 and EMEP have relatively smaller biases (especially EC-Earth3), i.e. positive in eastern-southeastern parts of Sahara and

slightly negative in central/western parts. MONARCH also simulates DOD higher than LIVAS over the subtropical North Atlantic and in the east of South Atlantic Ocean. EC-Earth3 has quite small (positive and negative respectively) biases compared to LIVAS DOD in the grid cells over North Atlantic Ocean, while overestimating over its equatorial part and South Atlantic Ocean. EMEP results are quite close to LIVAS DOD over the Atlantic north of equator, but underestimates over the South Atlantic Ocean. It should be noted that due to a rather simplified description of AOD in the EMEP model, the

uncertainties in DOD modelling are associated with both dust three-dimensional concentrations and assumptions for dust specific extinction.

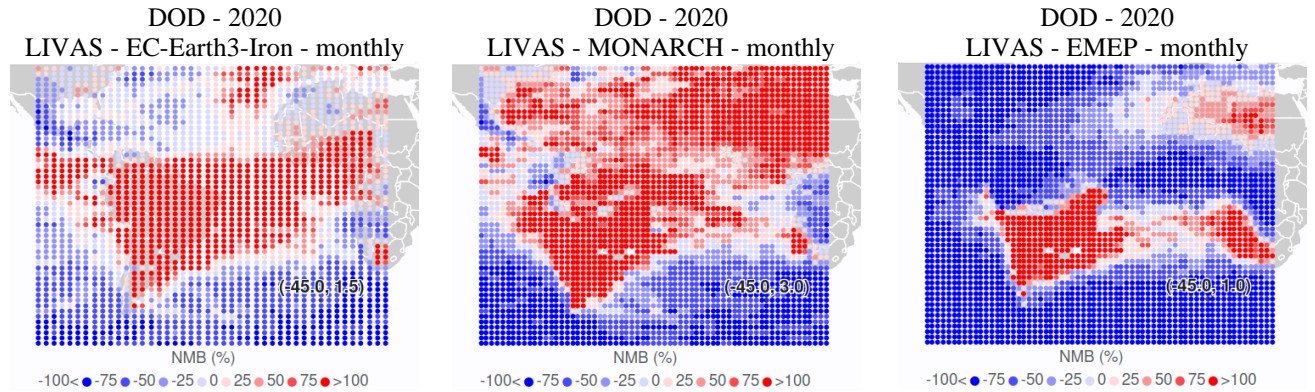

Figure 11: Normalized mean bias of modelled DOD compared with LIVAS data for EC-Earth3 (left), MONARCH (middle), and EMEP (right).


The maps of yearly mean dust total (dry and wet) deposition rate established on the basis of LIVAS and simulated with the EC-Earth3-Iron, EMEP, and MONARCH models are shown in Figure 12. The general pattern of dust deposition over the Atlantic Ocean is quite similar, though the models simulated dust deposition rates smaller compared to the EO-based dust deposition rate estimates, with MONARCH results being closer to those and EC-Earth3-Iron providing the lowest dust

deposition rates in terms of magnitude. Figure 13 presents the maps of relative biases for dust deposition rate from the three ESMs with respect to the estimates of the present study. Along the dust plume core, EC-Earth3-Iron has the largest negative



bias (mostly around -65 to -80%), EMEP mostly underestimates (with bias varying mostly between 30 and 60%), while MONARCH has the lowest bias (an order of 15-40%). Zonally, the models' biases remain quite similar across the Atlantic Ocean almost as far as the Caribbean Sea, indicating similar east-to-west dust deposition gradients. However, over the

Caribbean Sea, the models' underestimation of LIVAS dust deposition increases. To the north and south of the main dust plume, the models' results show smaller negative and positive biases with respect to LIVAS dust deposition, respectively.

(a)          (b)          (c)          (d)

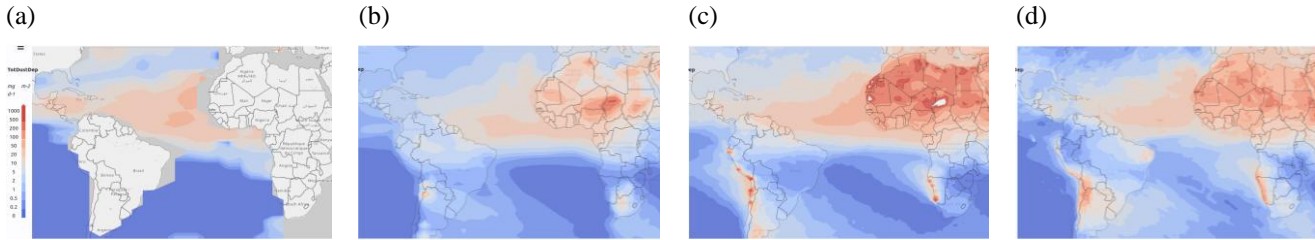

Figure 12: Yearly mean dust deposition rate (mg m$^{-2}$ d$^{-1}$) from LIVAS (a) and simulated with EC-Earth3-Iron (b), MONARCH (c), and EMEP (d). Year: 2020.


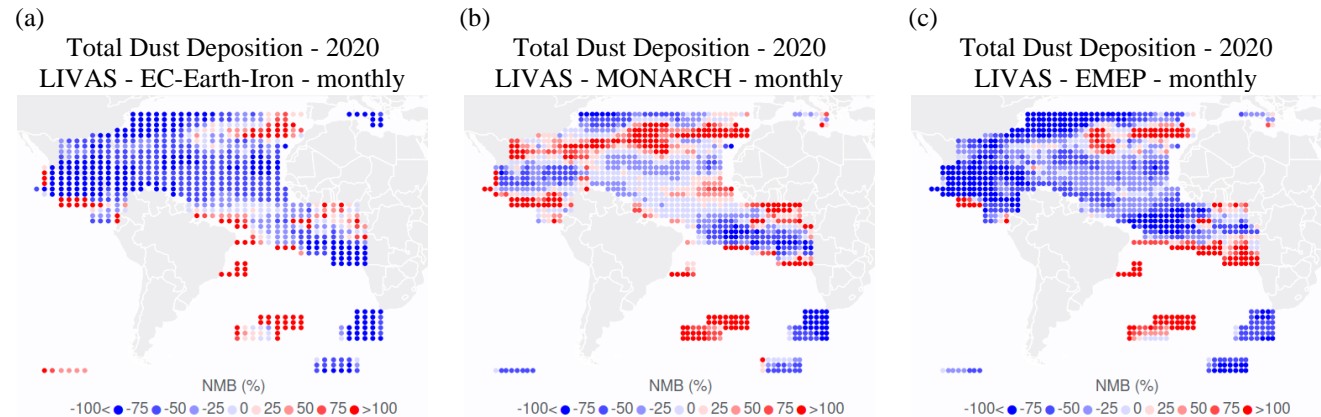

Figure 13: Yearly mean bias (mg m$^{-2}$ d$^{-1}$) for dust total deposition for EC-Earth3_Iron (a), MONARCH (b), and EMEP (c) with respect to LIVAS data. Year: 2020. Here, the empty grid cells are those without LIVAS data (corresponding to zeroes in Fig. 12).

The timeseries in Figure 14 compare profiles of DOD (a) and deposition rate (b), simulated by the three models and those produced by LIVAS. The profiles are the averages over all of 2°x2° grid cells with LIVAS data (Fig. 13). The EMEP model shows better agreement with LIVAS DOD, slightly underestimating it. EC-Earth3-Iron overestimates LIVAS DOD for January to March, showing quite good correspondence otherwise. MONARCH simulated DOD higher than LIVAS through the year. For dust deposition, no one of the models manage to reproduce LIVAS summer maximum, with EC-Earth3-Iron

underestimating LIVAS the most and simulating winter/autumn maxima instead. The reported higher dust deposition by

LIVAS compared to ESMs more probably relates to the of the historic "Godzilla" dust intrusion over the Atlantic Ocean in June 2020, with atmospheric dust load substantially underestimated by dust models (Yu et al., 2021), with dust removal processes more efficiently removed from the atmosphere (Yu et al., 2019; Kim et al., 2014). For the other seasons, MONARCH overestimates LIVAS dust deposition, while EMEP and Earth3-Iron dust deposition rates are quite close to LIVAS.


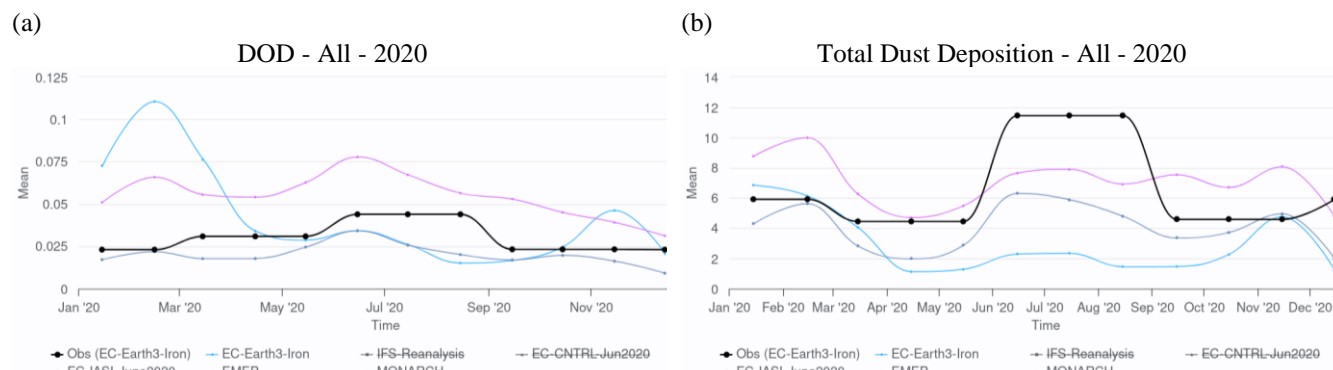

Figure 14: Monthly series of DOD and dust total deposition from LIVAS (Obs, black line), EC-Earth3, MONARCH, and EMEP (models' results are collocated with LIVAS data at grid cells of 2°x2°).

The overall evaluation statistics (bias and spatial correlation) for model simulated DOD and deposition rate with respect to

LIVAS data are summarized in the heatmaps (Fig. 15). Those are yearly mean statistics, averaged over all grid cells of 2°x2° where LIVAS data available (i.e. coloured cells in Fig. 13). The spatial correlation between the models and LIVAS is quite high (between 0.7 and 0.94). The EMEP model underestimates both DOD and deposition by about 35%, which partly could be due to somewhat lower African dust emissions compared with LIVAS (as seen from comparison with AERONET coarse-mode AOD (Figs. 8-9). EC-Earth3_Iron overestimates by 42% LIVAS DOD, but underestimates by 55% LIVAS dust

deposition. The largest positive bias with respect to LIVAS data is seen for DOD from MONARCH (81%), while its dust deposition is quite close to that from LIVAS.

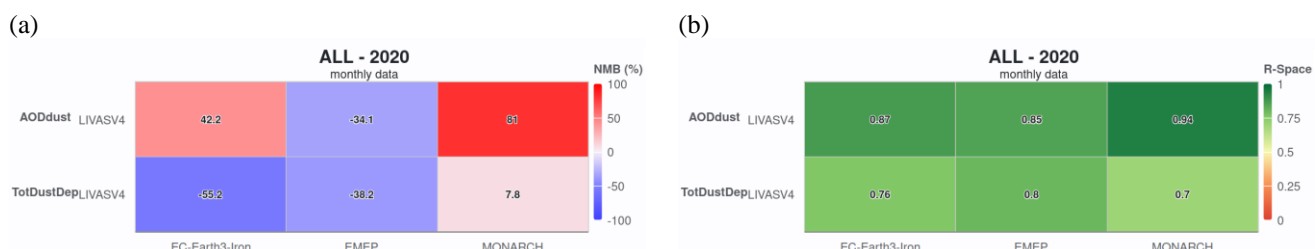

Figure 15: Summary statistics (relative bias, NMB% - Fig15.a, and spatial correlation, R-Space - Fig15.b) of models' comparison with LIVAS DOD and total deposition.





## 6 Total Dust Deposited into the broader Atlantic Ocean

Figure 16 provides a quantification of the total annual (Fig. 16a) and the seasonal-mean (Fig. 16b) dust deposition (Tg, 1 Tg = $10^{12}$ g) into the broader Atlantic Ocean (60°S-40°N, 100°W-20°E) per year. On the basis of sixteen full years of CALIPSO observations (12/2006-11/2022) it is estimated that on a basin scale the yearly-average dust deposition accounts to 274.79 ± 31.64 Tg (Fig. 16-left panel). With respect to the intrannual-seasonal variability (Fig. 16-right panel), dust deposition into the ocean is generally highest in summer and lowest in autumn, estimated to 93.10 ± 11.65 Tg and 40.62 ± 11.44 Tg for JJA and SON respectively. In spring and winter seasons, intermediate activity of dust deposition into the ocean is observed, estimated to 77.3 ± 19.93 Tg and 63.78 ± 12.03 Tg for MAM and DJF respectively. The apparent seasonal variability, subject to the high variability of dust emission, transport, and removal processes, propagates into the observed inter-annual fluctuations of the amount of dust deposited into the Atlantic Ocean, resulting to high year-to-year heterogeneity. Moreover, a negative statistically significant trend in dust deposition at the significance level of 0.05 is observed, characterized by slope -13.35 Tg yr$^{-1}$ and offset 306.97 Tg. In addition, with respect to the year-to-year variability, it is estimated that the total dust deposition on basin scale ranges between as low as ~221.1 Tg (2019) to as high as ~324.6 Tg (2008). As indicator to the interannual and seasonal-intrannual variability of dust deposition the normalized standard deviation (NSD) is provided, calculated as the ratio of the standard deviation of the seasonal dust deposition of each year to the mean seasonal dust deposition of each year and as the ratio of the standard deviation of the seasonal dust deposition for all seasons to the mean seasonal dust deposition for all seasons over the 12/2006-11/2022 period, respectively. The larger the NSD, the greater the variability of dust deposition. With respect to the year-to-year amount of dust deposited into the Atlantic Ocean basin, NSD shows mean intrannual variability about ~36.03%. With respect to the seasonal amount of dust deposited on basin scale, NSD shows largest and lowest variability in spring (MAM) and summer (JJA) season, of ~25.78% and ~12.52% respectively. In autumn and winter seasons, dust deposition into the ocean is slightly lower than in spring, estimated to ~28.16% and ~18.87% for SON and DJF respectively.

Table 7: Annual-mean and seasonal-mean dust deposition (Tg, 1 Tg = $10^{12}$ g) and Normalized Standard Deviation (%) into the broader Atlantic Ocean (60°S-40°N, 100°W-20°E) on the basis of sixteen full years of CALIPSO observations (12/2006-11/2022).

|  | Annual | DJF | MAM | JJA | SON |
|---|---|---|---|---|---|
| Dust Deposition (Tg) | 274.79 ± 31.64 | 63.78 ± 12.03 | 77.3 ± 19.93 | 93.10 ± 11.65 | 40.62 ± 11.44 |
| NSD (%) | ~36.03% | ~18.87% | ~25.78% | ~12.52% | ~28.16% |

(a)                                                                                          (b)



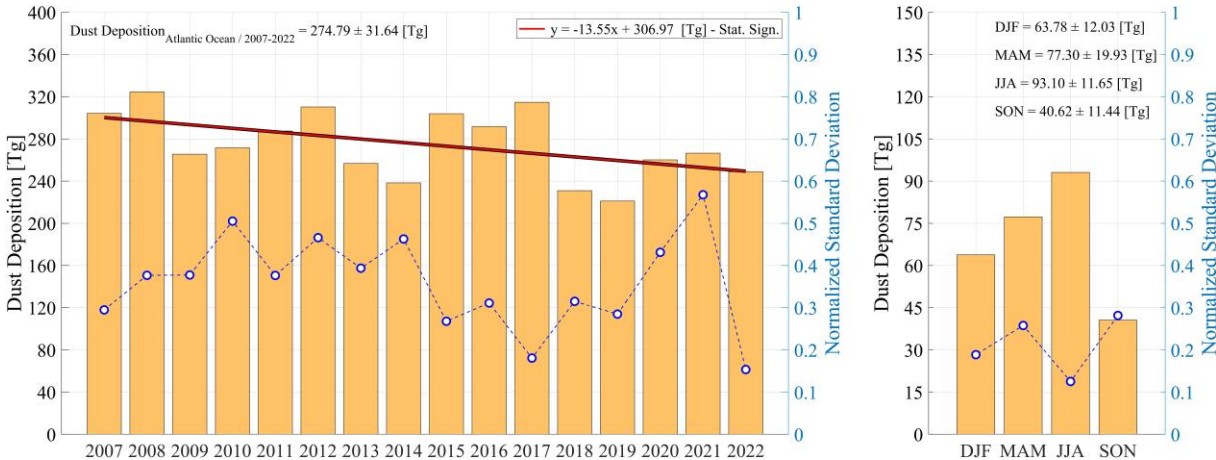

Figure 16: Quantification of the (a) total annual-mean and (b) seasonal-mean dust deposition (Tg yr$^{-1}$, 1 Tg = $10^{12}$ g) into the broader Atlantic Ocean (60°S-40°N, 100°W-20°E) on the basis of 16-years of CALIPSO observations (12/2006-11/2022).

Despite the vital role of atmospheric deposited lithogenic material for ocean biogeochemistry and for understanding environmental impact of the atmospheric dust cycle, the availability of direct mineral dust deposition measurements is limited. Thus, in order to connect these sparse in-situ observations and gain insight into the highly heterogeneous both in time and space influx of dust particles into the broader ocean several efforts have been undertaken (Table 8), primarily relying on numerical model simulations (i.e., Duce et al., 1991; Prospero et al., 1996; Ginoux et al., 2001; Zender et al., 2003; Luo et al.,

2003; Ginoux et al., 2004; Tegen et al., 2004; Jickells et al., 2005; Johnson et al., 2010; Kim et al., 2014; Kok et al., 2023) but also following the unprecedented increase in the range, quality, and frequency of satellite-based observations on remote sensing of dust (i.e., Kaufman et al., 2005; Foltz, 2014; Yu et al., 2015; 2019). Figure 17 compares the satellite-based estimates of dust deposition of the present study (12/2006-11/2022 average) across the broader Atlantic Ocean (AO), North Atlantic Ocean (NAO), and South Atlantic Ocean (SAO) with documented estimates of dust deposition. This approach serves more as cross-

assessment rather than a rigorous evaluation since the majority of the reported estimates rely on highly variable approaches, assumptions, parametrizations, considered meteorology, temporal periods, and domains.

As examples on the variability of the methods used, the numerical model simulations of Ginoux et al. (2001) on the sources and distributions of dust aerosols, reporting dust deposition rate of ~184 Tg yr$^{-1}$, ~20 Tg yr$^{-1}$, and ~204 Tg yr$^{-1}$ across the North Atlantic Ocean, South Atlantic Ocean, and the broader Atlantic Ocean, respectively, averaged over a five-year period (1987-

1990 and 1996), were based on the Georgia Global Ozone Chemistry Aerosol Radiation and Transport (GOCART) model, implementing input assimilated meteorology fields from the Goddard Earth Observing System Data Assimilation System (GEOS DAS; Schubert et al., 1993) and a dust size distribution of seven modes between 0.1 and 6 μm (radius) and dust density of 2.5 g m$^{-3}$ (Chin et al., 2000). The 1990s climatology of Zender et al. (2003), based on the Dust Entrainment and Deposition (DEAD) model driven by the National Center for Environment Prediction (NCEP) meteorology (Kalnay, 1996), reports dust



deposition rate of ~178 Tg yr[-1], ~29 Tg yr[-1], and ~207 Tg yr[-1] across the North Atlantic Ocean, South Atlantic Ocean, and the broader Atlantic Ocean, respectively, accounting though for larger dust particles of size 0.1 to 10 μm in terms of diameter. A few years later, the dust deposition estimates of Johnson et al. (2010) were based on simulations with the global chemical transport model GEOS-Chem for the period extending between October 2006 and September 2007, distributing dust in four modes between 0.2 and up to 12.0 μm in terms of diameter (Fairlie et al., 2007). The study reported dust deposition of ~22 Tg

yr[-1], slightly lower than the dust deposition estimates provided by Gaiero et al. (2003) of ~30 Tg yr[-1], focusing both on the South Atlantic Ocean domain and more specifically on the Patagonian dust source regions. The corresponding total South Atlantic Ocean total dust deposition, upon accounting in addition to South America arid areas (McConnell et al., 2007; Mazzonia and Vazquez, 2009; Ginoux et al., 2012) the major dust sources of South Africa (Eckardt and Kuring, 2005; Bryant et al., 2007; Ginoux et al., 2012; Vickery et al., 2013; Gkikas et al., 2022), was estimated significantly higher to ~86 Tg yr[-1]

by Kok et al. (2023). Kaufman et al. (2005), on the basis not of model simulations but on EOs of dust provided by Terra - Moderate Resolution Imaging Spectroradiometer (MODIS; Remer et al., 2005) observations in 2001 estimated dust emitted from Africa and deposited to the Atlantic Ocean as far as the Caribbean approximately ~190 Tg yr[-1] for the year 2001. Yu et al. (2019) implemented CALIPSO-CALIOP (Winker et al., 2010) dust profiles on synergy with DOD products established on the basis of observations provided by MODIS (Remer et al., 2005), MISR (Garay et al., 2020), and IASI (Capelle et al., 2014)

radiometers, over the Tropical Atlantic Ocean domain. On the basis of a ten-year period (2007-2016) and basin-scale average (5°S-35°N), Yu et al. (2010) reported ~151.6 Tg yr[-1] on the basis of CALIOP, and ~221.5 Tg yr[-1] on the basis of CALIOP-MODIS, ~168 Tg yr[-1] on the basis of CALIOP-MISR, and ~136 Tg yr[-1] on the basis of CALIOP-IASI synergies.

Table 8: Documented numerical model simulations and satellite-based estimates of dust deposition across the broader Atlantic
Ocean (AO), North Atlantic Ocean (NAO), and South Atlantic Ocean (SAO).

| Reference | Domain | Deposition (Tg yr[-1]) | |
| --- | --- | --- | --- |
| | | Partial Deposition | Total Deposition |
| Duce et al. (1991) | NAO | 220 | 244 |
| | SAO | 24 | |
| Prospero et al. (1996) | NAO | 220 | 225 |
| | SAO | 5 | |
| Ginoux et al. (2001) | NAO | 184 | 204 |
| | SAO | 20 | |
| Zender et al. (2003) | NAO | 178 | 207 |
| | SAO | 29 | |
| Gaiero et al. (2003) | SAO - Patagonian | 30 | |
| Luo et al. (2003) | NAO | 230 | 260 |
| | SAO | 30 | |
| Ginoux et al. (2004) | NAO | 161 | 181 |
| | SAO | 20 | |
| Tegen et al. (2004) | NAO | 259 | 294 |
| | SAO | 35 | |
| Kaufman et al. (2005) | 20°S - 30°N | 190 | |





| Jickells et al. (2005) | NAO | 193.5 | | 211.5 |
|---|---|---|---|---|
| | SAO | 18 | | |
| Johnson et al. (2010) | SAO - Patagonian | 22 | | |
| Mahowald et al. (2010) | NAO | 276 | | 311.8 |
| | SAO | 35.8 | | |
| Foltz (2014) | (0°-25°N) | 224 | | |
| Kim et al. (2014) | (90°W–17°W, 0°N–35°N) | GOCART: | 349 | |
| | | GISS: | 196 | |
| | | SPRINTARS: | 105 | |
| | | ECHAM5: | 158 | |
| | | HadGEM2: | 70 | |
| Yu et al. (2015) | 10°S - 30°N | 154 | | |
| Yu et al. (2019) | 5°S - 35°N | CALIOP: | 151.6 | |
| | | CALIOP-MODIS: | 221.5 | |
| | | CALIOP-MISR: | 168 | |
| | | CALIOP-IASI: | 136 | |
| Kok et al. (2023) | NAO | 230 | | 316 |
| | SAO | 86 | | |
| The present study | NAO | 243.98 ± 23.89 | | 274.79 ± 31.64 |
| | SAO | 30.81 ± 10.49 | | |

Figure 17 shows the comparison between the documented dust deposition estimates (Table 8) and our estimates of dust deposition across the broader Atlantic Ocean (Fig. 17a), North Atlantic Ocean (Fig. 17b), and South Atlantic Ocean (Fig. 17c). Overall, on the basis of CALIOP observations between 12/2006 and 11/2022 the annual-mean deposited dust into the broader

Atlantic Ocean is estimated 274.79 ± 31.64 Tg yr$^{-1}$, of which 243.98 ± 23.89 Tg yr$^{-1}$ of dust is deposited across the North Atlantic Ocean and 30.81 ± 10.49 Tg yr$^{-1}$ of dust is deposited across the South Atlantic Ocean. These satellite-derived estimates of dust deposition lie within the much larger documented range of dust deposition rates, varying by a factor of two and ranging from 181 Tg yr$^{-1}$ (Ginoux et al., 2004) to 316 Tg yr$^{-1}$ (Kok et al., 2023) for the case of the broader Atlantic Ocean, by a factor of five ranging and from 70 to 349 Tg yr$^{-1}$ for the case of the North Atlantic Ocean (Kim et al., 2014), and by a factor of

seventeen ranging and from 5 Tg yr$^{-1}$ (Prospero et al., 1996) to 86 Tg yr$^{-1}$ for the case of the South Atlantic Ocean (Kok et al., 2023). More specifically, on a basis of average of all documented dust deposition estimates (Table 8), the various approaches yield annual dust deposition of 245.43 ± 48.16 Tg yr$^{-1}$ into the broader Atlantic Ocean, 194.30 ± 59.39 Tg yr$^{-1}$ into the North Atlantic Ocean or Tropical Atlantic Ocean, and 29.57 ± 19.68 Tg yr$^{-1}$ into the South Atlantic Ocean. The apparent larger quantified dust deposition estimates of our study with respect to the average of all documented dust deposition estimates, of

29.36 Tg yr$^{-1}$ (11.96%) across the broader Atlantic Ocean, of 49.68 Tg yr$^{-1}$ (25.57%) across the North Atlantic Ocean, and of 1.24 Tg yr$^{-1}$ (4.21%) across the South Atlantic Ocean. However, the dust deposition estimates of our study fall well within the variability of the reported dust deposition outcomes, within one standard deviation. These satellite-derived estimates of dust deposition are rather promising, given that the documented quantifications of dust deposition used as cross-evaluation were performed over different time spans and spatial scales, the significant variability in model representations of emission and

transport processes which are highly heterogeneous in both space and time, the parameterizations of the vertical structure of





dust in the atmosphere and of dry and wet dust deposition, the substantial disparity in the size range, distribution, and density of dust in model simulations, and the different utilized satellite-based sensors and applied techniques.

(a)

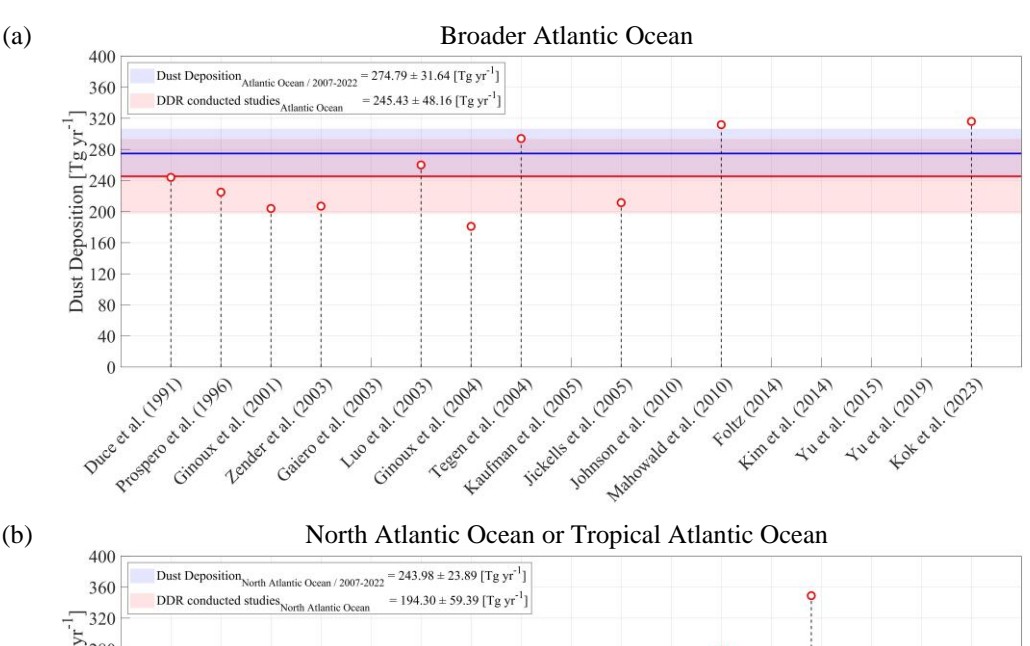

(b)

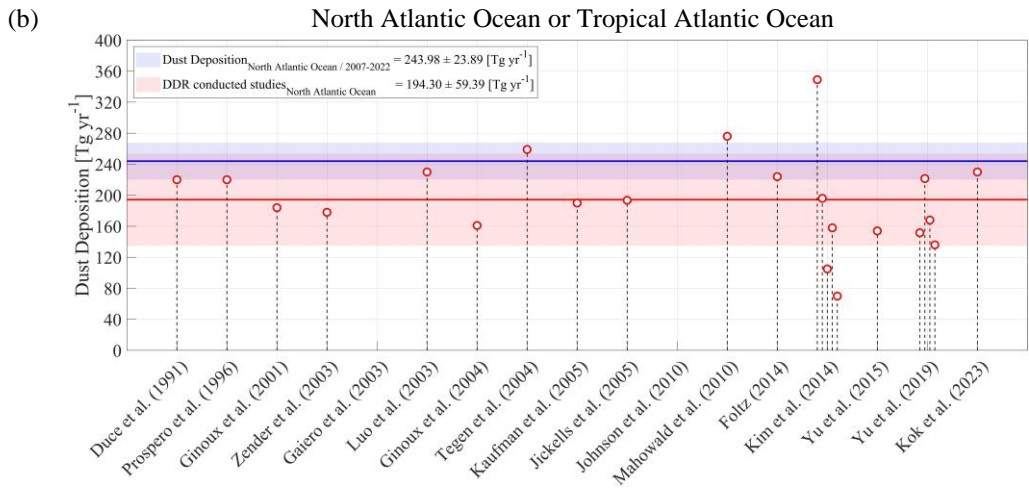

(c)                                        South Atlantic Ocean



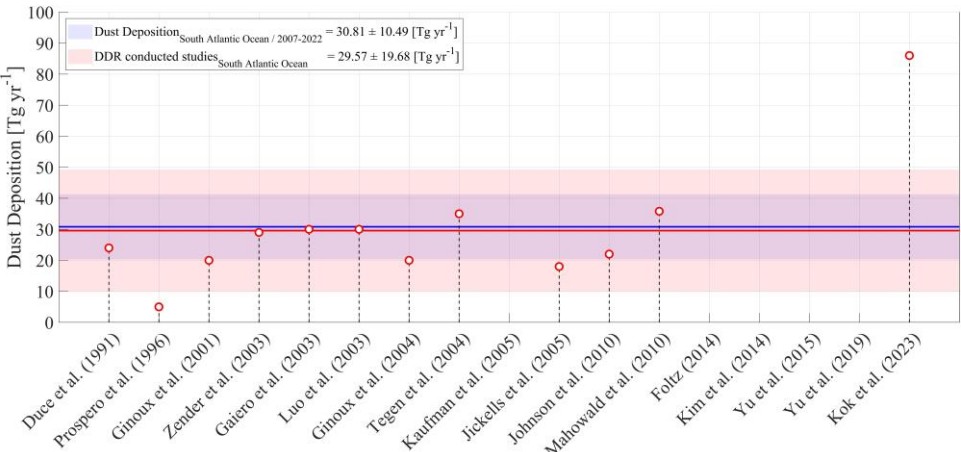

Figure 17: Comparison between the present satellite-based estimates of dust deposition (12/2006-11/2022 average) across (a) the broader Atlantic Ocean (AO), (b) North Atlantic Ocean (NAO), and (c) South Atlantic Ocean (SAO) with documented estimates of dust deposition.

## 7 Summary and conclusions

Deposition of atmospheric dust across the surface of the open ocean fertilizes marine ecosystems with essential nutrients, such as nitrogen (N), phosphorus (P), silica ($SiO_2$), and iron (Fe), critical for regulating phytoplankton growth, and consequently modulating marine productivity, ocean colour, the global carbon cycle through the ocean's capacity to absorb $CO_2$, weather and eventually climate. However, despite the vital role of atmospheric deposited lithogenic material for the ocean biogeochemistry and for understanding physical processes linked to the dust cycle, the amount of atmospheric dust that is actually deposited into the open ocean is still not well quantified. Numerous significant challenges inherent to the complex nature of oceanographic research hamper the feasibility of establishing long-term and continuous in-situ measurements of high spatial coverage over extensive geographical areas and temporal periods while numerical model simulations depend on highly parameterized representations of dust processes and on highly variable approaches and assumptions, including among others consideration on meteorology, the dust particle size distribution, and temporal periods and domains, with few constraints.

This study provides a novel satellite-based climate data record (CDR) of dust deposition rate ($mg/m^2 d$) estimates for the broader area encompassing the dust emission sources of West Africa and South America with focus on the Atlantic Ocean, the Caribbean Sea, and the Gulf of Mexico regions, confined between latitudes 60°S to 40°N and of 5° (zonal) x 2° (meridional) spatial resolution, in a seasonal-mean temporal resolution, and of sixteen-year temporal coverage, extending between 12/2006 and 11/2022. The product of dust deposition rate estimates into the ocean is established on the basis of atmospheric dust fluxes over the ocean and relies on two central enabling components. First, on the 4D structure of dust aerosols in the atmosphere in terms of mass concentration ($\mu g/m^3$), provided by the ESA-LIVAS CDR, established through the one-step POLIPHON technique applied to CALIPSO-CALIOP aerosol observations at 532 nm. Second, on the eastward and northward components



of wind (m/s), provided at different pressure levels by the ECMWF reanalysis dataset ERA5. Synergistic implementation of the two datasets provides the capacity to estimate dust mass fluxes ($\mu$g/m²s) into the ocean by means of differentiation of the zonal and meridional dust input/output atmospheric dust mass fluxes through the areas of conceptual cuboid atmospheric columns lying over the ocean surface and on the basis of the mass conservation hypothesis.

Towards verifying the accuracy, ensuring the reliability, and quantifying the uncertainties of the satellite-based estimations of dust deposition rate, includes implementation of sediment-trap observations of dust deposition fluxes as reference datasets. Despite the several sources of uncertainties and the several methodological factors driving the observed discrepancies, the satellite-based dust deposition rate product and the sediment-trap observations, a rather good agreement between the two datasets (slope of 0.85, intercept of 9.49 mg/m²/day, and Pearson correlation coefficient 0.79) is revealed, though the general

trend of satellite-based dust deposition is to overestimate those from the in situ provided observations (mean bias of 5.42 mg/m²/day, relative bias of 19.82%, and RMSE of 30.3 mg/m²/day). In addition, the EO-based atmospheric dust product of the ESA-LIVAS CDR in terms of DOD at 532 nm is evaluated against AERONET SDA coarse-mode AOD, as a significant intermediate step since the capacity of the LIVAS to accurately provide the spatiotemporal variability of the atmospheric dust conditions is crucial towards quantification of the dust deposited component across the dust transport over the ocean. The

evaluation activities performed in the framework of AeroVal between the two datasets reveals the high quality of the ESA-LIVAS atmospheric dust product, characterized by Normalized Mean Bias (NMB) of 0.4% and Pearson's correlation coefficient of 0.77. Accordingly, AeroVal intercomparison of the EO-based dust deposition product against the atmospheric dust outputs in terms of optical depth and the corresponding dust deposition fields provided by MONARCH, EMEP MSC-W, EC-Earth3-Iron ESMs for the year 2020 is performed. The intercomparison reported high agreement between the ESA-LIVAS atmospheric dust product and EC-Earth3-Iron (NMB of 9.2%, Pearson's correlation coefficient of 0.87), EMEP (NMB of -

50.2%, Pearson's correlation coefficient of 0.85), and MONARCH (NMB of 38.1%, Pearson's correlation coefficient of 0.93), corroborating the high-quality of the ESA-LIVAS atmospheric dust product. In terms of the AeroVal performed cross-comparison between the satellite-based estimates of dust deposition rate and the ESM dust deposition outputs relatively good agreement between the satellite-based estimates of dust deposition and EC-Earth3-Iron (NMB of -68.9%, Pearson's correlation coefficient of 0.69), EMEP (NMB of -57.1%, Pearson's correlation coefficient of 0.75), and MONARCH (NMB of -25%,

Pearson's correlation coefficient of 0.66) is revealed, with the satellite-based estimates within a factor of 2 from the corresponding ESMs dust deposition fields. Overall, it is notable that the satellite-based dust deposition rate product consistently reproduces the dust deposition patterns recorded by the sediment traps installed and operated across the broader Atlantic Ocean, with a positional accuracy and magnitude generally within a factor of 2 compared to sediment trap

measurements, and in addition the spatiotemporal characteristics of ESMs dust deposition as demonstrated through the AeroVal intercomparison. The performed evaluation reveals the capacity of the satellite-based product to quantitatively provide the amount of dust deposited into the broader Atlantic Ocean, consistent with the seasonal activation of the dust source regions, revealing the seasonal four-dimensional migration of dust transport pathways, and in good agreement with features reported by sediment-traps' in-situ measurements of lithogenic material and ESMs.





The annual-mean amount of dust deposition into the broader Atlantic Ocean is quantified at 274.79 ± 31.64 Tg, of which 243.98 ± 23.89 Tg of dust is deposited into the North Atlantic Ocean and 30.81 ± 10.49 Tg of dust is deposited into the South Atlantic Ocean, each year. With respect to the intrannual-seasonal variability, dust deposition into the ocean is generally highest in summer and lowest in autumn, estimated at 93.10 ± 11.65 Tg and 40.62 ± 11.44 Tg, for JJA and SON respectively. In spring and winter seasons, intermediate activity of dust deposition into the ocean is observed, estimated to 77.3 ± 19.93 Tg

and 63.78 ± 12.03 Tg, respectively. Furthermore, comparison of the satellite-based estimates of dust deposition (12/2006-11/2022 average) with the average of documented model-based and satellite-based estimates of dust deposition over the broader Atlantic Ocean (245.43 ± 48.16 Tg yr$^{-1}$), North Atlantic Ocean (194.30 ± 59.39 Tg yr$^{-1}$), and South Atlantic Ocean (29.57 ± 19.68 Tg yr$^{-1}$) is performed. The intercomparison reveals larger dust deposition estimates of our satellite-based dust deposition product with respect to the average of all documented dust deposition estimates, of 29.36 Tg yr$^{-1}$ (11.96%) across

the broader Atlantic Ocean, of 49.68 Tg yr$^{-1}$ (25.57%) across the North Atlantic Ocean, and of 1.24 Tg yr$^{-1}$ (4.21%) across the South Atlantic Ocean, falling though well within each-others variability, within one standard deviation. It should be noted that the performed intercomparisons serves more as cross-assessment of the satellite-based dust deposition estimates rather than as a rigorous evaluation since the majority of the reported estimates rely on highly variable approaches, sensors, models, assumptions, parametrizations, meteorology, temporal periods, and domains.

The satellite-based dust deposition climate data record is considered unique with respect to a diverse range of potential applications. These include filling spatial and temporal gaps in sediment-trap observational datasets thus extending their limited coverage, evaluating model simulations, and elucidating physical processes involved in the dust cycle from emission to transport and eventually deposition. The dust deposition estimates can further be used to address major knowledge gaps in marine sciences and advance our capacity to better understand, describe and predict complex and poorly understood processes

including the impact of deposited dust nutrients on the sustainability of the Atlantic Ocean biogeochemistry, and in this way to enhance science-based effective adaptation and mitigation strategies to preserve the oceans under the ongoing climate change.

**Data availability**

The CALIPSO Level 2 data products are publicly available from the Atmospheric Science Data Center at NASA Langley

Research Center (https://earthdata.nasa.gov/eosdis/daacs/asdc, EARTHDATA; last visit: 23/11/2023). The ESA-LIVAS CDR in terms of Level 2 and Level 3 pure-dust products (i.e., backscatter coefficient at 532 nm, extinction coefficient at 532 nm, mass concentration profiles and DOD at 532 nm) is available upon personal communication with Emmanouil Proestakis (proestakis@noa.gr) and/or Vassilis Amiridis (vamoir@noa.gr). The data from AERONET can be freely obtained from https://aeronet.gsfc.nasa.gov (last visit: 23/11/2023). The satellite-based dust deposition CDR is available through the Zenodo

repository: 10.5281/zenodo.14608538.





**Author contributions**

Emmanouil Proestakis: Conceptualization, Methodology, Software, Data Curation, Formal analysis. Vassilis Amiridis: Methodology, Supervision, Conceptualization. Carlos Pérez García-Pando: Methodology, Supervision, Conceptualization. Svetlana Tsyro: Software, Formal analysis, Visualization. Jan Griesfeller: Software, Formal analysis, Visualization. Antonis Gkikas: Methodology, Formal analysis, Conceptualization. Thanasis Georgiou: Software. María Gonçalves Ageitos: Software, Methodology, Formal analysis. Elisa Bergas Masso: Software, Methodology, Formal analysis. Stelios Myriokefalitakis: Software, Methodology, Formal analysis. Enza Di Tomaso: Software, Methodology, Formal analysis. Sara Basart: Project administration. Jan-Berend Stuut: Validation, Investigation, Data Curation. Jerónimo Escribano: Software, Methodology, Formal analysis. Angela Benedetti: Project administration, Supervision, Conceptualization.

**Competing interests.**

The authors declare that they have no conflict of interest.

**Acknowledgements**

Emmanouil Proestakis acknowledges support by the AXA Research Fund for postdoctoral researchers under the project entitled "Earth Observation for Air-Quality – Dust Fine-Mode (EO4AQ-DustFM)". This research was supported by the Dust Observation and Modelling Study (DOMOS) under ESA contract number 4000135024/21/I-NB. The support of the scientific project officer Dr Simon Pinnock is gratefully acknowledged. We would like to thank the NASA CALIPSO team and NASA/LaRC/ASDC for making the CALIPSO products available, which have been used to build the LIVAS products, and ESA, who funded the LIVAS project (contract no. 4000104106/11/NL/FF/fk). We thank AERONET (https://aeronet.gsfc.nasa.gov/, last access: 7th of January 2024), and AERONET-Europe for the data collection, calibration, processing, and dissemination. We thank ECMWF for providing fifth-generation ERA5 Reanalysis data and Marenostrum5 support team and Computational Earth Sciences team of BSC for their support in maintaining and using the HPC infrastructure. We would like to thank the anonymous reviewers, who were very helpful and who provided constructive comments that improved the paper.

**Financial support**

EP has been supported by the AXA Research Fund (Earth Observation for Air-Quality - Dust Fine-Mode (EO4AQ-DustFM). CPGP, EBM, MGA, JE and EdT were funded by the ESA-DOMOS contract, and acknowledge the funding received through the AXA-Chair in Sand and Dust Storms. JBS was funded by NWO with the TRAFFIC project 822.01.008 and the ERC with



starting grant 311152: DUSTTRAFFIC. MGA acknowledges the support of the BIOTA Spanish I+D+i Grant PID2022-139362OB-I00 funded by MICIU/AEI/10.13039/501100011033 and by ERDF, EU.

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
