# Peer review of "Quantifying Dust Deposition over the Atlantic Ocean"

_Earth System Science Data, 2025_

## Referee Comment (RC2)

[referee-annotated manuscript omitted]

---

## Author Comment (AC1)

**Quantifying Dust Deposition over the Atlantic Ocean**

High IF journals deserve very high-quality papers and in this case dataset. This is definitively the case and I strongly support its publication and I suggest it as a highlight paper.

The authors would like to thank the reviewer for his time, comments and suggestions. We did our best to incorporate the proposed changes and corrections in the revised manuscript, aiming at improving the presented paper. Following, you will find our responses, one by one to the comments addressed.

Kind regards, Emmanouil Proestakis et al.

**Reviewer's Comments**

Section 2.2.1: the error on dust concentration is not discussed at all here. The methodology is widely described in previous paper, but here it should be reported together with dust concentration profile method also an estimation of the error on it. This is crucial for all the following analysis within the paper.

Following the reviewer's comment and recommendation, the following paragraph was inserted in the manuscript, in the "Section 2.2.1 - Decoupling the atmospheric pure-dust component from the total aerosol load":

"Uncertainties in the retrieval of atmospheric dust properties (i.e., backscatter coefficient, extinction coefficient, and mass concentration) originate from multiple sources, primarily (a) the uncertainties in the CALIPSO L2 optical products and aerosol classification flags, (b) the methodology applied towards atmospheric dust decoupling from the total aerosol load, and (c) the assumed constants and conversion factors (i.e., lidar ratios (LRs) and extinction-to-mass conversion parameters) (Marinou et al., 2017; Proestakis et al., 2018; 2024). CALIPSO L2 retrieval uncertainties, particularly in backscatter coefficient and particulate depolarization ratio at 532 nm, are assumed to be random and uncorrelated (Vaughan et al., 2009; Winker et al., 2009), yet can be significant; particulate depolarization ratio uncertainties often exceed 100%, and the limitations in aerosol subtype classification introduce further biases, especially for mixed or tenuous layers. The aerosol subtype classification algorithm (Omar et al., 2009; Kim et al., 2018) may result in positive or negative biases depending on feature type misclassification, with particularly low accuracy (~35%) in identifying polluted dust layers (Burton et al., 2013). Moreover, CALIOP's limited sensitivity to optically thin layers (Kacenelenbogen et al., 2011; Rogers et al., 2014) leads to systematic underestimation of AOD, with negative biases of  $\sim 0.02$  (nighttime) and < 0.1 (daytime), primarily due to its detection sensitivity (Toth et al., 2018). The application of dust decoupling methodologies (Shimizu et al., 2004; Tesche et al., 2009; 2011; Ansmann et al., 2019) introduces additional uncertainties, with the depolarization-based separation approach contributing 5%-10% uncertainty during strong dust events and up to 20%–30% in less pronounced cases, primarily linked to variability in assumed dust depolarization ratios (Tesche et al., 2009, 2011; Ansmann et al., 2012; Mamouri et al., 2013). The conversion of decoupled backscatter profiles to extinction coefficients using regional LRs introduces relative uncertainties on the order of 15%-25%, while the final conversion to dust mass concentration profiles introduces further uncertainties of approximately 10%-15% (Tesche et al., 2009; Amiridis et al., 2013; Marinou et al., 2017; Proestakis et al., 2024). Cumulatively, uncertainties propagate and increase with each processing step, resulting in total uncertainties that can reach 10%-30% in backscatter coefficient, 15%-50% in extinction coefficient, and 20%-60% in mass concentration for ground-based lidar observations (Mamouri and Ansmann, 2017; Ansmann et al., 2019). However, in the case of CALIPSO-based retrievals, where the dominant uncertainty sources are the backscatter and depolarization ratio inputs, frequently of the same order of magnitude or even higher than the optical products, total uncertainties in mass concentration retrievals can escalate to 100%-150%, underscoring the challenge of achieving high precision in satellite-based dust mass estimates (Marinou et al., 2017; Proestakis et al., 2018; 2024).".

Section2.2.2: the authors report: "As a next step, a three-dimensional (3D) closed cuboid surface is assumed, of  $5^{\circ}$  length (zonal),  $2^{\circ}$  width (meridional), and 10km height (vertical), with the base surface at 0 km a.m.s.l." Could authors explain the choice of such dimension of the cuboid?

Towards establishing a 4-D reconstruction of the atmospheric dust aerosol component, in terms of mass concentration, and accordingly quantifying the deposited component over the surface of the broader Atlantic Ocean Atlantic, different spatial and temporal resolutions were tested.

The different spatiotemporal resolutions were applied in the framework of the developments made, with the objective to establish the 4-D atmospheric dust climate data record, as representative for the climatological characteristics of the dust transport events and pathways over the Ocean as possible, reducing at the same time CALIOP L2 5km noiseeffects propagating and contaminating eventually the output datasets and information. Aiming to provide an indicative example of CALIOP characteristics and CALIPSO limitations that had to be considered driving the final choice on the spatial and temporal resolutions is provided in the following figure. More specifically, the following figure provides the number of CALIPSO L2 5km quality assured profiles per  $1^{\circ}x1^{\circ}$  deg2 grid for 06/2020 (left) and the number of CALIPSO overpasses per  $1^{\circ}x1^{\circ}$  deg2 grid for 06/2020 (right), both for the region of interest of the study. It is evident that (i) the number of compasses varies highly from grid-to-grid, extending between no overpasses (0) and eight (8), a number of observational overpasses significantly lower than the equivalent from passive sensors, (ii) frequently "neighboring"  $1x1 deg^2$  grids are observed differently, with temporal intervals at least of a few days, thus capturing in some cases different dust transport events, and (iii) even a high number of CALIPSO overpasses may not provide a significant number of CALIPSO L2 5km quality assured aerosol profiles, for instance due to cloud contamination.

Figure: (left) the number of CALIPSO L2 5km quality assured profiles per  $1^{\circ}x1^{\circ}$  deg2 grid for 06/2020, and (right) the number of CALIPSO overpasses per  $1^{\circ}x1^{\circ}$  deg2 grid for 06/2020, for the region of interest.

Taking into consideration the conclusions of the above-provided figure, it concurs that specific grid areas, characterized by a low number either of CALIPSO overpasses or of quality assured profiles, may not provide representative information of atmospheric dust transport, thus not accurate information of the deposited dust component. In the framework of the study, and following the need to balance an approach aiming to ensure as possible representativeness of the CALIPSO-based atmospheric dust products and at the same time maintain a number of grids high enough to allow for dust deposition retrievals, different spatial and temporal resolutions were tested. More specifically, the pre-processing resolution tests included the following:

- Spatial resolution: 1°x1° deg2 grids / Temporal resolution: monthly mean / 10 km upper boundary (vertical).
- Spatial resolution: 2°x2° deg2 grids / Temporal resolution: monthly mean / 10 km upper boundary (vertical).
- Spatial resolution: 5°x2° deg2 grids / Temporal resolution: monthly mean / 10 km upper boundary (vertical).
- Spatial resolution: 1°x1° deg2 grids / Temporal resolution: seasonal mean / 10 km upper boundary (vertical).
- Spatial resolution: 2°x2° deg2 grids / Temporal resolution: seasonal mean / 10 km upper boundary (vertical).
- Spatial resolution: 5°x2° deg2 grids / Temporal resolution: seasonal mean / 10 km upper boundary (vertical).

With respect to the "10 km upper boundary (vertical)" threshold, this was selected on the basis of CALIOP observations over the North Atlantic Ocean – Saharan Aerosol Layer (SAL). In should be noted that the approach assumes no leak at the top of the atmospheric columns, and no dust sources over the Atlantic Ocean. The net input and output fluxes over closed hypothetical cuboid surfaces should equal zero, assuming no sources/sinks inside the conceptual column. With respect to the vertical extend of the transported pure-dust over the Atlantic Ocean, we implement one year (2010) of CALIPSO overpasses to investigate the climatological maximum altitude in the SAL, for the December-January-February (DJF), March-April-May (MAM), June-July-August (JJA), and September-October-November (SON). According to the atmospheric dust climatology, the vertical extend is not expended -on a

---

## Author Comment (AC2)

Quantifying Dust Deposition over the Atlantic Ocean

The manuscript will become a strong contribution to the literature which deals with desert dust in the Earth atmospheric system (from the source region to the deposition region). The use of modern active spaceborne remote sensing and state of the art atmospheric modelling to investigate the flux of desert dust into the Atlantic is a very important topic in Earth science.

The authors would like to thank the reviewer for his time, comments and suggestions. We did our best to incorporate the proposed changes and corrections in the revised manuscript, aiming at improving the presented paper.
Following, you will find our responses, one by one to the comments addressed.

Kind regards,
Emmanouil Proestakis et al.

Reviewer's Comments

The paper is well written, however, to my opinion, it is much too long and tries to cover two main topics that can easily be separated. The length of the manuscript and the attempt to cover many different aspects in ONE paper leaves behind a confusing impression of the entire work.
I strongly recommend to separate this manuscript into two parts, i.e., in two articles!
Part 1:  Sections 1, 2.1.1, 2.2, 3, 4, 6 plus respective summary
Part 2: Sections 1, 2.1.2, 2.1.3, 2.1.4, 2.1.5., 5 and respective summary
If the authors do not want to follow my recommendation, then one should at least move section 5 to the end, before the summary. It is confusing when one switches from measurements and observational facts to comparisons with model products and then, again back to the observations and comparison with published (literature) values. For me it is very clear, the presentation in two papers will attract more readers to study BOTH articles carefully. Many readers will give up to study the contents carefully if it remains an overloaded paper.

We sincerely thank the reviewer for the thoughtful and detailed feedback, including the manuscript's scope and structure. We have carefully considered the suggestion to split the manuscript into two separate parts, and while we acknowledge the arguments and appreciate the intent behind the recommendation, following internal discussion we have decided towards maintaining the work as a single, unified paper. The decision is underpinned by a number of points. As a unified work, the study provides a full scientific narrative, from conceptualization to generation of the geoinformation products of (i) atmospheric dust over- and (ii) dust deposition rate across- the broader Atlantic Ocean, evaluation, discussion of the outcomes and spatiotemporal patterns, and finally, to AeroVal integration. It should be mentioned, that central part in the decision of not separating the present work into two parts played the scarcity of in-situ observations of dust deposition fluxes as reference datasets, largely attributed to the numerous significant challenges inherent to the complex nature of oceanographic research. This scarcity hampers the potential for a completely rigorous validation, thus for verification of the accuracy and reliability, and full quantification of the uncertainties of the satellite-based estimations of dust deposition rate. As such, the final part of AeroVal EO-based dust deposition intercomparison with state-of-the-art ESMs, serves two purposes. First, to provide an indicative example of the wide range of potential applications of the developed EO-based climate data record with focus on supporting the evaluation assessment of model simulation outputs. Second, in absence of validation/evaluation extended in-situ records to enhance the understanding of the developed dataset, addressing the whether the EO-based dust deposition rate product and ESM deposition estimates share similar characteristics in terms of spatio-temporal variability. As such separating the section of ESMs would weaken the discussion of the generated product itself. Splitting them would disrupt the logical flow and weaken the overall message. Moreover, though we have considered the reviewer's suggestion carefully, we have decided towards not splitting the present paper into two publications, since though the paper in an extended work is not to the degree that would fully support publishing the ESMs intercomparison without substantial extension and discussion of the outcomes. In addition, we have considered that separating the work in two studies would lead to unnecessary redundancy, particularly in the introduction, methodology, and contextual framing. We also feel that key insights gained from cross-comparisons between observation and model would be diluted if presented separately.
However, considering carefully the reviewer's suggestion, and in order to improve clarity, we have revised the manuscript to enhance the transitions between sections and have taken the suggestion to move Section 5 closer to

the summary to create a smoother progression. This adjustment, we believe, addresses the concern about switching between topics, while preserving the scientific strength of an integrated paper.

I have a number of comments.

1.  line 23: According to the new rules of ACP/AMT/ESSA the abstract should contain 250-300 words, only.

The authors agree with the reviewer, the abstract exceeded ACP/AMT/ESSA rules, and was too detailed. The abstract was re-written.

From (390 words):
"Quantification of atmospheric dust deposition into the Atlantic Ocean is provided. The estimates rely on the four-dimensional structure of atmospheric dust provided by the European Space Agency (ESA) - "LIdar climatology of Vertical Aerosol Structure" (LIVAS) climate data record (CDR) established on the basis of Cloud–Aerosol Lidar and Infrared Pathfinder Satellite Observations (CALIPSO) – Cloud-Aerosol Lidar with Orthogonal Polarization (CALIOP) routine observations. The data record of atmospheric dust deposition rate is provided for the broader Atlantic Ocean region, the Caribbean Sea, and the Gulf of Mexico, confined between latitudes 60°S to 40°N, and is characterized by 5° (zonal) x 2° (meridional) spatial resolution, seasonal-mean temporal resolution, and for the period extending between 12/2006 and 11/2022. The estimates of dust deposition are evaluated on the basis of sediment-trap measurements of deposited lithogenic material implemented as reference dataset with good agreement between the two datasets, revealing the capacity of the satellite-based product to quantitatively provide the amount of dust deposited into the Atlantic Ocean region, as shown by the evaluation intercomparison, evaluation intercomparison characterized by correlation coefficient ~0.79 and mean bias of 5.42 $mg/m^2d$. Moreover, integration of the satellite-based dust deposition rate dataset into AeroVal allows assessment comparison of the variability amongst the dust deposition CDR and dust deposition field estimates provided by the Multiscale Online Nonhydrostatic AtmospheRe CHemistry (MONARCH), EMEP MSC-W, and EC-Earth3-Iron Earth System Models (ESM), with the comparison revealing the capacity of the satellite-based product to follow the seasonal activation of dust source regions and the four-dimensional migration of dust transport pathways. Overall, the annual-mean amount of dust deposition into the Atlantic Ocean is estimated at $274.79 \pm 31.64$ Tg $yr^{-1}$, of which $243.98 \pm 23.89$ Tg $yr^{-1}$ of dust is deposited into the North Atlantic Ocean and $30.81 \pm 10.49$ Tg $yr^{-1}$ of dust is deposited into the South Atlantic Ocean. Moreover, a negative statistically significant trend in Atlantic Ocean dust deposition is also revealed. The satellite-based dust deposition CDR is considered unique with respect to a wide range of potential applications, including compensating for geographical and temporal gaps of sediment-trap measurements, supporting evaluation assessments of model simulations, shedding light into physical processes related to the cycle of dust from emission to transport and eventually deposition, and providing a solid basis to better understand dust biogeochemical impacts on oceanic ecosystems, as well as impacts on weather and climate.".

To (305 words):
"Quantification of atmospheric dust deposition into the Atlantic Ocean is provided. The estimates rely on the four-dimensional structure of atmospheric dust provided by the ESA-LIVAS climate data record established on the basis of CALIPSO-CALIOP observations. The data record of atmospheric dust deposition rate is provided for the Atlantic Ocean region, between latitudes 60°S to 40°N, and is characterized by 5° (zonal) x 2° (meridional) spatial resolution, seasonal-mean temporal resolution, for the period 12/2006-11/2022. The estimates of dust deposition are evaluated on the basis of sediment-trap measurements of deposited lithogenic material. The evaluation intercomparison shows a good agreement between the two datasets, revealing the capacity of the satellite-based product to quantitatively provide the amount of dust deposited into the Atlantic Ocean, characterized by correlation coefficient of 0.79 and mean bias of 5.42 $mgm^{-2}d^{-1}$. Integration of the satellite-based dust deposition rate dataset into AeroVal allowed assessment comparison of the dust deposition product and dust deposition field estimates provided by MONARCH, EMEP MSC-W, and EC-Earth3-Iron ESMs. The comparison revealed the capacity of the satellite-based product to follow the seasonal activation of dust source regions and the four-dimensional migration of dust transport pathways. Overall, the annual-mean amount of dust deposition into the Atlantic Ocean is estimated at $274.79 \pm 31.64$ Tg$yr^{-1}$, of which $243.98 \pm 23.89$ Tg$yr^{-1}$ of dust is deposited into the North Atlantic Ocean and $30.81 \pm 10.49$ Tg$yr^{-1}$ of dust is deposited into the South Atlantic Ocean. Moreover, a negative statistically significant trend in Atlantic Ocean dust deposition is revealed. The satellite-based dust deposition product is considered unique with respect to a wide range of

potential applications, including compensating for geographical and temporal gaps of sediment-trap measurements, supporting evaluation assessments of model simulations, unravelling physical processes related to the atmospheric cycle of dust, and providing better understanding of dust biogeochemical impacts on oceanic ecosystems, weather, and climate.".

2. line 33: … evaluation intercomparison … is given 2x.

   Following the reviewer's comment the sentence is corrected.

3. line 88: To my opinion, the paper of Kaufmann et al. (JGR, 2005) belongs to the first publications that systematically investigated dust deposition over the North Atlantic. For me, it was the first trustworthy attempt to quantify dust deposition into the ocean. Therefore, I would mention Kaufmann et al. (2005) already in the introduction.

   According to the reviewer's recommendation the following reference to Kaufmann et al. (JGR, 2005) was included to the introduction section:
   "Kaufman et al. (2005), on the basis of Terra - Moderate Resolution Imaging Spectroradiometer (MODIS) routine aerosol observations, quantified the amount of Saharan dust deposited into the Atlantic Ocean and the Caribbean Sea to approximately 190 Tgyr$^{-1}$.".

4. line 147: Do you want to say: …has been extensively …  used to analyze ... CALIOP and CATS optical products…? Some words are missing.

   We would like to thank the reviewer. The word "applied" was missing.

5. line 192: It is not clear what you explicitly did with the aerosol height profiles! This should be better explained. You calculated daily mean dust profiles in the first step? In the next step, you formed monthly means for given lat-long grid cells?
   In the framework of the study not the original ESA-LIVAS four-dimensional global pure-dust CDR is applied, but a modified version of different spatial and temporal resolution. More specifically, following the same original CALIPSO-CALIOP L2 5km aerosol profiles and quality-assurance criteria, the original ESA-LIVAS four-dimensional global pure-dust CDR is characterized at Level 3 by 1°x1° deg$^2$ spatial-resolution and monthly-mean temporal resolution, while the updated tailored LIVAS-based Level 3 CDR implemented in the framework of the study is characterized by 5° (zonal) x 2° (meridional) spatial resolution and seasonal-mean temporal resolution. In the framework of the Level 2 - to - Level 3 aggregation and averaging method, the ESA-LIVAS CDR applied first per-overpass aerosol profile averaging, and accordingly, averaging of all mean per-overpass profiles, as mentioned by the reviewer. This approach is selected due to the "relatively-small" geographical area of each 1°x1° deg$^2$, where due to CALIPSO orbital characteristics, CALIOP provides few overpass observations per month (~4 overpasses/month close to the equator), resulting to weighting thus in representativeness issues, in case of different approach. However, increasing the grid area and decreasing the spatial resolution as in the present study as much as 10 times, to 5° (zonal) x 2° (meridional) spatial resolution, offers increasing flexibility in the aggregation and averaging method. More specifically, the spatial resolution of the present study is equal to the original CALIPSO Level 3 spatial resolution, allowing to follow the same aggregation and averaging method to the original CALIPSO Level 3 products. As such, and as described in Tackett et al. (2018), the following averaging method is applied:
   Mean aerosol extinction is calculated from all quality-screened level 2 dust aerosol extinction coefficients (σ) and clear-air samples within each latitude, longitude, altitude grid cell using Eq. (1).

$$\bar{\sigma} = \frac{\sum_{i=1}^{N_{aer}} \sigma_{aer,i}}{N_{avg}} \tag{1}$$

   Here, $\bar{\sigma}$ is the seasonally mean aerosol extinction coefficient, $N_{aer}$ is the total number of aerosol extinction samples accepted, and $N_{clear}$ is the number of clear-air samples in the grid cell, resulting to Eq.(1) under the assumption that $\sigma_{clear} = 0\ km^{-1}$ and the definition $N_{avg} = N_{aer} + N_{clear}$. In detail description of the averaging approach is provided in Tackett et al. (2018) - Sect.4.3 - "Averaging method".

However, the authors agree with the reviewer's comment on the methodology and the approach, and the following addition is included in Sect.2.1.1 - "CALIPSO-CALIOP":
"Accordingly, mean pure dust extinction coefficient at 532 nm profiles within each 5° (zonal) x 2° (meridional) spatial resolution are produced through aggregation and averaging of all quality-assured cloud-free level 2 5 km pure dust profiles located within each latitude, longitude, altitude grid cell per each season, following the averaging approach provided and discussed in Tackett et al. (2018).".

6. line 202: Hobbs (2006) is not in the references.

The reviewer is right, the reference was wrong with a typo. The correct reference is: "Wallace and Hobbs, 2006".
The following reference was added to the Section "References":
Wallace, J. M. and Hobbs, P. V.: 3 – Atmospheric Thermodynamics, in: Atmospheric Science, 2nd edn., Academic Press, San Diego, 63–111, https://doi.org/10.1016/B978-0-12-732951-2.50008-9, 2006.

7. line 209 Doescher et al. (2022) is not in the references. There many cited papers not in the references. Please check carefully all references.

The reviewer is right, a number of references were not included in the Section "References".
We have gone through the references list and the following scientific studies that were identified used in the manuscript though not included in the references list, were added (in alphabetical order):

| | |
|---|---|
| Ben-Ami et al., 2010: | Ben-Ami, Y., Koren, I., Rudich, Y., Artaxo, P., Martin, S. T., and Andreae, M. O.: Transport of North African dust from the Bodélé depression to the Amazon Basin: a case study, Atmospheric Chemistry and Physics, 10, 7533–7544, https://doi.org/10.5194/acp-10-7533-2010, 2010. |
| Berrisford et al., 2009: | Berrisford P, Dee DP, Fielding K, Fuentes M, Kållberg P, Kobayashi S, Uppala SM. 2009. 'The ERA-Interim Archive'. ERA Report Series, No.1. ECMWF: Reading, UK. |
| Chin et al., 2002: | Chin, M., Ginoux, P., Kinne, S., Torres, O., Holben, B. N., Duncan, B. N., Martin, R. V., Logan, J. A., Higurashi, A., and Nakajima, T.: Tropospheric Aerosol Optical Thickness from the GOCART Model and Comparisons with Satellite and Sun Photometer Measurements, 2002. |
| Dee et al., 2011: | Dee, D. P., Källén, E., Simmons, A. J., and Haimberger, L.: Comments on "Reanalyses Suitable for Characterizing Long-Term Trends," https://doi.org/10.1175/2010BAMS3070.1, 2011. |
| Döscher et al., 2022: | Döscher, R., Acosta, M., Alessandri, A., Anthoni, P., Arsouze, T., Bergman, T., Bernardello, R., Boussetta, S., Caron, L.-P., Carver, G., Castrillo, M., Catalano, F., Cvijanovic, I., Davini, P., Dekker, E., Doblas-Reyes, F. J., Docquier, D., Echevarria, P., Fladrich, U., Fuentes-Franco, R., Gröger, M., v. Hardenberg, J., Hieronymus, J., Karami, M. P., Keskinen, J.-P., Koenigk, T., Makkonen, R., Massonnet, F., Ménégoz, M., Miller, P. A., Moreno-Chamarro, E., Nieradzik, L., van Noije, T., Nolan, P., O'Donnell, D., Ollinaho, P., van den Oord, G., Ortega, P., Prims, O. T., Ramos, A., Reerink, T., Rousset, C., Ruprich-Robert, Y., Le Sager, P., Schmith, T., Schrödner, R., Serva, F., Sicardi, V., Sloth Madsen, M., Smith, B., Tian, T., Tourigny, E., Uotila, P., Vancoppenolle, M., Wang, S., Wårlind, D., Willén, U., Wyser, K., Yang, S., Yepes-Arbós, X., and Zhang, Q.: The EC-Earth3 Earth system model for the Coupled Model Intercomparison Project 6, Geosci. Model Dev., 15, 2973–3020, https://doi.org/10.5194/gmd-15-2973-2022, 2022. |
| Eck et al., 1999: | Eck, T. F., Holben, B. N., Reid, J. S., Dubovik, O., Smirnov, A., O'Neill, N. T., Slutsker, I., and Kinne, S.: Wavelength dependence of the optical depth of biomass burning, urban and desert dust aerosols, J. Geophys. Res., 104, 31333–31350, 1999. |
| Fairlie et al., 2007: | Duncan Fairlie, T., Jacob, D. J., and Park, R. J.: The impact of transpacific transport of mineral dust in the United States, Atmospheric Environment, 41, 1251–1266, https://doi.org/10.1016/j.atmosenv.2006.09.048, 2007. |

Gassó et al., 2010:        Gassó, S., Stein, A., Marino, F., Castellano, E., Udisti, R., and Ceratto, J.: A combined observational and modeling approach to study modern dust transport from the Patagonia desert to East Antarctica, Atmos. Chem. Phys., 10, 8287–8303, https://doi.org/10.5194/acp-10-8287-2010, 2010.

Gates et al., 1999:        Gates, W. L., Boyle, J. S., Covey, C., Dease, C. G., Doutriaux, C. M., Drach, R. S., Fiorino, M., Gleckler, P. J., Hnilo, J. J., Marlais, S. M., Phillips, T. J., Potter, G. L., Santer, B. D., Sperber, K. R., Taylor, K. E., and Williams, D. N.: An Overview of the Results of the Atmospheric Model Intercomparison Project (AMIP I), 1999.

Gidden et al., 2019:       Gidden, M. J., Riahi, K., Smith, S. J., Fujimori, S., Luderer, G., Kriegler, E., van Vuuren, D. P., van den Berg, M., Feng, L., Klein, D., Calvin, K., Doelman, J. C., Frank, S., Fricko, O., Harmsen, M., Hasegawa, T., Havlik, P., Hilaire, J., Hoesly, R., Horing, J., Popp, A., Stehfest, E., and Takahashi, K.: Global emissions pathways under different socioeconomic scenarios for use in CMIP6: a dataset of harmonized emissions trajectories through the end of the century, Geoscientific Model Development, 12, 1443–1475, https://doi.org/10.5194/gmd-12-1443-2019, 2019.

Haustein et al., 2012:     Haustein, K., Pérez, C., Baldasano, J. M., Jorba, O., Basart, S., Miller, R. L., Janjic, Z., Black, T., Nickovic, S., Todd, M. C., Washington, R., Müller, D., Tesche, M., Weinzierl, B., Esselborn, M., and Schladitz, A.: Atmospheric dust modeling from meso to global scales with the online NMMB/BSC-Dust model – Part 2: Experimental campaigns in Northern Africa, Atmospheric Chemistry and Physics, 12, 2933–2958, https://doi.org/10.5194/acp-12-2933-2012, 2012.

Hoesly et al., 2018:       Hoesly, R. M., Smith, S. J., Feng, L., Klimont, Z., Janssens-Maenhout, G., Pitkanen, T., Seibert, J. J., Vu, L., Andres, R. J., Bolt, R. M., Bond, T. C., Dawidowski, L., Kholod, N., Kurokawa, J.-I., Li, M., Liu, L., Lu, Z., Moura, M. C. P., O'Rourke, P. R., and Zhang, Q.: Historical (1750–2014) anthropogenic emissions of reactive gases and aerosols from the Community Emissions Data System (CEDS), Geosci. Model Dev., 11, 369–408, https://doi.org/10.5194/gmd-11-369-2018, 2018.

Iacono et al., 2000:       Iacono, M. J., Mlawer, E. J., Clough, S. A., and Morcrette, J. J.: Impact of an improved longwave radiation model, RRTM, on the energy budget and thermodynamic properties of the NCAR community climate model, CCM3, J. Geophys. Res., 105, 14873–14890, 2000.

Iacono et al., 2008:       Iacono, M. J., Delamere, J. S., Mlawer, E. J., Shephard, M. W., Clough, S. A., and Collins, W. D.: Radiative forcing by long-lived greenhouse gases: Calculations with the AER radiative transfer models, Journal of Geophysical Research: Atmospheres, 113, https://doi.org/10.1029/2008JD009944, 2008.

Ito et al., 2019:          Ito, A., Myriokefalitakis, S., Kanakidou, M., Mahowald, N. M., Scanza, R. A., Hamilton, D. S., Baker, A. R., Jickells, T., Sarin, M., Bikkina, S., Gao, Y., Shelley, R. U., Buck, C. S., Landing, W. M., Bowie, A. R., Perron, M. M. G., Guieu, C., Meskhidze, N., Johnson, M. S., Feng, Y., Kok, J. F., Nenes, A., and Duce, R. A.: Pyrogenic iron: The missing link to high iron solubility in aerosols, Sci. Adv., 5, eaau7671, https://doi.org/10.1126/sciadv.aau7671, 2019.

Janjic, 2003:              Janjic, Z. I.: A nonhydrostatic model based on a new approach, Meteorol Atmos Phys, 82, 271–285, https://doi.org/10.1007/s00703-001-0587-6, 2003.

Janjic and Gall, 2012:     Janjic, A. Z. and Gall, A. R. L.: Scientific documentation of the NCEP nonhydrostatic multiscale model on the B grid (NMMB). Part 1 Dynamics, n.d.

Klose et al., 2021:        Klose, M., Jorba, O., Gonçalves Ageitos, M., Escribano, J., Dawson, M. L., Obiso, V., Di Tomaso, E., Basart, S., Montané Pinto, G., Macchia, F., Ginoux, P., Guerschman, J., Prigent, C., Huang, Y., Kok, J. F., Miller, R. L., and Pérez García-Pando, C.: Mineral dust cycle in the Multiscale Online Nonhydrostatic AtmospheRe CHemistry model (MONARCH) Version 2.0, Geoscientific Model Development, 14, 6403–6444, https://doi.org/10.5194/gmd-14-6403-2021, 2021.

Kok, 2011:                 Kok, J. F.: A scaling theory for the size distribution of emitted dust aerosols

suggests climate models underestimate the size of the global dust cycle, PNAS, 108, 1016–1021, https://doi.org/10.1073/pnas.1014798108, 2011.

Krishnamurthy et al., 2010:    Krishnamurthy, A., Moore, J. K., Mahowald, N., Luo, C., and Zender, C. S.: Impacts of atmospheric nutrient inputs on marine biogeochemistry, J. Geophys. Res., 115, G01006, https://doi.org/10.1029/2009JG001115, 2010.

Luo et al., 2003:    Luo, C., Mahowald, N. M., and del Corral, J.: Sensitivity study of meteorological parameters on mineral aerosol mobilization, transport, and distribution, Journal of Geophysical Research: Atmospheres, 108, https://doi.org/10.1029/2003JD003483, 2003.

Myriokefalitakis et al., 2021:    Myriokefalitakis, S., Bergas-Massó, E., Gonçalves-Ageitos, M., Pérez García Pando, C., van Noije, T., and Le Sager, P.: EC-Earth3.3.2.1-Fe., Zenodo [data set], https://doi.org/10.5281/zenodo.5752596, 2021.

Myriokefalitakis et al., 2022:    Myriokefalitakis, S., Bergas-Massó, E., Gonçalves-Ageitos, M., Pérez García-Pando, C., van Noije, T., Le Sager, P., Ito, A., Athanasopoulou, E., Nenes, A., Kanakidou, M., Krol, M. C., and Gerasopoulos, E.: Multiphase processes in the EC-Earth model and their relevance to the atmospheric oxalate, sulfate, and iron cycles, Geosci. Model Dev., 15, 3079–3120, https://doi.org/10.5194/gmd-15-3079-2022, 2022.

van Noije et al., 2021:    van Noije, T., Bergman, T., Le Sager, P., O'Donnell, D., Makkonen, R., Gonçalves-Ageitos, M., Döscher, R., Fladrich, U., von Hardenberg, J., Keskinen, J.-P., Korhonen, H., Laakso, A., Myriokefalitakis, S., Ollinaho, P., Pérez García-Pando, C., Reerink, T., Schrödner, R., Wyser, K., and Yang, S.: EC-Earth3-AerChem: a global climate model with interactive aerosols and atmospheric chemistry participating in CMIP6, Geoscientific Model Development, 14, 5637–5668, https://doi.org/10.5194/gmd-14-5637-2021, 2021.

O'Neill et al., 2001a:    O'Neill, N. T., Eck, T. F., Holben, B. N., Smirnov, A., Dubovik, O., and Royer, A.: Bimodal size distribution influences on the variation of Angstrom derivatives in spectral and optical depth space, J. Geophys. Res., 106, 9787–9806, 2001a.

O'Neill et al., 2001b:    O'Neill, N. T., Dubovik, O., and Eck, T. F.: A modified Angstrom coefficient for the characterization of sub-micron aerosols, App. Opt., 40, 2368–2374, 2001b.

O'Neill et al., 2003:    O'Neill, N. T., Eck, T. F., Smirnov, A., Holben, B. N., and Thulasiraman, S.: Spectral discrimination of coarse and fine mode optical depth, J. Geophys. Res., 108, 4559–4573, https://doi.org/10.1029/2002JD002975, 2003.

Pérez et al., 2011:    Pérez, C., Haustein, K., Janjic, Z., Jorba, O., Huneeus, N., Baldasano, J. M., Black, T., Basart, S., Nickovic, S., Miller, R. L., Perlwitz, J. P., Schulz, M., and Thomson, M.: Atmospheric dust modeling from meso to global scales with the online NMMB/BSC-Dust model and ash; Part 1: Model description, annual simulations and evaluation, Atmospheric Chemistry and Physics, 11, 13001–13027, https://doi.org/10.5194/acp-11-13001-2011, 2011.

Prospero and Carlson, 1981:    Prospero, J. M. and Carlson, T. N.: Saharan air outbreaks over the tropical North Atlantic, Pure Appl. Geophys., 119, 677–691, https://doi.org/10.1007/BF00878167, 1981.

Vickery et al., 2013:    Vickery, K. J., Eckardt, F. D., and Bryant, R. G.: A sub-basin scale dust plume source frequency inventory for southern Africa, 2005–2008, Geophysical Research Letters, 40, 5274–5279, https://doi.org/10.1002/grl.50968, 2013.

Vignati et al., 2004:    Vignati, E., Wilson, J., and Stier, P.: M7: An efficient size-resolved aerosol microphysics module for large-scale aerosol transport models, Journal of Geophysical Research: Atmospheres, 109, https://doi.org/10.1029/2003JD004485, 2004.

8. line 212: The references are in a bad shape, the year 2021 is 2020, the year 2022 is 2023.

The references are checked.

9.  line 252: Perez et al., Haustein et al., Klose et al. (2021), all these papers are not in the references! I will stop here complaining about missing literature. There are many cited papers that are not in the references list.

    The cites papers have been included in the Section "References". P
    lease see "Comment 7 - Response".

10. line 272: Table 1 contains basic information about the models. But, what is the exact data analysis procedure? Is it: (1) Identification of dust in the model outputs, (2) selection of all dust events per month and (3) computation of monthly means of respective dust products? Is that the procedure applied to all model data? Is the data analysis procedure similar to the way LIVAS data are computed? All this needs to be well described so that the reader can better follow the discussion later on, on the partly huge differences.

    We thank the reviewer for pointing out that a better explanation is needed for model data processing:
    (1) Identification of dust in the model outputs:
        Dust particles are traced in the models, including their dry and wet removal. Thus, dust concentrations, AOD, and dry/wet deposition are direct outputs from the models and were provided as daily fields (at their respective grids).
    (2) selection of all dust events per month:
        All models provided dust AOD and deposition on a daily basis, which were averaged to obtain monthly means for AeroVal comparison. This means that the dataset included all days with dust events (as well as not-dusty days).
    (3) computation of monthly means of respective dust products? Is that the procedure applied to all model data?
        That is correct. Gridded monthly mean dust AOD and depositions were calculated in the same way (i.e. by averaging the gridded daily fields) for all models.
    (4) Is the data analysis procedure similar to the way LIVAS data are computed?
        The extraction of the atmospheric dust mass concentration profiles in terms of backscatter coefficient at 532 nm, extinction coefficient at 532 nm, and mass concentration profiles and DODs at 532 nm is discussed and presented in Section 2.2 – "Methodology" and more specifically at the subsection 2.2.1 – "Decoupling the atmospheric pure-dust component from the total aerosol load". Hoverer, here below we include an additional example of the analysis to clarify it more. More specifically, starting from CALIOP backscatter coefficient at 532 nm profiles and the particulate depolarization ratio at 532 nm profiles – L2 5km – along the CALIPSO overpass, the pure dust backscatter coefficient at 532 nm profiles along the CALIPSO overpass are extracted. As next steps the backscatter coefficient at 532 nm profiles are converted to extinction coefficient at 532 nm profiles along the CALIPSO overpass through the implementation of suitable dust LRs. As a next step the dust extinction coefficient at 532 nm profiles are converted to mass concentration profiles through the implementation of suitable EARLINET-AERONET established conversion factors. Spatiotemporal averaging according to selected resolution provides the mean profiles. Finally, vertical integration of the mean extinction coefficient at 532 nm profiles over an area with respect to altitude provides the DOD at 532 nm.

[Figure]

↓

Pure-Dust Backscatter Coefficient at 532 nm

[Figure]

Processing CALIPSO overpasses                          Mean Pure-Dust
                                                       Extinction Coefficient at 532nm profile

[Figure]

Moreover, according to the reviewer's recommendation, the following description has been added in the manuscript:

"Dust concentrations, AOD, and dry/wet deposition rates are direct outputs from the models and were provided as daily fields (at their respective grids), which were- then averaged to obtain the monthly mean fields to be used in the AeroVal evaluations and consistency checks. No special selection of days with dust events has been made, so that the monthly mean values include both data in dusty and not-dusty days, consistent with EO-based dust products used here.".

11. line 316, Figure 2: Please mention the aerosol types in Fig.2c? Also, what are the feature types in Fig.2b? Please explain in the figure caption, or in the panels.

According to the reviewer's recommendation, the figures have been changed as follows:

[Figure]

12. Note the following general remark to the UNITS! To my knowledge, ESSA/ACP/AMT only accept, e.g., 'm s^-1' and not 'm/s' …. , 'μg cm^-3' and not 'μg/cm^+3' …, 'mg m^-2 d^-1' and not 'mg/(m^+2 d)'.

We have gone through the manuscript's text and units have been adapted to the journal's unit format.

13. You write 'mg/m+2 d' is mathematically bad. Correct is: 'mg/(m+2 d)'

The unit's format is corrected.

14. line 333: Haarig.

First author's name corrected.

15. line 335: Provide references for pollen depolarization ratios, maybe Sicard et al. (20??) and recent papers from the Finish lidar group (2022-2024).

According to the reviewer's recommendation the following references were in addition included:
- Sicard, M., Izquierdo, R., Alarcón, M., Belmonte, J., Comerón, A., and Baldasano, J. M.: Near-surface and columnar measurements with a micro pulse lidar of atmospheric pollen in Barcelona, Spain, Atmos. Chem. Phys., 16, 6805–6821, https://doi.org/10.5194/acp-16-6805-2016, 2016.
- Shang, X., Giannakaki, E., Bohlmann, S., Filioglou, M., Saarto, A., Ruuskanen, A., Leskinen, A., Romakkaniemi, S., and Komppula, M.: Optical characterization of pure pollen types using a multi-wavelength Raman polarization lidar, Atmos. Chem. Phys., 20, 15323–15339, https://doi.org/10.5194/acp-20-15323-2020, 2020.
- Bohlmann, S., Shang, X., Vakkari, V., Giannakaki, E., Leskinen, A., Lehtinen, K. E. J., Pätsi, S., and Komppula, M.: Lidar depolarization ratio of atmospheric pollen at multiple wavelengths, Atmos. Chem. Phys., 21, 7083–7097, https://doi.org/10.5194/acp-21-7083-2021, 2021.

16. line 340: I would prefer all depol numbers in %.

Central component of the present study are CALIPSO-CALIOP optical products. Among the optical products implements are the aerosol profiles of particulate depolarization ratio at 532 nm. Since CALIOP observations are

so central in the study, the authors have preferred to used the same conversion for all optical products, including among others the depolarization between 0 and 1.0, to be consistent with CALIOP conversion.
For more information please visit: https://www-calipso.larc.nasa.gov/products/lidar/browse_images/production/ (last visit: 19/04/2025).

17. line 361: Haarig, Bohlmann.

Corrected.

18. line 365: Here, you can introduce $C_{v,d}$. The conversion factor needs to be introduced somewhere.

According to the reviewer's recommendation, the following part was included in the methodology section, providing briefly the input required along with main scientific publications discussing the implemented in the framework of the study performed developments:

"Finally, regionally-dependent extinction coefficient at 532 nm (Fig. 2g) to mass concentration conversion factors (Ansmann et al., 2019) and typical particle density of $\rho d$: 2.6 g cm-3 for dust (Ansmann et al., 2012) are applied towards establishing the final pure-dust mass concentration product (Fig. 2h) along the CALIPSO orbit-path (Eq. (3)). The climatologically representative extinction coefficient-to-mass concentration conversion factors, discussed in detail by Mamouri and Ansmann (2014, 2015, 2016, 2017) and Ansmann et al. (2019) were determined on the basis of AERONET (Version 3; Level 2.0) long-term observations during atmospheric conditions characterized by dust presence (Ångström exponent <0.3 and AOT > 0.1), synergistically obtained with lidar-provided particle extinction coefficient profiles, allowing eventually provision of extinction-to-volume conversion factors $C_{v,d}$.".

19. line 371: I am confused, you calculate seasonal means (by using data collected in 15 years) and in Figure 3 you show annual means.

The ESA-LIVAS database provides the four-dimensional, multiyear, and near-global structure of atmospheric dust, component of the total aerosol load. The climate data record (CDR) is established on the basis of long-term CALIOP observations and optical products, along the CALIPSO orbit path. The final products consist of the atmospheric dust, quality-assured profiles of backscatter coefficient at 532 nm, extinction coefficient at 532 nm, and mass concentration. The datasets are established primarily with the original L2 horizontal (5 km) and vertical (60 m) resolution of CALIOP along the CALIPSO orbit path and secondly in averaged profiles of seasonal–temporal resolution, 1°×1° spatial resolution, and the original vertical resolution of CALIPSO, focusing on the latitudinal band extending between 70° S and 70° N and covering more the CALIPSO lifetime period (June 2006–August 2023). As such, the CDR is dynamically created in different spatiotemporal resolutions, tailored each time to the different scientific questions the database aims to address. In the framework of the present study and following a number of feasibility tests it was decided that the ESA-LIVAS L3 had to be updated with spatial resolution from 1°×1° deg$^2$ grids to 5°×2° deg$^2$ grids and temporal resolution from monthly-mean to seasonal-mean. Figure 3 though, as part of the methodology Section, aims to provide information on the climate data record, and through the briefly discussed approach, an image of the long-term mean atmospheric dust conditions over the region of interest. As such the original ESA-LIVAS CDR in terms of spatial resolution (finer) is implemented, providing the fifteen-years mean dust atmospheric conditions in terms of total backscatter coefficient at 532 nm (Fig. 3a), dust backscatter coefficient at 532 nm (Fig. 3b), dust extinction coefficient at 532 nm (Fig. 3c), and dust mass concentration (Fig. 3d). However, the reviewer is right that since the figure provides the long-term mean the four-dimensional is reduced to three-dimensional. As such the text and the caption of the figure are adjusted and in addition a reference to the original LIVAS in terms of spatiotemporal resolution of the figure was included.

20. line 380 Figure 3: Only the experienced eye is able to get an idea what is shown. The readers not familiar with lidar (or CALIOP) are probably lost without further explanations! Where is Africa, where is South America? What do the color plumes show? …height-latitude cross sections of x,y,z…..! What does the longitude of -100° mean? What does 4D mean here, I only see 3D?

According to the reviewer's comment, the following description relative to figure 3 is inserted:

"Based on the ESA-LIVAS dust CDR, figure 3 provides the long-term annual-mean total backscatter coefficient at 532 nm (Fig. 3a), pure-dust backscatter coefficient at 532 nm (Fig. 3b), pure-dust extinction coefficient at 532 nm (Fig. 3c), and pure-dust mass concentration (Fig. 3d), with the later one subsequently implemented with spatial resolution from $1°\times1°$ deg$^2$ grids to $5°\times2°$ deg$^2$ grids and temporal resolution from monthly-mean to seasonal-mean, towards computation of the atmospheric pure-dust component deposited into the broader Atlantic Ocean. More specifically, the figure constitutes a three-dimensional reconstruction of the atmosphere in the geographical region confined between 100°W and 20°E longitude (x-axis) and 60°S and 40°N latitude (y-axis), and for the altitudinal range 0 to 8 km a.m.s.l (z-axis). Through utilizing parallel slices in the atmosphere for every 20° longitude, each figure provides insight into the three-dimensional total atmospheric aerosol component, in terms of total backscatter coefficient at 532 nm (Fig. 3a), and for the dust component of the total aerosol load in terms of backscatter coefficient at 532 nm (Fig. 3b), extinction coefficient at 532 nm (Fig. 3c), and mass concentration (Fig. 3d), over the broader region covering South America (lower left corner of figures), part of North America (upper left corner of figures), Central-Western Africa (right side of figures), and the broader Atlantic Ocean confined in-between. The state of the atmospheric aerosol load in figure 3 is provided in terms of annual mean EO-based products computed for the period 12/2006-11/2022.".

21. line 384: I miss an equation for mass flux, or a clear definition, the link between mass concentrations and wind field components. How is the zonal and meridional dust flux and transport defined in terms of dust mass concentration and wind components?

On the basis of the ESA-LIVAS dust mass concentration (µg m$^{-3}$) profiles, and on the basis of ERA5 profiles of U and V components of wind (m s$^{-1}$), the mass flux (µg m$^{-2}$ s$^{-1}$) is computed per unit area (m$^2$). The mass flux is computed as the amount of mass transported per unit time (s) across a unit area (m$^2$) that is perpendicular to the direction of mass transport, thus in the conceptual approach adopted in the framework of the study in meridional and zonal directions, through the implementation of ERA5 U and V wind fields, respectively. Accordingly, considering the three-dimensional (3D) closed cuboid

consideration, of 5° length (zonal), 2° width (meridional), and 10 km height (vertical), with the base surface at 0 km a.m.s.l, the total mass flow rates (µg s$^{-1}$) across the meridional and zonal directions are computed, through the consideration of the total surface areas of the conceptual cuboid that atmospheric dust is transported through. The approach is discussed in Sect. 2.2.2 – "Extracting the atmospheric pure-dust component deposited into the Atlantic Ocean" and with the help of figures 4 and 5. However, following the reviewer's comment the section was enriched accordingly.

22. line 404: Having such definitions and equations, it is much easier to discuss Fig.4.

Please see response to the reviewer's comment #21.

23. Figure 4: In confused: There is almost no wind at height 4 km height and strong wind at 2 km height, and the mass concentrations at both heights are similar, and finally the mass fluxes at 2 and 4 km height are similar. Why is that?

Actually, this is a really good one. We would like to thank the reviewer for his/her really careful review of the manuscript, leading to identification of this error. The figure was wrong. Clearly. More specifically, during the creation of the three-dimensional quiver figure of ERA5 U and V components of wind as demonstration of the methodology, the wind vector was plotted upside-down, and as a result, the figure could not relate with the rest of the subplots of Figure 4. Following the reviewer's comment, the figure was corrected and replaced. Though we think that the figure at the present version, accompanied by the relative description, explanation, and

discussion provides a sufficiently clear explanation of the methodology, we have also added following the figure hereinafter a table including the (i) ERA5 Altitude [km], (ii) the U and V components of wind [m s$^{-1}$], and (iii) the atmospheric dust Northward, Southward, Eastward, and Westward transported components [μg m$^{-3}$]. We hope that in this way the methodology is supported towards a more complete explanation of the methodology.

ERA5 U/V wind components [m s$^{-1}$]

[Figure]

Figure 4: Illustration case of the ERA5 U and V profiles of wind (Fig. 4b) in the zonal and meridional directions for the Atlantic Ocean area extending between 20° and 22° latitude and -30° and -25° longitude and for JJA 2020.

| ERA5 Height [km] | V [m s$^{-1}$] | Dust Mass Concentration [μg m$^{-3}$] | | U [m s$^{-1}$] | Dust Mass Concentration [μg m$^{-3}$] | |
|---|---|---|---|---|---|---|
| | | Northward Transported | Southward Transported | | Eastward Transported | Westward Transported |
| 7.63 | 1.01 | 0.00 | 0.00 | -3.31 | 0.00 | 0.00 |
| 6.73 | 1.13 | 0.01 | 0.00 | -4.76 | 0.00 | 0.19 |
| 5.91 | 1.73 | 0.17 | 0.00 | -5.93 | 0.00 | 1.99 |
| 5.16 | 2.43 | 1.35 | 0.00 | -6.79 | 0.00 | 10.49 |
| 4.46 | 2.62 | 4.37 | 0.00 | -7.22 | 0.00 | 33.27 |
| 3.81 | 2.09 | 3.66 | 0.00 | -7.38 | 0.00 | 45.78 |
| 3.20 | 1.09 | 1.14 | 0.00 | -7.53 | 0.00 | 54.41 |
| 2.62 | 0.00 | 0.00 | 0.00 | -7.78 | 0.00 | 45.92 |
| 2.34 | -0.49 | 0.00 | 0.18 | -7.91 | 0.00 | 46.11 |
| 2.06 | -0.93 | 0.00 | 0.64 | -8.00 | 0.00 | 47.75 |
| 1.80 | -1.38 | 0.00 | 1.24 | -8.08 | 0.00 | 42.86 |
| 1.54 | -1.84 | 0.00 | 1.82 | -8.16 | 0.00 | 35.59 |
| 1.29 | -2.32 | 0.00 | 2.32 | -8.22 | 0.00 | 29.12 |
| 1.04 | -2.91 | 0.00 | 3.67 | -8.19 | 0.00 | 29.07 |
| 0.81 | -3.71 | 0.00 | 5.92 | -8.02 | 0.00 | 27.63 |
| 0.58 | -4.57 | 0.00 | 6.91 | -7.91 | 0.00 | 20.66 |
| 0.35 | -5.23 | 0.00 | 7.18 | -7.50 | 0.00 | 14.77 |
| 0.13 | -5.11 | 0.00 | 4.31 | -6.63 | 0.00 | 7.24 |

24. Further questions here: Are all individual mass concentration profiles combined with respective wind profiles (in the first step)? And in the next step, the average flux profile (for JJA) is computed? Or did you combine the JJA mean dust profile in Fig. 4a with the JJA mean wind profile in Fig. 4b. All this may be explained in the text and I overlooked it! Sorry, if that is the case!

It is as the reviewer is saying, the JJA mean dust profile, as shown in Fig. 4a, is combined with the JJA mean wind profile, as shown in Fig. 4b. Since it seems it was not very clear, in Sect. 2.2.2 – "Extracting the atmospheric pure-dust component deposited into the Atlantic Ocean" the following sentence is added:
"The mass fluxes are computed on a seasonal basis, meaning based on seasonal-mean atmospheric dust profiles (Fig. 4a) and seasonal-mean profiles of ERA5 zonal and meridional wind components (Fig. 4c).".

25. What is the meaning of the color arrows at the horizontal bars in Fig. 4c? This indicates the wind speed (according to the color bar)? And the color of the bars indicate transport direction?

The reviewer is right. Since it seems it was not very clear, in the caption of Figure 4 the following sentence is added:
"In Fig. 4c the horizontal arrows indicates the wind speed, according to the colorbar, and the different colors in the legend indicate the dust transport direction.".

26. line 454: I miss a general section: RESULTS, …. before discussing the different products, and later on comparisons, uncertainties, etc. Section 3 is the main and most important section but presented as just one of many sections, from section 3 to section 6.

According to the reviewer's recommendation a generic section "Results and Discussion" was included, hosting the sub-sections, as follows:

3 Results and Discussion
      3.1 Atmospheric Dust and Dust Deposition
      3.2 Evaluation of EO-based Dust Deposition estimates
      3.3 Total Dust Deposited into the broader Atlantic Ocean
      3.4 EO-based Dust Deposition Rate vs respective ESM results

27. line 485: Besides HABOOBS (Knippertz et al., JGR, 2007, SAMUM campaign in Morocco) the SAMUM researchers even monitored dust devils with LIDAR in Morocco (Ansmann et al., Tellus, 61B, 2009, 10.3402/tellusb.v61i1.16833). Could be cited in this LIDAR paper on dust.

According to the reviewer's recommendation, the following Ansmann et al. papers are included:
Ansmann, A., Tesche, Matthias, Knippertz, Peter, Bierwirth, Eike, Althausen, Dietrich, MüLLER, Detlef, and Schulz, O.: Vertical profiling of convective dust plumes in southern Morocco during SAMUM, Tellus B: Chemical and Physical Meteorology, 61, 340–353, https://doi.org/10.1111/j.1600-0889.2008.00384.x, 2009.
Ansmann, A., Petzold, Andreas, Kandler, Konrad, Tegen, Ina, Wendisch, Manfred, Müller, Detlef, Weinzierl, Bernadett, Müller, Thomas, and Heintzenberg, J.: Saharan Mineral Dust Experiments SAMUM–1 and SAMUM–2: what have we learned?, Tellus B: Chemical and Physical Meteorology, 63, 403–429, https://doi.org/10.1111/j.1600-0889.2011.00555.x, 2011.

28. line 520: The Saharan Air Layer is also well illustrated, based on LIDAR observations, in the more recent papers by Weinzierl et al. (2016) and Rittmeister et al. (2017). These papers are already listed in the references section.

According to the reviewer's recommendation, the Weinzierl et al., 2016 and Rittmeister et al., 2017 were also referenced.

29. line 544: Haarig.

  First author's name corrected.

30. Page 22, Fig.6: The color scale for DDR is ok! One can even see the deposition values east of southern South America. The color scale for DOD is not ok. The DOD east of southern South America is not visible, at least not in the printed version. This not satisfactory.

According to the reviewer's comment, the DOD at 532 nm figures have been recreated and adapted, as follows:

[Figure]

Figure: LIVAS Dust Optical Depth at 532 nm (DOD), provided in annual-mean, DJF, MAM, JJA, and SON, estimated for the period 12/2006-11/2022. Left column: Initial version of figures. Right column: updated version of figures.

31. line 557: Here the unit for DDR is well written!

    Indeed. As mentioned in response to reviewer's comment #12, we have gone through the manuscript's text and the units have been adapted to the journal's unit format.

32. Table 3 seems to contain the key results of the paper! This could or should be better highlighted in histogram plots or in box-and-whisker plots (25%, 75% percentiles, mean, median, …).

    According to the reviewer's recommendation, the following figure was included in the manuscript:

[Figure]

Figure 7: Visualization of seasonal DDR averages (in mg·m-2·day-1) for the period 12/2006-11/2022, along with the associated variability, for the North-East Atlantic Ocean (NEAO), North-Middle Atlantic Ocean (NMAO), North-West Atlantic Ocean (NWAO), Gulf of Guinea (GG), South-East Atlantic Ocean (SEAO), and South-West Atlantic Ocean (SWAO) sub-domains of the Atlantic Ocean.

33. Table 4 presents an excellent summary and review of sediment-trap climatologies! However, this very long and detailed discussion on arguments, why we should not try to compare sedimentation trap deposition rates and EO deposition rates, is not needed to my opinion.

    Towards verifying the accuracy, ensuring the reliability, and quantifying the uncertainties of the satellite-based estimations of dust deposition rate, implementation of in-situ observations of dust deposition fluxes as reference datasets is essential. As presented in the "sediment-traps evaluation" section, the satellite-derived (EO-based) dust deposition rate product demonstrates a good agreement with sediment trap measurements, of regression slope = 0.85, intercept = 9.49 mg m$^{-2}$ day$^{-1}$, and Pearson correlation coefficient ≈ 0.79. However, it is also discussed that a general apparent tendency for the EO-based estimates is to exceed in situ observations, with mean bias = 5.42 mg m$^{-2}$ day$^{-1}$, relative bias = 19.82%, and RMSE = 30.3 mg m$^{-2}$ day$^{-1}$. The apparent difference between the EO-based dust deposition rate product and the dust deposition rate measurements by in-situ instrumentation raises the following logical question: "Is the difference an overestimation of the EO-based product, an underestimation of the in-situ measurements, or an outcome of both?". As such, the Section "Evaluation of EO-based Dust Deposition estimates" discusses three main components: (i) CALIOP optical products as a source of possible differences in the final output product, (ii) the applied methodology as a possible source of overestimation, and (iii) finally the limitations of the in-situ outputs leading to possible underestimations of the provided measurements. Though (i) and (ii) are not discussed extensively, due to their discussion also in Sections "2.1 Datasets", "2.1.1 CALIPSO-CALIOP", "2.2 Methodology", "2.2.1 Decoupling the atmospheric pure-dust component from the total aerosol load", and "2.2.2 Extracting the atmospheric pure-dust component deposited into the Atlantic Ocean", the uncertainties as possible sources of in-situ measurements discrepancies with the EO-based output product are not discussed in some other section. As such we consider the extended discussion a necessary step towards better understanding and justifying the good performance of the output EO-based dust deposition rate product.

34. line 756: Even after checking the text several times, it seems that Table 5 was never mentioned in the main text body.

The reviewer is right. Reference to "Table 5" has been included as follows:
"Overall, considering the several sources of uncertainties and the methodological factors driving the observed discrepancies, the EO-based dust deposition rate product and the sediment-trap observations are in rather good agreement (slope of 0.85, intercept of 9.49 mg m$^{-2}$ day$^{-1}$, and Pearson correlation coefficient ~0.79). The general trend of satellite-based dust deposition is to overestimate those from the in situ provided observations (mean bias of 5.42 mg m$^{-2}$ day$^{-1}$, relative bias of 19.82%, and RMSE of 30.3 mg m$^{-2}$ day$^{-1}$). However, it is notable that the satellite-based dust deposition rate product consistently reproduces the dust deposition patterns recorded by the sediment traps installed and operated across the broader Atlantic Ocean, with a positional accuracy and magnitude generally within a factor of 2 compared to sediment trap measurements (Table 5).".

35. Section 5 (pages 30-36): The Figures 8-15 and Table 6 should be moved into another article (part 2). The manuscript is heavily overloaded.

The reviewer's recommendation is discussed in the beginning of the our "response to the review's comments" document.

36. line 763: Section title is bad, better: EO-based Dust Deposition Rate vs respective ESM results.

We would like to thank the reviewer for the recommendation. The title was modified.

37. line 770: The AERONET coarse mode AODs may underestimate the total dust AERONET (fine and coarse dust) by 20-30%.

The reviewer is right on this insightful comment. AERONET is a globally distributed network of ground-based remote sensing aerosol monitoring stations of sunphotometers that provides high-quality, standardized, and long-term observations of aerosol optical properties such as AOD, AOD$_{fine-mode}$, AOD$_{coarse-mode}$, AE, which are critical for climate research, satellite validation, and air quality assessments. It is selected as an intercomparison input source of atmospheric remote sensing observations and relative products on the basis of the major strengths of AERONET, and more specifically for its rigorous calibration and data quality control procedures, ensuring consistent and reliable data across all sites. Of high significance for the study are the network's global coverage, spanning urban, rural, coastal, and remote regions, allowing for comprehensive spatial and temporal analysis. In addition, AERONET transparency and scientific utility makes the use of AERONET observations and products an essential tool for understanding atmospheric aerosols and their role in the Earth system. With respect to the reviewer's comment, we fully acknowledge that AERONET coarse-mode AODs may underestimate the total dust AOD by approximately 20–30%. In our study, we specifically used the coarse mode AOD as a proxy for airborne dust because (1) coarse-mode retrievals from AERONET are generally dominated by mineral dust in the dust-source regions and frequently over the outflow regions, (2) fine-mode contributions to dust are relatively smaller and more variable, and (3) on the fact that the use of coarse-mode AOD ensures a more consistent dust signal, minimizing contamination from fine-mode aerosols such as biomass burning or anthropogenic pollution. We fully acknowledge that AERONET observations are not exactly tailored for the present study as airborne in-situ would, however the network provides a unique dataset for this type of validation, evaluation, and cross-comparison activities. For these reasons they are quite frequently implemented, since extremely useful insight in developments may be drawn through AERONET network observation and implementation (e.g., Amiridis et al., 2015; Ansmann et al., 2012; Ansmann et al., 2019; Bibi et al., 2015; Kim et al., 2018; Liu et al., 2014; 2018; Mamouri and Ansmann, 2014; 2017; Marinou et al., 2017; McPherson et al., 2010; de Oliveira et al., 2019; Omar et al., 2013; Proestakis et al., 2018; 2024; Schuster et al., 2012; Wu et al., 2014). However, accounting for the reviewer's remark, the following clarification sentence was included in the manuscript:
"It should be emphasized that implementation of AERONET coarse-mode AODs may underestimate the total dust AERONET (fine- and coarse-mode) by ~20-30% (Mamouri and Ansmann, 2014).".

38. line 789 and Figure 8: It makes obviously no sense to compare monthly mean DOD and deposition fluxes. The uncertainties are in many cases -50 to -100% or 50 to 100%. Integrated CALIOP dust extinction profiles and AERONET DOD should not deviate much (<20%), on a case-by-case basis.

The reviewer is right. In the manuscript, the sentence "The comparison is made on a basis of monthly mean DOD and deposition fluxes" is used. Obviously, it makes no sense to compare DOD and deposition fluxes, since the two observations/products refer to different quantities and processes. The sentence is wrong. What we mean is that (i) DODs are compared against ESMs optical depth outputs and AERONET coarse-mode AOTs and that (ii) the EO-based dust deposition product is compared against ESMs dust deposition outputs. Since the sentence led to confusion it is modified in the updated version of the manuscript. We agree with the reviewer also on the concept that integrated CALIOP dust extinction profiles and AERONET DOD should not deviate much (<20%), on a case-by-case basis. However, this is not an exact spatiotemporal evaluation of AERONET measurements during CALIPSO overpasses in the proximity of the stations within a time-window not allowing mesoscale variability in the atmosphere to modify significantly the observed air masses. The outputs presented are monthly-mean intercompared. The present intercomparison serves as an indirect consistency assessment. The analysis does not constitute a direct validation or an evaluation process for several reasons, primarily due to the fundamental distinction that the datasets represent different physical quantities. More specifically, AERONET measurements and retrievals provide information on the total aerosol load in terms of column-integrated optical properties at 500 nm (AOT), especially during daytime illumination conditions, spanning the entire atmospheric column from the Earth's surface to the top of the atmosphere. In contrast, the satellite-based derived products and ESMs outputs (DOD) represent the column-integrated mean extinction coefficient at 532 nm obtained during both daytime and nighttime. This fundamental difference between the intercomparison datasets, discrepancies related to the distinct characteristics of CALIOP, ESMs, and sunphotometer measurements, discrepancies due to the retrieval algorithms and applied quality assurance criteria, and as a result to the dust decoupling technique, may lead to significant uncertainties, in many cases as high as -50 to -100% or 50 to 100%, as is commented by the reviewer.

39. Further remark: Gray circles on a gray continent (Africa background) are not easy to find.

The figures are extracted directly from AeroVal. The developer keep this note to modify and improve the tool accordingly.

40. line 822: What do we learn from Fig. 9?

The timeseries in Fig. 12 (in the updated version of the manuscript "Fig.9" is "Fig.12") help us to look into how well LIVAS and the models manage to match AERONET observed monthly variability of DOD, which is relevant for analyzing the seasonality in African dust emissions, transport and deposition. In addition to evaluating annual mean dust AOD, this would contribute to strengthening our confidence in the physical/mathematical soundness of the assumptions/process descriptions applied to obtain the dust products presented here. Fig. 12, left panel shows that in general, the observed temporal pattern of coarse dust AOD is reasonably well reproduced by the models (with EMEP and MONARCH being consistently biased low and high, respectively, while EC-Earth3-Iron shifting from overestimation in the cold season to underestimation in the warm season), and also by LIVAS. The monthly profile from LIVAS lies well within the modelled spread of monthly DODs, though it looks smoother because seasonal mean DOD was used for all three months within the season. LIVAS tends to underestimate DOD for January-February, which is particularly pronounced at two sites in Tenerife (La Laguna and Santa Cruz), which could to some extent be explained by using DJF seasonal mean. Similar to EC-Earth3-Iron and EMEP, LIVAS also underestimates in June-July (which is mostly driven by underestimation beyond the main dust emission regions, i.e. at the coastal and island-type sites).
Further, we also present the monthly time series of calculated DOD and coarse-mode AOD at Cabo Verde, which is representative of the dust plume transported westward off the African deserts over the subtropical eastern North Atlantic and thus, provides information regarding the consistency of LIVAS and modelled airborne dust relevant for dust deposition analysis. Thanks to the reviewer's question, we also realized that the scatterplot for the whole 2020 was not necessarily relevant here, therefore it was removed.

The following explanatory text is included in the paper:

"The monthly timeseries of LIVAS and modelled DOD, compared on Figure 12, reflect the seasonality in African dust emissions, transport and deposition. Fig. 12, left panel shows that averaged over the 12 AERONET sites, the observed temporal pattern of coarse dust AOD is reasonably well reproduced by the models (with EMEP and MONARCH being consistently biased low and high, respectively, while EC-Earth3-Iron shifting from overestimation in the cold season to underestimation in the warm season). The monthly profile from LIVAS lies well within the modelled spread of monthly DODs, though it looks smoother because seasonal mean DOD was used for all three months within the season. LIVAS tends to underestimate DOD for January-February, which is particularly pronounced at two sites in Tenerife (La Laguna and Santa Cruz), which could to some extent be explained by using DJF seasonal mean. Similar to EC-Earth3-Iron and EMEP, LIVAS also underestimates in June-July (which is mostly driven by underestimation beyond the main dust emission regions, i.e. at the coastal and island-type sites). Figure 12 (right panel) compares the monthly time series of EO-based and modelled DOD and coarse-mode AOD at Cabo Verde - the location, representative of the dust plume transported westward off the African deserts over the subtropical eastern North Atlantic. LIVAS DOD variation is very close to those simulated by EC-Earth3-Iron and EMEP, underestimating AERONET data, especially in June 2020.".

41. line 841: Instead of Fig. 10, one could present a Table.

According to the reviewer's recommendation the following figure has been changed to a table, as follows.

from:

[Figure]

Figure 13: Overall evaluation statistics in terms of relative bias (NMB%; Fig.13a) and spatial correlation (R-Space; Fig.13b) for the LIVAS DOD at 532 nm and the modelled DOD with respect to AERONET coarse-mode AOD (SDAVL2).

to:

Table: Overall evaluation statistics in terms of relative bias (NMB%; Fig.13a) and spatial correlation (R-Space; Fig.13b) for the LIVAS DOD at 532 nm and the modelled DOD with respect to AERONET coarse-mode AOD (SDA V3 L2).

|  |  | LIVAS | EC-Earth3-Iron | EMEP | MONARCH |
|---|---|---|---|---|---|
| $AOD_{dust}$ | NMB (%): | -22.9 | 8.1 | -47.3 | 32.5 |
| AERONET SDA V3 L2 | R-Space: | 0.95 | 0.76 | 0.55 | 0.87 |

42. line 858: So large mean biases over large areas!!! What is the reason to show all this in a dust deposition paper? Is the conclusion: If the DOD product of the models is so uncertain, what can we expect from modeled dust deposition rates? Does it really make sense to compare observations and modeling on the AOD or DOD level? Furthermore, the differences in the results are partly so large. What is the message, what do we learn?

It is not the author's intention, in any way, to diminish the huge effort by modelling groups towards the developments made. It should be emphasized that the figures provide the comparison between the satellite-based LIVAS DOD product and EC-Earth3-Iron, MONARCH, and EMEP DOD outputs in percentage (%) and not as absolute differences. As it can be seen, larger difference in percentage is in areas where the atmospheric load of dust is extremely low. Considering the several sources of differences between the intercomparison datasets, the fact that the differences (i) over the dust emission sources and (ii) over the main dust transport pathways are in general below 25% is rather encouraging, denoting that the effort made by modelling groups is towards the right direction in terms of atmospheric dust modelling.

43. As mentioned, better move the entire Section 5 into another article. The comparison of DOD and deposition rates (observations vs modeling) is an independent story, a new article! The main goal of this article (part 1) should be just EO dust deposition rates in the context with other dust deposition studies performed during the last 4-5 decades.

The reviewer's recommendation is discussed in the beginning of the our "response to the review's comments" document.

44. Section 6 (should remain in part 1).

The reviewer's recommendation is discussed in the beginning of the our "response to the review's comments" document.

45. line 926: Table 7 should have 3 columns instead of three lines.

According to the reviewer's recommendation, Table 7 providing the annual-mean and seasonal-mean dust deposition (Tg, 1 Tg = $10^{12}$ g) and Normalized Standard Deviation (%) into the broader Atlantic Ocean (60°S-40°N, 100°W-20°E) on the basis of sixteen full years of CALIPSO observations (12/2006-11/2022), was modified from:

|  | Annual | DJF | MAM | JJA | SON |
|---|---|---|---|---|---|
| Dust Deposition (Tg) | 274.79 ± 31.64 | 63.78 ± 12.03 | 77.3 ± 19.93 | 93.10 ± 11.65 | 40.62 ± 11.44 |
| NSD (%) | ~36.03% | ~18.87% | ~25.78% | ~12.52% | ~28.16% |

to:

|  | Dust Deposition (Tg) | NSD (%) |
|---|---|---|
| Annual | 274.79 ± 31.64 | ~36.03% |
| DJF | 63.78 ± 12.03 | ~18.87% |
| MAM | 77.3 ± 19.93 | ~25.78% |
| JJA | 93.10 ± 11.65 | ~12.52% |
| SON | 40.62 ± 11.44 | ~28.16% |

46. line 933: Open circles and thin dashed line in Fig. 16 is not explained in the caption or in the figure.

According to the reviewer's notice, description of the open circles was included. In addition, since the dashed line seemed to create confusion, it was removed, and the figure was modified as follows:

[Figure]

Figure: Quantification of the (a) total annual-mean and (b) seasonal-mean dust deposition (Tg yr$^{-1}$, 1 Tg = 10$^{12}$ g) into the broader Atlantic Ocean (60°S-40°N, 100°W-20°E) on the basis of 16-years of CALIPSO observations (12/2006-11/2022). Open circles in blue colour denote the annual-mean (left) and seasonal-mean (right) Normalized Standard Deviation (NSD).

47. Figure 17: What show the colored horizontal bars, what show the colored horizontal lines, what show the open circles? Please add this information.

According to the reviewer's comment, the caption of the figure was modified to include the missing information, as follows:
"Figure 10: Comparison between the present satellite-based estimates of dust deposition (12/2006-11/2022 average) across (a) the broader Atlantic Ocean (AO), (b) North Atlantic Ocean (NAO), and (c) South Atlantic Ocean (SAO) with documented estimates of dust deposition. In each sub-figure, the horizontal blue and red lines correspond to the EO-based long-term mean dust deposition and the reported in relevant scientific studies mean dust deposition, respectively, while the light blue and light red shaded enveloped areas correspond to their standard deviation. The open circles in red colour provide the quantified dust deposition within the Atlantic Ocean region reported by relevant studies."
Please note that following rearrangement of the sections, as recommended, the figure numbering has also changed.

48. Section 7 must be re-written (adjusted to part 1 and part 2). Summaries should be compact, highlight the most important aspects only, and present an outlook.

According to the reviewer's comment the summary was re-written, reduced by ~40% in length and becoming significantly more dense, summarizing and highlighting the most important aspects, and providing an outlook of the study.

49. My personal outlook-related question: Since CALIOP/LIVAS provides global dust profiles, one could compute dust deposition rates as a function of height, e.g., flux from the layer at 4-6 km to the layer below 4 km, etc. In this way one may learn a lot about residence times of dust at different heights and dust particle sedimentation behavior and speed.

We sincerely thank the reviewer for this thoughtful and stimulating suggestion. We fully agree that the proposed idea, to compute dust deposition rates as a function of height, is highly interesting and could open new and valuable directions for research relative to dust residence times at different heights and dust particle sedimentation behavior and speed. There is the possibility that LIVAS has the capacity to facilitate this analysis, however several spatiotemporal resolution sensitivity tests will have to be performed to address the feasibility and in addition the challenging issue of existence of tailored validation/evaluation datasets of the outcomes will have to be tackled. While it falls somewhat outside the immediate scope of the present study, we recognize its potential to significantly enrich the field. We will certainly take this suggestion into careful consideration and aim to address it in the framework of a future study. We greatly appreciate the reviewer's contribution to enhancing the broader perspective of our work.